# Clip-and-Verify: Linear Constraint-Driven Domain Clipping for Accelerating Neural Network Verification

**Duo Zhou**[*]    **Jorge Chavez**[*]    **Hesun Chen**    **Grani A. Hanasusanto**    **Huan Zhang**

University of Illinois Urbana-Champaign    [*]Equal Contribution

{duozhou2,jorgejc2,hesunc2,gah}@illinois.edu, huan@huan-zhang.com

## Abstract

State-of-the-art neural network (NN) verifiers demonstrate that applying the branch-and-bound (BaB) procedure with fast bounding techniques plays a key role in tackling many challenging verification properties. In this work, we introduce the *linear constraint-driven clipping* framework, a class of scalable and efficient methods designed to enhance the efficacy of NN verifiers. Under this framework, we develop two novel algorithms that efficiently utilize linear constraints to 1) reduce portions of the input space that are either verified or irrelevant to a subproblem in the context of branch-and-bound, and 2) directly improve intermediate bounds throughout the network. The process novelly leverages linear constraints that often arise from bound propagation methods and is general enough to also incorporate constraints from other sources. It efficiently handles linear constraints using a specialized GPU procedure that can scale to large neural networks without the use of expensive external solvers. Our verification procedure, Clip-and-Verify, consistently tightens bounds across multiple benchmarks and can significantly reduce the number of subproblems handled during BaB. We show that our clipping algorithms can be integrated with BaB-based verifiers such as $\alpha,\beta$-CROWN, utilizing either the split constraints in activation-space BaB or the output constraints that denote the unverified input space. We demonstrate the effectiveness of our procedure on a broad range of benchmarks where, in some instances, we witness a 96% reduction in the number of subproblems during branch-and-bound, and also achieve state-of-the-art verified accuracy across multiple benchmarks. Clip-and-Verify is part of the $\alpha,\beta$-`CROWN` verifier, **the VNN-COMP 2025 winner**. Code available at https://github.com/Verified-Intelligence/Clip_and_Verify.

## 1 Introduction

The neural network (NN) verification problem is imperative in mission-critical applications [70, 63, 58, 16, 75, 72] where formally proving properties such as safety and robustness over a specified input domain is essential. Recent approaches make neural network verification more tractable by relaxing the original non-convex problem into convex formulations that are amenable to linear programming (LP) [25, 68], semidefinite programming (SDP) [12, 26, 50, 40, 18, 19], and bound propagation-based solvers [77, 29, 55, 67, 24, 66, 64]. These convex relaxations can be further strengthened by tightening single-neuron relaxations through convex geometric analysis [55], constructing convex-hull approximations to capture multi-input dependencies [48], and introducing cutting planes that encode inter-neuron dependencies [76, 79]. To handle properties that cannot be certified by a single relaxation, state-of-the-art verifiers couple bound propagation methods with the branch-and-bound (BaB) paradigm [14, 66, 53, 13] as this technique can be efficiently parallelized and scaled on GPUs. Linear bound propagation methods such as CROWN [77] recursively compute bounds on the activations of each layer, referred to as *intermediate bounds*, which serve as critical building blocks for determining the tightness of the overall relaxation and for guiding the BaB search.

39th Conference on Neural Information Processing Systems (NeurIPS 2025).

While BaB and cutting-plane techniques can further refine the relaxation by incorporating additional constraints at the final layer, they cannot *directly* and *efficiently* improve the intermediate bounds themselves. In BaB, where the number of subproblems grows exponentially, loose intermediate bounds weaken the relaxation, resulting in deeper branching and longer verification times. Although algorithms such as $\beta$-CROWN [66] theoretically support optimizing bounds at intermediate layers, doing so in practice is prohibitively expensive as the number of hidden neurons typically outnumber the output neurons used for property verification by several orders of magnitude. Consequently, updating bounds for all intermediate layers introduces significant computational overhead that far outweigh the gains brought from their tighter convex relaxations. As a result, existing implementations fix the global intermediate bounds computed at initialization (e.g., via $\alpha$-CROWN [74]) and focus their optimization efforts solely at the final layer. While this design choice preserves scalability, it limits the ability to tighten the relaxation throughout the network, motivating methods that can refine intermediate bounds both *effectively* and *efficiently*.

To this end, we introduce **Clip-and-Verify**, a verification pipeline designed to enhance NN verifiers by opportunistically refining bounds at **any layer with minimal computational overhead**. Our core insight is that the bounding planes generated by linear bound propagation at all layers naturally align with our pipeline, enabling us to exploit their geometry to eliminate infeasible regions of the input domain and prune redundant subproblems early in verification. We formalize the task of tightening any layer's bounds as the objective of our linear constraint-driven clipping framework, and we propose two novel algorithms: complete clipping, which directly optimizes neurons' bounds via a specialized coordinate ascent procedure, and relaxed clipping, which refines the input domain to enhance the intermediate relaxations and consequently improves the NN's bounds. An intuitive illustration of this refinement is given in Figure 1, and our main contributions are as follows:

- We propose two specialized GPU algorithms within our novel *linear constraint-driven clipping* framework for tightening bounds at **any layer**, preserving the scalability of state-of-the-art NN verifiers without relying on external solvers. *Relaxed clipping* optimizes the input domain bounds as a proxy, offering good improvements with little costs, while *complete clipping* employs a customized coordinate ascent solver to directly refine the bounds at **every layer** of the NN.

- We show that linear bound propagation methods produce linear constraints that can be obtained "for free" during both input and activation BaB procedures. By leveraging these cheaply available constraints and integrating our two efficient clipping algorithms, we introduce **Clip-and-Verify**, a verification pipeline that tightens the all neurons bounds with minimal computational overhead.

- Across a large number of benchmarks from the Verification of Neural Networks Competition (VNN-COMP) [11, 10] and existing literature, we demonstrate that our Clip-and-Verify framework is capable of reducing the number of BaB subproblems by as much as 96% and consistently verifying more properties on benchmarks from the Verification of Neural Networks Competition.

## 2 Preliminaries

**The NN Verification Problem.** Given some input $\boldsymbol{x}$ belonging to the set $\mathcal{X}$, and a feed-forward network $f(\cdot)$ with general activation functions, the goal of NN verification can be formulated as verifying $f(\boldsymbol{x}) \geq 0$ for all inputs in $\mathcal{X}$. One manner of verifying this property involves solving $\min_{\boldsymbol{x} \in \mathcal{X}} f(\boldsymbol{x})$, which is challenging due to the non-convexity of the NN and is generally NP-complete [35]. On the other hand, convex-relaxation algorithms compute a sound, approximate lower bound, $\underline{f}(\boldsymbol{x})$, to the network's true minimum such that when $\underline{f}(\boldsymbol{x}) > 0$, the property is sufficiently verified, otherwise the problem is unknown without further refinement or falsification.

**Bound Propagation.** Linear bound propagation methods [56, 55, 64, 5, 77] approximate neuron bounds layer by layer by relaxing the nonlinearities of activation functions, making these techniques a fast and popular approach for NN verification. For an $L$-layered, feedforward network, the bounds for the $j^{\text{th}}$ neuron at the $i^{\text{th}}$ layer may be expressed as:

$$\underline{\boldsymbol{z}}_j^{(i)} := \min_{\boldsymbol{x} \in \mathcal{X}} \underline{\mathbf{A}}_j^{(i)\top} \boldsymbol{x} + \underline{\mathbf{c}}_j^{(i)} \leq \boldsymbol{z}_j^{(i)}, \quad \overline{\boldsymbol{z}}_j^{(i)} := \max_{\boldsymbol{x} \in \mathcal{X}} \overline{\mathbf{A}}_j^{(i)\top} \boldsymbol{x} + \overline{\mathbf{c}}_j^{(i)} \geq \boldsymbol{z}_j^{(i)} \tag{1}$$

When $\mathcal{X}$ is an $\ell_\infty$ box (i.e. $\{\boldsymbol{x} \mid \|\boldsymbol{x} - \hat{\boldsymbol{x}}\|_\infty \leq \boldsymbol{\epsilon}\}$), we may "concretize" the lower and upper bounds of the layer using Lemma B.1: $\underline{\boldsymbol{z}}^{(i)} = \underline{\mathbf{A}}^{(i)}\hat{\boldsymbol{x}} - |\underline{\mathbf{A}}^{(i)}|\boldsymbol{\epsilon} + \underline{\mathbf{c}}^{(i)}$ and $\overline{\boldsymbol{z}}^{(i)} = \overline{\mathbf{A}}^{(i)}\hat{\boldsymbol{x}} + |\overline{\mathbf{A}}^{(i)}|\boldsymbol{\epsilon} + \overline{\mathbf{c}}^{(i)}$. Once concretized, the post-activation neuron, $\hat{\boldsymbol{z}}_j^{(i)}$, may be bounded (e.g. via Planet relaxation [25] for

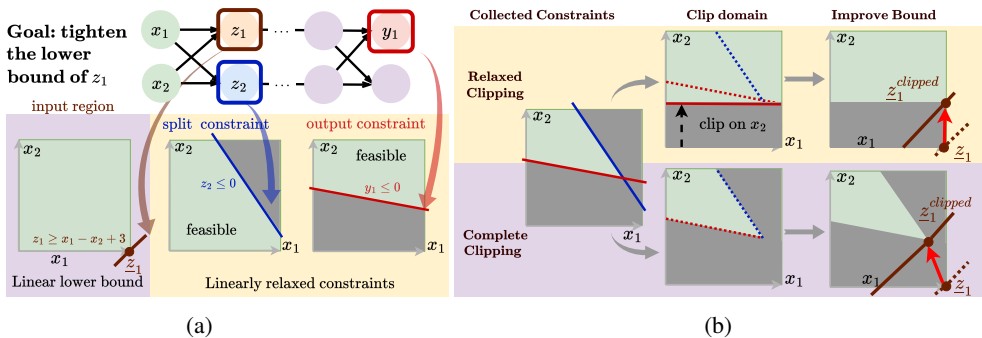

(a)                  (b)

Figure 1: (a) Linear bound propagation produces linear bounds on all neurons w.r.t. the input. These linear bounds are later used as linearly relaxed constraints. In figure (a), the blue and the red lines are used as the linearly relaxed boundary of constraint $z_2 \leq 0$ and $y_1 \leq 0$. Our goal is to further tighten the lower bound of $z_1$ via these constraints; (b) Linear constraints (e.g. split constraint $z_2 \leq 0$ and output constraint $y_1 \leq 0$) can be applied to shrink the input region and provide tighter bounds. In *Relaxed Clipping*, the feasible region is relaxed to its tightest covering box. In *complete clipping*, the infeasible region is completely cropped off, leaving the exact feasible region to improve bounds on. Both two clipping methods improve the bound to $\underline{z}_1^{clipped}$, but due to the relaxation nature, Relaxed Clipping yields a looser $\underline{z}_1^{clipped}$. A detailed numerical example is given in Appendix C.5.

ReLU). The lower/upper bounding planes, $\underline{A}^{(i)}/\overline{A}^{(i)}$ and $\underline{c}^{(i)}/\overline{c}^{(i)}$, are produced via backpropagation, and their definitions are given in Appendix A. The lower bound at the final layer, $\underline{z}^{(L)}$, is used to determine if the problem is verified.

**Branch-and-Bound.** The BaB paradigm [14, 66, 53, 76, 79, 21, 65, 25, 45] systematically partitions the verification problem into smaller subproblems, $\mathcal{X} = \mathcal{X}_1 \cup \mathcal{X}_2$, enabling tighter bounds on each subdomain. BaB can split upon the input space (e.g. axis-aligned constraints) or the activation space (e.g. split activation neurons). See Appendix A.1 for the formal definition and complexity implications. The exponential growth of subdomains can lead to high computational cost but can be mitigated by verifying domains early and often, a direct consequence of *Clip-and-Verify*.

## 3 Enhancing Neural Network Verification with Clip-and-Verify

### 3.1 Motivation and Overview

BaB-based verifiers manage complexity by partitioning the input or activation space into tractable subproblems. Within each branch, linear relaxations provide bounds on the network's neurons which can guide branching decisions and verify properties. Crucially, the constraints introduced at each BaB split implicitly define a tighter feasible input domain and offer opportunities to refine the bounds at any layer. As shown in Fig. 1b, linear constraints can shrink an $\ell_\infty$-norm ball during robustness verification, yielding tighter intermediate bounds and enabling more effective search-space pruning.

In large NNs, full linear bound propagation is often limited to an initial pass for efficiency. Alternative methods like using LPs to update intermediate bounds [51] or full re-propagation after each split also prove too costly. Such overhead restricts frequent bound updates in deep NNs or extensive branching.

To fully exploit these opportunities, we propose *Clip-and-Verify*, a pipeline which tightens the input box and re-concretizes intermediate bounds at each BaB node using linear constraints (e.g. activation split and final layer bound), without running a full pass. It contains two algorithms: *Complete Clipping*, which performs a fast coordinate-wise dual search with sorted breakpoints per constraint to obtain near-LP tightening at a fraction of the cost, and *Relaxed Clipping*, which performs per-constraint *exact* tightening for axis-aligned boxes by solving the one-dimensional dual in closed form and then re-concretizes cached linear constraints over the shrunk box.

### 3.2 Complete Clipping: Optimizing Intermediate Bounds Directly via Linear Constraints

**Exact Bound Refinement with a Single Constraint via Optimized Duality.** Suppose we wish to improve the lower bound of a linear function $\boldsymbol{a}^\top \boldsymbol{x} + c$ over an input domain $\mathcal{X}$, given a new linear

constraint $\boldsymbol{g}^\top \boldsymbol{x} + h \leq 0$. The primal optimization problem is:

$$L^\star = \min_{\boldsymbol{x} \in \mathcal{X}} \{\boldsymbol{a}^\top \boldsymbol{x} + c : \boldsymbol{g}^\top \boldsymbol{x} + h \leq 0\} \tag{2}$$

Instead of directly solving this potentially high-dimensional LP, we formulate its Lagrangian dual. The key advantage and insight is that, for a fixed dual variable $\beta \in \mathbb{R}_+$, the inner minimization over $\boldsymbol{x}$ can be solved analytically for box domains, transforming the problem into a simpler optimization over a **single** dual variable. This leads to the following theorem:

**Theorem 3.1** (Exact Bound Refinement under a Single Linear Constraint). *Let* $\boldsymbol{a} \in \mathbb{R}^n$, $c \in \mathbb{R}$, $\boldsymbol{g} \in \mathbb{R}^n$, $h \in \mathbb{R}$, *and the input domain be* $\mathcal{X} = \{\boldsymbol{x} \mid \hat{\boldsymbol{x}} - \boldsymbol{\epsilon} \leq \boldsymbol{x} \leq \hat{\boldsymbol{x}} + \boldsymbol{\epsilon}\}$. *The optimal value* $L^\star$ *of the constrained minimization problem (2) is given by the solution to the dual problem:*

$$L^\star = \max_{\beta \in \mathbb{R}_+} \underbrace{(\boldsymbol{a} + \beta\boldsymbol{g})^\top \hat{\boldsymbol{x}} - \sum_{j=1}^{n} |(\boldsymbol{a} + \beta\boldsymbol{g})_j| \epsilon_j + c + \beta h}_{\textit{Dual Objective } D(\beta)} \tag{3}$$

$D(\beta)$ *in (3) is concave and piecewise-linear in* $\beta \in \mathbb{R}_+$. *Its maximum* $L^\star$ *and the optimal* $\beta^\star$ *can be determined exactly and efficiently by identifying its breakpoints (values of* $\beta$ *where* $(\boldsymbol{a} + \beta\boldsymbol{g})_j = 0$ *for some* $j \in [n]$) *and analyzing the super-gradients within the resulting linear segments.*

For proof, see Appendix B.2. Theorem 3.1 converts a potentially expensive $n$-dimensional LP into a 1D concave maximization problem ($D(\beta)$) that can be solved without iterative gradient methods. The process of finding breakpoints and the optimal segment (detailed below and in Algorithm 1 for the multi-constraint case) is highly amenable to efficient computation, making it suitable for refining bounds across many neurons and subdomains **in parallel**. An analogous theorem holds for tightening the upper bound by solving $\max_{\boldsymbol{x} \in \mathcal{X}} \{\boldsymbol{a}^\top \boldsymbol{x} + c \mid \boldsymbol{g}^\top \boldsymbol{x} + h \leq 0\}$.

Before optimizing Eq. (3), infeasibility can be detected *a priori*: the problem is infeasible iff $\boldsymbol{g}^\top \hat{\boldsymbol{x}} + h - \sum_{i=1}^{n} |g_i| \epsilon_i > 0$. Such a scenario arises when the property being verified imposes constraints unsatisfiable within the current input domain $\mathcal{X}$ (e.g., contradictory ReLU assignments), and may be considered *verified* without further refinement.

Note that our primal optimization problem is mathematically equivalent to a continuous knapsack problem which can be obtained via a change of variables as detailed in Appendix B.3. The breakpoints $\beta_i = -a_i/g_i$ in our dual formulation are identical to the efficiency ratios $r_j/s_j$ used in the standard greedy knapsack algorithm. Thus, our dual-based solver and the greedy knapsack algorithm are equivalent, both finding the provably optimal solution with $\mathcal{O}(n \log n)$ complexity.

**Coordinate Ascent for Multiple Constraints** When multiple linear constraints $\boldsymbol{Gx} + \boldsymbol{h} \leq \boldsymbol{0}$ (where $\boldsymbol{G} \in \mathbb{R}^{m \times n}$, $\boldsymbol{h} \in \mathbb{R}^m$) are available, the dual problem involves optimizing multiple Lagrange multipliers $\boldsymbol{\beta} \in \mathbb{R}_+^m$:

$$L^\star = \max_{\boldsymbol{\beta} \in \mathbb{R}_+^m} (\boldsymbol{a} + \boldsymbol{\beta}^\top \boldsymbol{G})^\top \hat{\boldsymbol{x}} - \sum_{j=1}^{n} |(\boldsymbol{a} + \boldsymbol{\beta}^\top \boldsymbol{G})_j| \epsilon_j + c + \boldsymbol{\beta}^\top \boldsymbol{h} \tag{4}$$

The single constraint problem (3) is easy to solve using Theorem 3.1, thus we can use coordinate ascent to solve (4). Algorithm 1 details a single pass of this coordinate ascent procedure. This iterative approach optimizes one dual variable $\beta_k$ at a time, keeping others fixed. Each step thus reduces to solving the 1D problem described in Theorem 3.1 (with $\boldsymbol{a}$ replaced by $\boldsymbol{a} + \sum_{p \neq k} \beta_p \boldsymbol{G}_{p,:}$ and $\boldsymbol{g}$ replaced by $\boldsymbol{G}_{k,:}$). Since the dual objective remains concave and piecewise-linear along each coordinate $\beta_k$, we can efficiently exploit the breakpoint structure for each update. An order dependency is discussed in Appendix B.5. Algorithm 1 is significantly more efficient and scalable than using general-purpose LP solvers, even when those solvers are used as fast heuristics to get a bound rather than converge to optimal. We conducted a detailed comparison in Appendix D.3, integrating a state-of-the-art LP solver (Gurobi) into our BaB framework. The results show that even when running dual simplex with 10-iteration limit, the LP solver was over **880× slower** than our GPU-parallelized coordinate ascent (0.0028s vs. 2.47s per round) while achieving comparable bound accuracy (0.00085 vs 0.0007 mean error).

**Algorithm 1** Coordinate Ascent with Multiple Constraints

---

**Require:** Objective $\boldsymbol{a}^\top \boldsymbol{x} + c$: $\boldsymbol{a} \in \mathbb{R}^n$, $c \in \mathbb{R}$; Constraints $\boldsymbol{G}\boldsymbol{x} + \boldsymbol{h} \le 0$: $\boldsymbol{G} \in \mathbb{R}^{m \times n}$, $\boldsymbol{h} \in \mathbb{R}^m$.

1: $\boldsymbol{\beta} \leftarrow [0, \ldots, 0]^\top$ { Initialize vector of $m$ Lagrange multipliers as zero}

2: $\hat{\boldsymbol{x}} \leftarrow \frac{\overline{\boldsymbol{x}} + \underline{\boldsymbol{x}}}{2}$, $\boldsymbol{\epsilon} \leftarrow \frac{\overline{\boldsymbol{x}} - \underline{\boldsymbol{x}}}{2}$ { Initialize centroid and the radius of the input hyper-rectangular}

3: **for** constraint $k$ in $[m]$ **do**

4: $\quad \boldsymbol{q} \leftarrow -(\boldsymbol{a} + \boldsymbol{\beta}^\top \boldsymbol{G})/\boldsymbol{G}_{k,:}$ {Calculate the breakpoints}

5: $\quad \boldsymbol{I} \leftarrow \operatorname{argsort}(\boldsymbol{q})$

6: $\quad \boldsymbol{g} \leftarrow |\boldsymbol{G}_{k,:}| \odot \boldsymbol{\epsilon}$ {Scale by box half-widths}

7: $\quad \boldsymbol{g}_{\text{sorted}} \leftarrow \boldsymbol{g}_{\boldsymbol{I}}$ {Reorder by the same argsort index $\boldsymbol{I}$}

8: $\quad \boldsymbol{g}_i^{(-)} \leftarrow -\sum_{j=1}^i (|\boldsymbol{g}_{\text{sorted}}|_j), j \in [n]$ {(Negative) cumulative sum of sorted breakpoints}

9: $\quad \boldsymbol{g}^{(+)} \leftarrow \boldsymbol{g}^{(-)} - \boldsymbol{g}_n^{(-)}$ {Shift $\boldsymbol{g}^{(-)}$ to its positive range}

10: $\quad \nabla \boldsymbol{g} \leftarrow \boldsymbol{g}^{(+)} + \boldsymbol{g}^{(-)} + \boldsymbol{G}_{k,:}^\top \hat{\boldsymbol{x}} + \boldsymbol{h}_k$ {Supergradient; monotone nondecreasing in $i$}

11: $\quad i^\star \leftarrow \min\{i \in [n] : \nabla \boldsymbol{g}_i \le 0\}$ {First sign change}

12: $\quad j^\star \leftarrow \boldsymbol{I}_{i^\star}$ {Map the $i^\star$-th breakpoint in the sorted order back to the original coordinate index}

13: $\quad \boldsymbol{\beta}_k \leftarrow \max\{\boldsymbol{q}_{j^\star}, 0\}$ {Ensure feasibility}

14: $\quad l^\star \leftarrow -\left\| \boldsymbol{a} + \boldsymbol{\beta}^\top \boldsymbol{G} \right\|_1 \cdot \boldsymbol{\epsilon} + \left( \boldsymbol{a} + \boldsymbol{\beta}^\top \boldsymbol{G} \right) \hat{\boldsymbol{x}} + \boldsymbol{\beta}^\top \boldsymbol{h} + c$ {Calculate dual objective}

**Ensure:** $l^\star$

---

### 3.3 Relaxed Clipping: Optimizing Intermediate Bounds Indirectly via Input Refinement

Although the coordinate ascent is significantly efficient compared to LP solvers, it is still computationally expensive when there are various subproblems with many unstable neurons and multiple constraints. To address this issue, we propose a more efficient *Relaxed Clipping* algorithm, which **shrinks the input domain** by leveraging linear constraints (e.g., from final-layer outputs or activation branchings) to shrink axis-aligned portions of the input. By adopting a box input domain as a proxy, we avoid repeatedly solving linear programs for each intermediate-layer neuron, and can solve the resulting optimization problem efficiently via the dual norm without relying on external solvers. The remaining task is to determine the clipped box $\mathcal{X}' \subseteq \mathcal{X}$ that best respects the constraints.

**Formulating Relaxed Clipping for a Hyper-Rectangle.** We assume a hyper-rectangular input region and a set of $m$ linear constraints $\boldsymbol{A}\boldsymbol{x} + \boldsymbol{c} \le 0$, with $\boldsymbol{A} \in \mathbb{R}^{m \times n}$ and $\boldsymbol{c} \in \mathbb{R}^m$. To refine the lower and upper bounds of the input along each input dimension $i \in [n]$, we solve:

$$\underline{\boldsymbol{x}}_i := \min_{\boldsymbol{x} \in \mathcal{X}} \{\boldsymbol{x}_i \mid \boldsymbol{A}\boldsymbol{x} + \boldsymbol{c} \le 0\}, \qquad \overline{\boldsymbol{x}}_i := \max_{\boldsymbol{x} \in \mathcal{X}} \{\boldsymbol{x}_i \mid \boldsymbol{A}\boldsymbol{x} + \boldsymbol{c} \le 0\}. \tag{5}$$

These refined bounds, $\underline{\boldsymbol{x}}_i$ and $\overline{\boldsymbol{x}}_i$, **clip** the original domain $\mathcal{X}$ to reflect only the portion that satisfies all the linear constraints. We derive a simple **closed-form solution** when there is only one linear inequality, $\boldsymbol{a}^\top \boldsymbol{x} + c \le 0$, foregoing the need to solve (5) via LP solvers or gradient-based methods.

**Theorem 3.2** (Relaxed clipping under a single constraint). *Let $\boldsymbol{x} \in \mathcal{X}$ and $\boldsymbol{a}^\top \boldsymbol{x} + c \le 0$ be the sole constraint. For brevity, denote the closed-form solution as $\boldsymbol{x}_i^{(clip)} = (-\sum_{j \ne i} \{\boldsymbol{a}_j \hat{\boldsymbol{x}}_j - |\boldsymbol{a}_j| \boldsymbol{\epsilon}_j\} - c)/\boldsymbol{a}_i$. Then, for each coordinate $i$, the new upper (or lower) bound is updated as follows:*

$$\begin{cases} \overline{\boldsymbol{x}}_i^{(new)} = \min\left\{\boldsymbol{x}_i^{(clip)}, \overline{\boldsymbol{x}}_i\right\} & \textit{if } \boldsymbol{a}_i > 0 \\ \underline{\boldsymbol{x}}_i^{(new)} = \max\left\{\boldsymbol{x}_i^{(clip)}, \underline{\boldsymbol{x}}_i\right\} & \textit{if } \boldsymbol{a}_i < 0 \\ \textit{no change} & \textit{otherwise} \end{cases}$$

Appendix B.4 gives a proof showing that, for a single linear constraint $\boldsymbol{a}^\top \boldsymbol{x} + c \le 0$, we can **clip** the box $\mathcal{X}$ to a new box $\mathcal{X}'$ in one pass using the closed-form updates in Theorem 3.2, without any external solvers. The resulting box is the *tightest axis-aligned over-approximation* of the feasible set $\mathcal{X} \cap \{\boldsymbol{x} : \boldsymbol{a}^\top \boldsymbol{x} + c \le 0\} \subseteq \mathcal{X}' \subseteq \mathcal{X}$, and it is *component-wise tight*: no coordinate bound of $\mathcal{X}'$ can be further refined while remaining a box that still contains the feasible set. The computation costs only $O(n)$ arithmetic operations. For example, re-propagating bounds for a network with 3 intermediate layers (4096, 2048, and 100 neurons respectively) and containing 800-1600 unstable neurons can take 10s per subdomain, versus 0.3s when re-concretizing the intermediate bounds.

This Relaxed Clipping step complements Complete Clipping. Whereas Complete Clipping solves problem (3) *for each neuron*, Relaxed Clipping tightens the shared input box $\mathcal{X}'$ *once*, and then

all intermediate bounds are cheaply re-concretized over this tighter box, yielding network-wide bound improvements from a single cheap update. By keeping the shared domain as a box, we avoid polyhedral operations and repeated per-neuron optimizations, substantially improving scalability.

**Clipping for Multiple Constraints.** When several linear constraints are present, we apply Theorem 3.2 in *parallel* to each constraint in $\boldsymbol{A}\boldsymbol{x} + \boldsymbol{c} \leq 0$. Algorithm 2 outlines this procedure: Given the original box bounds, $\underline{\boldsymbol{x}}$ and $\overline{\boldsymbol{x}}$, we compute its center and radius, $\hat{\boldsymbol{x}}$ and $\boldsymbol{\epsilon}$. Then, for each constraint $k$ and dimension $i$, we apply Theorem 3.2 independently and in parallel to refine $\underline{\boldsymbol{x}}_i$ and $\overline{\boldsymbol{x}}_i$. After processing all constraints, the resulting clipped bounds are aggregated, and the tightest bounds are selected to form the final clipped domain. This formulation preserves scalability and supports parallelization. We present a sequential variation in Appendix C.2 that foregoes parallelization for further refinement in which the box center and radius are recalculated after each constraint is applied. Nonetheless, we emphasize our current formulation as a core strength of domain clipping, maintaining both efficiency and effectiveness. Similar to direct clipping, Algorithm 2 may identify *infeasibility* by returning clipped bounds where $\underline{\boldsymbol{x}}$ *is larger* than $\overline{\boldsymbol{x}}$ along some dimension(s).

---

**Algorithm 2** Linear Constraint-Driven Relaxed Clipping *(Parallel)*

---

**Require:** $\underline{\boldsymbol{x}}$ : Input lower bounds; $\overline{\boldsymbol{x}}$ : Input upper bounds; $\boldsymbol{A}\boldsymbol{x} \leq \boldsymbol{c}$ : Constraints.
1: $\hat{\boldsymbol{x}} \leftarrow \frac{\overline{\boldsymbol{x}}+\underline{\boldsymbol{x}}}{2}, \boldsymbol{\epsilon} \leftarrow \frac{\overline{\boldsymbol{x}}-\underline{\boldsymbol{x}}}{2}, \boldsymbol{x}^{\text{(clipped)}} \leftarrow \underline{\boldsymbol{x}}, \overline{\boldsymbol{x}}^{\text{(clipped)}} \leftarrow \overline{\boldsymbol{x}}$ { Initialize the original bounds}
2: **for** each constraint $k \in \{1, \ldots, \text{rows}(\boldsymbol{A})\}$ **do**
3:     **for** each input dimension $i \in \{1, \ldots, \text{cols}(\boldsymbol{A})\}$ **do**
4:         $\boldsymbol{x}_i^{\text{(new)}} \leftarrow \frac{-\sum_{j \neq i} \boldsymbol{A}_{k,j} \hat{\boldsymbol{x}}_j + \sum_{j \neq i} |\boldsymbol{A}_{k,j}| \boldsymbol{\epsilon}_j - \boldsymbol{c}_k}{\boldsymbol{A}_{k,i}}$ {Update $x$ using Theorem 3.2}
5:         **if** $\boldsymbol{A}_{k,i} \geq 0$ **then**
6:             $\overline{\boldsymbol{x}}_i^{\text{(clipped)}} \leftarrow \min(\overline{\boldsymbol{x}}_i^{\text{(clipped)}}, \boldsymbol{x}_i^{\text{(new)}})$ {Iteratively tighten the upper bound}
7:         **else**
8:             $\underline{\boldsymbol{x}}_i^{\text{(clipped)}} \leftarrow \max(\underline{\boldsymbol{x}}_i^{\text{(clipped)}}, \boldsymbol{x}_i^{\text{(new)}})$ {Iteratively tighten the lower bound}
**Ensure:** $\underline{\boldsymbol{x}}^{\text{(clipped)}}, \overline{\boldsymbol{x}}^{\text{(clipped)}}$

---

We have discussed how to use the linear constraints to tighten bounds, the next step is to find these linear constraints. We can use output constraints or activation split constraints during BaB. The next two sections we will discuss how to find the constraints and incorporate our algorithm into BaB.

## 3.4 Integrating Clipping into Input BaB Verification

In input BaB, the verifier partitions the network's *input region* (often an $\ell_\infty$-box) along axis-aligned splits, generating multiple subdomains $\{\mathcal{X}_i\}_{i=1}^b$. Each subdomain is then fed into a bound-propagation verifier (e.g., CROWN) to obtain linear hyper-planes that are used to lower-bound the *final-layer output* with respect to the input. Existing verifiers typically use these final-layer hyperplanes *only* to compute the bounds and decide whether the *entire* subdomain can be immediately verified or must be further subdivided. It may be the case that the hyperplanes used for computing the final layer bounds are tight enough to verify a subset of the input, but not tight enough to verify the input in its entirety (see Fig. 4 in Appendix). The *key insight* is that we can remove infeasible parts in a subproblem via linear constraints and then consider a partially verified problem in the next iteration of BaB. These already calculated hyperplanes can then be effectively re-used as constraints, making them appropriate for our clipping paradigms.

**Using Final-Layer Bounds for Clipping.** After a subdomain has been bounded, the final-layer's bounding plane(s) are retained after *concretization* if the subdomain cannot be fully verified. For the verification objective, $f(\boldsymbol{x}) \geq 0$, it suffices to verify the input region, $\{\boldsymbol{x} \mid \underline{\boldsymbol{a}}^{(L)\top}\boldsymbol{x} + \underline{c}^{(L)} \geq 0\}$. In this case, the bounding plane acts as a constraint that separates this verified subset from the complimentary subset of the input domain that requires further analysis, i.e., $\{\boldsymbol{x} \mid \underline{\boldsymbol{a}}^{(L)\top}\boldsymbol{x} + \underline{c}^{(L)} \leq 0\}$. For verification problems with multiple output conditions in the form of $f_1(\boldsymbol{x}) \geq 0 \wedge f_2(\boldsymbol{x}) \geq 0 \wedge \cdots \wedge f_T(\boldsymbol{x}) \geq 0$, we can have multiple linear bounds (1), one for each clause, and all of them can be used. Rather than treating each property separately, we jointly collect all relevant final-layer constraints and apply them using Algorithms 1 and 2, accelerating batch verification on a GPU.

**Modifications to the Standard Input BaB Procedure.** Algorithm 4 in Appendix C.3 outlines our modification to the standard input BaB loop, highlighted in brown. Our first key modification after bounding a batch of domains, is that we perform relaxed clipping **after** we split the domains. This **reordering** allows two child domains to inherit their parent's constraint after axis-aligned split. The inheritance enables a *more tailored domain clipping* procedure for each child, which is often more effective than clipping the parent domain directly. The resulting subdomains are reinserted into the domain list along with their associated constraints. In the subsequent bounding step, we perform complete clipping for the unstable neurons in each layer , where constraints are leveraged to *directly* to tighten the lower/upper bound objectives. When the number of unstable neurons is large, we heuristically select a subset of critical neurons for Complete Clipping; the selection heuristics are detailed in Appendix D.4. Neurons not selected for complete clipping still benefit indirectly, as their concretized bounds are refined due to the clipped domain inherited from the prior BaB iteration.

## 3.5 Integrating Clipping into Activation BaB Verification

We now demonstrate how our clipping framework naturally extends to branch-and-bound on the activation space. Our method is **activation-agnostic**, as it operates on general linear constraints derived from **any activation** split. For a given neuron $z_j^{(i)}$, BaB can introduce splits of the form $z_j^{(i)} \geq s$ or $z_j^{(i)} \leq s$, where $s$ is a split point ($s = 0$ for a ReLU split)[53]. This inequality can be relaxed by bound propagation algorithms and expressed as a linear constraint on the input, $\mathbf{g}^\top \mathbf{x} + h \leq 0$, which is directly usable by our algorithms. This flexibility allows our framework to apply to networks with various activation functions, as demonstrated by our experiments on Vision Transformer models in Section 4. By assigning certain unstable neurons to either regime, we obtain linear relaxation split constraints that can further optimize the *intermediate bounds*, boosting verification efficacy. A key insight is that **many constraints** from multiple activation assignments can accumulate, potentially tightening intermediate bounds significantly.

**Using Linear Constraints in Activation Space for Clipping.** We prioritize neurons whose convex envelopes incur the largest relaxation error; for ReLU this typically coincides with "unstable" units (Appendix A.1). At any BaB node with input box $\mathcal{X}_t$, linear bound propagation yields, for each neuron $(i, j)$,

$$\underline{\boldsymbol{A}}_j^{(i)\top} \boldsymbol{x} + \underline{\boldsymbol{c}}_j^{(i)} \leq z_j^{(i)}(\boldsymbol{x}) \leq \overline{\boldsymbol{A}}_j^{(i)\top} \boldsymbol{x} + \overline{\boldsymbol{c}}_j^{(i)} \quad \forall \boldsymbol{x} \in \mathcal{X}_t. \tag{6}$$

Then we can **validate** of activation-space linear constraints. If a branch assigns $z_j^{(i)} \geq s$, then any feasible $\boldsymbol{x}$ must satisfy

$$\overline{\boldsymbol{A}}_j^{(i)\top} \boldsymbol{x} + \overline{\boldsymbol{c}}_j^{(i)} \geq s, \tag{7}$$

since $z_j^{(i)}(\boldsymbol{x}) \leq \overline{\boldsymbol{A}}_j^{(i)\top} \boldsymbol{x} + \overline{\boldsymbol{c}}_j^{(i)}$. Symmetrically, for $z_j^{(i)} \leq s$ we obtain the necessary condition $\underline{\boldsymbol{A}}_j^{(i)\top} \boldsymbol{x} + \underline{\boldsymbol{c}}_j^{(i)} \leq s$. Thus the activation-branching provide *sound linear constraints in the input space* that encode the assigned activation regime.

We can cache and **reuse** the initialized activation branching linear constraints. When bound initialization is executed at a node, we cache $(\underline{\boldsymbol{A}}, \underline{\boldsymbol{c}})$ and $(\overline{\boldsymbol{A}}, \overline{\boldsymbol{c}})$ for all "unstable" neurons, and further use the constraints based on the branching domains during BaB to do (i) *Relaxed Clipping*: treat (7) (and its lower-bound analogue) as box-consistency constraints and update $\mathcal{X}_t \mapsto \mathcal{X}_t'$ in closed form following Algorithm. 2). (ii) *Complete Clipping*: on a selected set $\mathcal{C}^{(p)}$ of neurons, we directly tighten their per-neuron affine bounds using the same constraints using Algorithm. 1. Neurons not in $\mathcal{C}^{(p)}$ keep their original bounds, but still tighten indirectly when re-concretized over $\mathcal{X}_t'$.

Here, $\mathcal{C}^{(p)}$ is the set of "critical neurons" that are heuristically expected to benefit most from bound refinement. While one could apply our method to every neuron in the network, this may be challenging for networks with high-dimensional hidden layers. To address this, we propose a top-k objective selection heuristic that adaptively selects neurons based on the BaBSR intercept score [13]. This heuristic strategically prioritizes neurons whose refinement is most likely to tighten the overall verification bounds. A detailed justification for this choice, including an ablation study comparing BaBSR to other common heuristics, is provided in Appendix D.4. Neurons not selected by this heuristic retain their standard bounds as computed by CROWN but may still benefit *indirectly* from relaxed clipping. Since relaxed clipping is computationally lightweight and highly scalable, it should always be applied to refine the input domain as effectively as possible.

Table 1: Comparison of different toolkits on a few representative VNN-COMP benchmarks with input BaB. "-" indicates that the benchmark was not supported. Time is calculated as the total time taken to verify verified instances. The number of BaB subproblems by $\alpha, \beta$-CROWN is set as the baseline, and the reduction rate is calculated from this number. Complete clipping significantly reduces the number of subproblems during BaB, while relaxed clipping is sometimes faster overall. Reordering generally helps reduce time and subproblems.

| | lsnc | | | acasxu | | | nn4sys | | |
|---|---|---|---|---|---|---|---|---|---|
| Method | time(s) | subproblems | # verified | time(s) | subproblems | # verified | time(s) | subproblems | # verified |
| nnenum* [5, 7] | - | - | - | 213.41 | - | **139** | 167.55 | - | 22 |
| Marabou†‡ [36, 71] | - | - | - | 1342.03 | - | 134 | 151.31 | - | 24 |
| PyRAT‡ [30] | 90.40 | - | 15 | 1484.39 | - | 137 | 704.55 | - | 53 |
| Never2‡ [30] | - | - | - | 1368.78 | - | 121 | - | - | - |
| NNV‡[62] | - | - | - | 2631.49 | - | 70 | - | - | - |
| Cora‡[1] | - | - | - | 1566.80 | - | 134 | 22.88 | - | 2 |
| NeuralSAT‡[22] | - | - | - | 1316.85 | - | 138 | - | - | - |
| $\alpha, \beta$-CROWN‡ | 115.27 | 142,293,985 | **40** | 280.51 | 7,154,387 | 138 | 1580.66 | 4,440,252 | **194** |
| Clip-and-Verify (Ours) | | | | | | | | | |
| Relaxed clipping | 99.12 | $92,402,227^{\downarrow 35.1\%}$ | **40** | 151.37 | $3,124,100^{\downarrow 56.3\%}$ | **139** | 1193.89 | $2,691,750^{\downarrow 39.4\%}$ | **194** |
| Relaxed + Reorder | 98.42 | $66,412,652^{\downarrow 53.3\%}$ | **40** | **150.25** | $2,557,715^{\downarrow 64.2\%}$ | **139** | **1166.08** | $2,300,894^{\downarrow 48.2\%}$ | **194** |
| Complete clipping | **84.30** | $\mathbf{5,334,421}^{\downarrow 96.3\%}$ | **40** | 168.57 | $\mathbf{1,533,068}^{\downarrow 78.6\%}$ | **139** | 2846.06 | $2,141,288^{\downarrow 51.8\%}$ | **194** |

Table 2: Performance of Clip-and-Verify on challenging control system verification tasks. Complete clipping is essential for verifying the most difficult properties. The timeout is 3 days (259,200s).

| | cartpole | | Quadrotor-2D | | Quad-2D-Large | |
|---|---|---|---|---|---|---|
| Method | time(s) | subproblems | time(s) | subproblems | time(s) | subproblems |
| $\alpha, \beta$-CROWN (No clipping) | 1602 | 54,260,909 | timeout | | timeout | |
| Clip-and-Verify (Relaxed clipping) | 484 | $16,453,971^{\downarrow 69.7\%}$ | 209,504 | 2,630,043,050 | timeout | |
| Clip-and-Verify (Complete clipping) | **142** | $\mathbf{2,438,359}^{\downarrow 95.5\%}$ | **78,818** | $\mathbf{1,112,917,436}^{\downarrow 57.7\%}$ | **104,614** | **1,472,433,971** |

(a) acasxu

(b) lsnc

(c) cifar-cnn-b-adv

(d) cifar10-resnet

(e) cifar100

(f) tinyimagenet

Figure 2: Representative benchmarks visualization. (a) and (b) are input BaB benchmarks with timeout 120s and 100s respectively, (c) and (d) represent medium-size ReLU nets, whereas (e) and (f) are substantially larger. These latter 4 benchmarks utilize ReLU splitting, for which we use complete clipping. Despite their differences in scale, our Clip-and-Verify algorithm demonstrates strong performance on both. Please refer to appendix D.2 for a detailed interpretation.

**Modifications to the Standard Activation BaB Procedure.** Our activation space BaB Algorithm 5, shown in Appendix C.4, iteratively uses activation split constraints to optimize the intermediate bounds via Complete Clipping (Theorem 3.1) . This strategy fully exploits accumulated constraints through complete clipping on intermediate bounds directly, bypassing costly LP solves.

## 4 Experiments

We evaluate the effectiveness of Clip-and-Verify on several benchmarks from VNN-COMP 2021-2024 [6, 49, 11, 10]. We first demonstrate its benefits on three benchmarks and three hard NN control system problems that are commonly solved with the input BaB procedure, then evaluate our approach on challenging activation space BaB benchmarks in VNN-COMPs, as well as SDP-FO benchmarks

Table 3: Comparison of different toolkits and Clip-and-Verify on VNN-COMP benchmarks with activation split BaB. Results of $\beta$-CROWN and BICCOS were from the same hardware of our experiments for a direct comparison; other results are from VNN-COMP reports. "-" indicates the benchmark was not supported.

| | oval22 time(s) | oval22 # verified | cifar10-resnet time(s) | cifar10-resnet # verified | cifar100-2024 time(s) | cifar100-2024 # verified | tinyimagenet-2024 time(s) | tinyimagenet-2024 # verified |
|---|---|---|---|---|---|---|---|---|
| nnenum* [5, 7] | 630.06 | 3 | - | - | - | - | - | - |
| ERAN*[47, 46] | 233.84 | 6 | 24,74 | 43 | - | - | - | - |
| OVAL* [21, 20] | 393.14 | 11 | - | - | - | - | - | - |
| Venus2 [9, 39] | 386.71 | 17 | - | - | - | - | - | - |
| VeriNet† [32, 33] | 73.65 | 17 | 8.11 | 48 | - | - | - | - |
| MN-BaB† [27] | 137.13 | 19 | - | - | - | - | - | - |
| Marabou†‡ [36, 71] | 5.33 | 19 | 40.42 | 39 | - | 0 | - | 0 |
| PyRAT‡ [30] | - | - | - | - | 42.38 | 68 | 55.64 | 49 |
| $\beta$-CROWN [77, 66] | 29.39 | 20 | 9.17 | 60 | 8.15 | 119 | 8.65 | 135 |
| BICCOS [76, 79] | 31.72 | 25 | 16.73 | 63 | 8.75 | 121 | 9.73 | 138 |
| Clip-and-Verify with $\beta$-CROWN | 28.15 | 22 | 6.06 | 63 | 6.59 | 126 | 9.48 | 140 |
| Clip-and-Verify with BICCOS | 46.48 | **27** | 11.81 | **64** | 8.17 | **131** | 10.48 | **144** |
| Upper Bound | | 29 | | 72 | | 168 | | 157 |

* Results from VNN-COMP 2021 report [6].    † from VNN-COMP 2022 report [49]    ‡ from VNN-COMP 2024 report [10]

Table 4: Verified accuracy (Var.%), avg. per-example verification time (s) on VNN-COMP benchmarks and other commonly used benchmarks. The average time is calculated on verified images only. Clip-and-Verify consistently outperforms all baselines when combined with state-of-the-art BaB verifiers such as BICCOS [79]

| Dataset | Model | $\beta$-CROWN Ver.% | $\beta$-CROWN Time (s) | GCP-CROWN with MIP cuts Ver.% | GCP-CROWN with MIP cuts Time(s) | BICCOS Ver.% | BICCOS Time(s) | Clip-and-Verify with $\beta$-CROWN Ver.% | Clip-and-Verify with $\beta$-CROWN Time(s) | Clip-and-Verify with MIP cuts Ver.% | Clip-and-Verify with MIP cuts Time(s) | Clip-and-Verify with BICCOS Ver.% | Clip-and-Verify with BICCOS Time(s) | Upper bound |
|---|---|---|---|---|---|---|---|---|---|---|---|---|---|---|
| $\epsilon = 0.3$ and $\epsilon = 2/255$ | | | | | | | | | | | | | | |
| MNIST | CNN-A-Adv | 71.0 | 4.28 | 71.5 | 6.64 | 76.0 | 6.91 | 74.0 | 3.33 | 73.5 | 5.15 | **76** | 6.39 | 76.5 |
| CIFAR | CNN-A-Adv | 45.5 | 5.26 | **48.5** | 4.81 | **48.5** | 3.91 | 45.5 | 1.35 | **48.5** | 3.59 | **48.5** | 3.08 | 50.0 |
| | CNN-A-Adv-4 | 46.5 | 1.25 | **48.5** | 2.64 | **48.5** | 1.55 | 46.5 | 0.62 | **48.5** | 2.26 | **48.5** | 1.31 | 49.5 |
| | CNN-A-Mix | 42.0 | 4.00 | 47.5 | 10.49 | **48.0** | 9.00 | 43.0 | 4.24 | 47.5 | 7.45 | **48.0** | 7.45 | 53.0 |
| | CNN-A-Mix-4 | 51.0 | 1.05 | 55.0 | 5.94 | 56.0 | 9.22 | 51.0 | 0.84 | 55.0 | 4.44 | **56.5** | 5.12 | 57.5 |
| | CNN-B-Adv | 47.0 | 8.25 | 49.5 | 12.16 | 51.0 | 10.79 | 49 | 7.54 | **51.5** | 11.68 | **51.5** | 7.68 | 65.0 |
| | CNN-B-Adv-4 | 55.0 | 2.91 | 58.5 | 7.01 | 59.5 | 4.7 | 56.5 | 0.82 | 60.0 | 6.67 | **60.5** | 5.55 | 63.5 |
| cifar10-resnet | | 83.33 | 9.17 | 87.5 | 17.99 | 87.5 | 16.73 | 86.11 | 6.06 | **88.89** | 16.73 | **88.89** | 11.80 | 100.0 |
| oval22 | | 66.66 | 29.39 | 83.33 | 63.53 | 83.33 | 31.72 | 73.33 | 28.15 | **90.00** | 45.52 | **90.00** | 46.48 | 96.67 |
| cifar100-2024 | | 59.5 | 8.15 | - | - | 60.5 | 8.75 | 63.0 | 6.59 | - | - | **65.5** | 8.17 | 84.0 |
| tinyimagenet-2024 | | 67.5 | 8.65 | - | - | 69.0 | 9.73 | 70.0 | 9.48 | - | - | **72.0** | 10.48 | 78.5 |
| vision-transformer 2024 [53] | | 59.0 | 22.04 | - | - | - | - | **61.0** | 10.81 | - | - | - | - | 100.0 |

introduced in previous studies [19, 66]. Fig. 2 shows the overview of results of selected benchmarks. All results visualization and details about the configuration see Appendix D.1.

**Verification Results on Input Split Benchmarks and Hard NN Control Systems.** For evaluation, we focus on the following benchmarks from VNN-COMP 23 and VNN-COMP 24: `acasxu`, `lsnc`, and `nn4sys`. we have considered all input benchmarks and these three are the mostly challenging ones when solved using input BaB. We compare our approach to other baseline tools as shown in Table 1. Overall, Clip-and-Verify reduces the number of branches by over 50% and accelerating the verification process. For `lsnc`, we reduced the number of domains by 96%. We further tested our method on three **challenging** verification tasks from a recent study on **provably stable neural network control systems** [75, 41, 42]: `cartpole`, `Quadrotor-2D`, and `Quadrotor-2D-Larger-ROA`. These tasks require certifying Lyapunov-based stability over high-volume state domains, and can not be solved by existing verifiers. The challenge of the verification problem is that we need to verify inside the intersection of a large box and a level set of its Lyapunov function. Detailed settings see Appendix D.1. These instances therefore test whether a verifier can reason about Lyapunov decrease and boundary non-escape over large, physically meaningful state ranges, rather than over small adversarial balls. While general-purpose robustness verifiers are not designed for such volumes, our method targets exactly this regime. As shown in Table 2, baseline methods without clipping fail to solve these problems within the time limit. Both relaxed and complete clipping enable verification, with **complete clipping** demonstrating superior performance by drastically reducing both the number of visited BaB subproblems and the total runtime, turning previously intractable problems into verifiable ones.

**Verification Results on General Activation Split Benchmarks** Shown in Table 3, we further compare the proposed methods against a wide range of existing neural network verification toolkits on six challenging VNN-COMP benchmarks: `oval22`, `cifar10-resnet`, `cifar100-2024`, `tinyimagenet-2024`, and `vit-2024` Each method is evaluated in terms of average runtime (seconds) and the number of verified properties (# verified). A dash (-) denotes that the corresponding tool

was not applicable or did not support that particular benchmark. Clip-and-Verify with BICCOS attains state-of-the-art verification coverage on the evaluated VNN-COMP benchmarks, pushing closer to the theoretical upper bound in terms of the number of properties verified. When combined with BICCOS, whose tight bounding routines complement Clip-and-Verify's efficient splitting, coverage reaches to 131 and 144, underscoring the method's broad applicability and scalability. Besides of the ReLU nets, we further evaluated our method on a **Vision Transformer** (ViT) model [53], which contains **non-ReLU** layers such as **Softmax**. Verifying such architectures is challenging due to the complex, non-linear constraints introduced by these operators. The model tested has approximately 76k parameters and an input dimension of 3072. Tables 4, 6 show that Clip-and-Verify significantly outperforms the baseline, verifying more properties in about half the time and with nearly 45% fewer subproblems in ViT. This demonstrates the applicability of our framework on general networks.

**Ablation Studies.** We compare the BaB baseline without clipping, *Relaxed Clipping* only, and the full pipeline (Relaxed + Complete) in Table 1. We further vary the comparison of complete clipping and LP solvers in Section D.3, and *scoring rule* for choosing critical neurons (e.g., BaBSR intercept, envelope gap, bound width) in Complete Clipping in Section D.4. For each configuration we report time and visited subproblems under identical branching/timeout settings in Table 4 and Table 6; detailed breakdowns, per-model trends, and ablation protocols are in Appendix D.2.

## 5    Related Work

Verification becomes easier when we tighten the *base relaxations* that bound intermediate layers and *contract the input region*, so that a subsequent branch-and-bound (BaB) search explores fewer and easier subproblems. BaB frameworks unify many complete verifiers and show that stronger node bounds reduce tree size [14, 13, 55, 64, 77, 5]. Subsequent improvements include faster dual/active-set solvers and better branching [20, 66, 27]. Our clipping steps make each node *strictly easier* (smaller box and tighter intermediate relaxations) without spawning additional subproblems, and are therefore complementary to any branching heuristic. Our method contracts the box domain using linear constraints and injects the same constraints to tighten intermediate affine bounds, thus improving the node-wise base problem before any additional branching is introduced.

For tighter base relaxations, linear-relaxation-based bound propagation (e.g., CROWN [77] and DeepPoly [56]) provide fast base bounds. Beyond this, early single-neuron triangle relaxations such as PLANET [25] and the convex adversarial polytope/Wong–Kolter line [69] underpin modern propagation methods; symbolic/abstract-interpretation variants (DeepZ [55]/AI2 [29]/Neurify [64]/OSIP [31]) further improve hidden-layer bounds in practice. Multi-neuron relaxations overcome single-neuron limits by coupling activations [54, 2, 48, 57]. Beyond k-ReLU and PRIMA [54, 48], tightened layer-wise convex relaxations [59] and barrier-oriented analyses (e.g., for simplex inputs or active-set scaling) show how to systematically overcome single-neuron limits. Beyond pure LP, SDP-based relaxations with linear cuts capture neuron coupling when affordable [8, 40, 44, 18]. Orthogonal to envelopes, NN-specific cutting planes have been injected directly into bound propagation e.g., general cutting planes (GCP-CROWN [76]) and conflict-driven cuts (BICCOS [79]) to strengthen node-wise relaxations before or alongside BaB. Exact formulations (MILP/SMT) and their strengthened variants remain an important baseline and hybrid component [60, 2, 35, 36], though scalability constraints often shift emphasis toward stronger relaxations and BaB. Recent work also tightens bounds via optimization-driven rolling-horizon decomposition, effectively performing layerwise optimization-based bound tightening to improve intermediate bounds [78]. Our *Complete Clipping* is orthogonal: instead of inventing new global envelopes, we reuse **existing** linear constraints produced during verification to tighten per-neuron affine bounds inside the same bound-propagation framework. On the other hand, Contract-Simple [7] contracts a zonotope using one output inequality. We generalize this in a way tailored to bound propagation: *Relaxed Clipping* applies *arbitrary* linear constraints from final-layer bounds or activation branchings to a box proxy with closed-form updates, avoiding repeated LP solves while yielding globally useful domain reductions reused across all neurons.

## 6    Conclusion

We proposed **Clip-and-Verify**, a framework that enhances the Branch-and-Bound paradigm in neural network verification through our novel *linear constraint-driven domain clipping* algorithms. An efficient and light-weight paradigm for tightening intermediate-layer bounds has been a remaining gap in neural network verification that has finally been addressed by our framework. Limitation and broader impacts discussed in Appendix E.

## Acknowledgements

Huan Zhang is supported in part by the AI2050 program at Schmidt Sciences (AI2050 Early Career Fellowship) and NSF (IIS-2331967). Grani A. Hanasusanto is supported in part by NSF (CCF-2343869 and ECCS-2404413). We thank the anonymous reviewers for their constructive feedback. In particular, we acknowledge Reviewer vsjp for their invaluable insights on the theoretical foundations of our work, particularly for identifying the connection between our dual optimization approach and the continuous knapsack problem.

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

# Appendix

# Contents

## A  Formulations

### A.1  Unstable Neurons and Complexity

**Unstable neurons for General Activations**   Let $z_j^{(i)}$ be the pre-activation of neuron $(i, j)$ in layer $i$ number $j$ with interval bounds $[\underline{z}_j^{(i)}, \overline{z}_j^{(i)}]$, and let the component-wise activation be $\phi^{(i)}(\cdot)$. Given a sound linear relaxation (convex envelope) for the scalar graph of $\phi^{(i)}$ over $[\underline{z}_j^{(i)}, \overline{z}_j^{(i)}]$, denote by $l_j^{(i)}(z)$ and $u_j^{(i)}(z)$ its lower and upper affine bounds, and define the envelope gap (or relaxation error):

$$\delta_j^{(i)} \triangleq \max_{z \in [\underline{z}_j^{(i)}, \overline{z}_j^{(i)}]} \left( u_j^{(i)}(z) - l_j^{(i)}(z) \right). \tag{8}$$

We call neuron $(i, j)$ *stable* if $\delta_j^{(i)} = 0$ (the relaxation is exact on the interval), and *unstable* if $\delta_j^{(i)} > 0$. This definition is activation-agnostic: for piecewise-linear (PWL) activations, stability is equivalent to $[\underline{z}_j^{(i)}, \overline{z}_j^{(i)}]$ lying inside a single linear piece (e.g., for ReLU it reduces to $\overline{z}_j^{(i)} \leq 0$ or $\underline{z}_j^{(i)} \geq 0$); for smooth nonlinear activations one typically adopts a PWL relaxation, in which case instability means the interval spans at least two segments (hence a nonzero envelope gap).

**Complexity.**   Let $U$ be the number of unstable neurons at a BaB node. Once every unstable neuron has its activation region (piece) fixed, the network reduces to an affine map over the input box, and the remaining subproblem is convex. For PWL activations where each unstable neuron has at most

$K$ admissible pieces,[1] the number of global activation patterns is bounded by $K^U$. Consequently, any exact verifier that distinguishes activation pieces (e.g., MILP/SMT encodings with one discrete choice per unstable neuron, or BaB that branches on activation pieces) faces, in the worst case,

$$\# \text{ subproblems} \in \Omega\big(K^U\big) \quad \text{and hence} \quad \text{time} = \Omega\big(T_{\text{bound}} \cdot K^U\big),$$

where $T_{\text{bound}}$ is the per-node cost to evaluate/tighten bounds. In particular, for ReLU networks $K = 2$, yielding the familiar exponential scaling $\Omega(2^U)$; thus any mechanism that reduces $U$ (e.g., by stabilizing neurons via tighter pre-activation intervals) produces a multiplicative reduction in worst-case search, by a factor of $K^{\Delta U}$ when $U$ decreases by $\Delta U$.

## A.2 MIP Formulation and LP & Planet Relaxation

Early approaches pursue exact guarantees for verification through Mixed-Integer Linear Programming (MILP) [17, 43, 23, 28, 61, 73] and Satisfiability Modulo Theories (SMT) [52, 35, 15, 25].

**The MIP Formulation** The mixed integer programming (MIP) formulation is the root of many NN verification algorithms. Given the RELU activation function's piecewise linearity, the model requires binary encoding variables, or ReLU indicators $\delta$ only for unstable neurons. We formulate the optimization problem aiming to minimize the function $f(\mathbf{x})$, subject to a set of constraints that encapsulate the DNN's architecture and the perturbation limits around a given input $\mathbf{x}$, as follows:

$$f^\star = \min_{\mathbf{x}, \hat{z}, \delta} f(\mathbf{x}) \qquad \text{s.t. } f(\mathbf{x}) = z^{(L)}; \hat{z}^{(0)} = \mathbf{x} \in \mathcal{X} \tag{9a}$$

$$\hat{\mathbf{z}}^{(i)} = \mathbf{W}^{(i)}\hat{\mathbf{z}}^{(i-1)} + \mathbf{b}^{(i)}; \quad i \in [L] \tag{9b}$$

$$\mathcal{I}^{+(i)} := \{j : l_j^{(i)} \geq 0\} \tag{9c}$$

$$\mathcal{I}^{-(i)} := \{j : u_j^{(i)} \leq 0\} \tag{9d}$$

$$\mathcal{I}^{(i)} := \{j : l_j^{(i)} < 0, u_j^{(i)} > 0\} \tag{9e}$$

$$\mathcal{I}^{+(i)} \cup \mathcal{I}^{-(i)} \cup \mathcal{I}^{(i)} = \mathcal{J}^i \tag{9f}$$

$$\hat{x}_j^{(i)} \geq 0; j \in \mathcal{I}^{(i)}, i \in [L-1] \tag{9g}$$

$$\hat{z}_j^{(i)} \geq z_j^{(i)}; j \in \mathcal{I}^{(i)}, i \in [L-1] \tag{9h}$$

$$\hat{z}_j^{(j)} \leq u_j^{(i)}\delta_j^{(i)}; j \in \mathcal{I}^{(i)}, i \in [L-1] \tag{9i}$$

$$\hat{z}_j^{(i)} \leq z_j^{(i)} - l_j^{(i)}(1 - \delta_j^{(i)}); j \in \mathcal{I}^{(i)}, i \in [L-1] \tag{9j}$$

$$\delta_j^{(i)} \in \{0, 1\}; j \in \mathcal{I}^{(i)}, i \in [L-1] \tag{9k}$$

$$\hat{z}_j^{(i)} = z_j^{(i)}; j \in \mathcal{I}^{+(i)}, i \in [L-1] \tag{9l}$$

$$\hat{z}_j^{(i)} = 0; j \in \mathcal{I}^{-(i)}, i \in [L-1] \tag{9m}$$

To initialize intermediate bounds for each neuron, we replace the original objective $f(\mathbf{x})$ with the neuron's pre-activation value $z_j^{(i)}$. This lets us solve the following bounds for every neuron $j$ in layer $i$, with $i \in [L-1]$ and $j \in \mathcal{J}^{(i)}$:

$$l_j^{(i)} = \min_{x \in \mathcal{X}} f_j^{(i)}(\mathbf{x}), \quad u_j^{(i)} = \max_{x \in \mathcal{X}} f_j^{(i)}(\mathbf{x}).$$

Here, the set $\mathcal{J}^{(i)}$ comprises all neurons in layer $i$, which can be categorized into three groups: 'active' ($\mathcal{I}^{+(i)}$), 'inactive' ($\mathcal{I}^{-(i)}$), and 'unstable' ($\mathcal{I}^{(i)}$).

Next, the MIP formulation is initialized with the constraints

$$l_j^{(i)} \leq z_j^{(i)} \leq u_j^{(i)},$$

---

[1]For ReLU, $K = 2$ (inactive/active). For HardTanh or ReLU6, $K \leq 3$. For smooth activations under an $S$-segment relaxation, take $K = S$.

across all neurons and layers $i$. These bounds can be computed recursively, propagating from the first layer up to the $i$-th layer. However, since MIP problems involve integer variables, they are generally NP-hard, reflecting the computational challenge of this approach.

**The LP and Planet relaxation.** By relaxing the binary variables in (9k) to $\delta_j^{(i)} \in [0, 1], j \in \mathcal{I}^{(i)}, i \in [L-1]$, we can get the LP relaxation formulation. By replacing the constraints in (9i), (9j), (9k) with

$$\hat{z}_j^{(i)} \le \frac{u_j^{(i)}}{u_j^{(i)} - l_j^{(i)}}(z_j^{(i)} - l_j^{(i)}); \quad j \in \mathcal{I}^{(i)}, i \in [L-1] \tag{10}$$

we can eliminate the $\delta$ variables and get the well-known Planet relaxation formulation [25]. Both of these two relaxations are solvable in polynomial time to yield lower bounds.

## A.3 Formulating Bounding Hyperplanes in Linear Bound Propagation

In a feedforward network, $\underline{\boldsymbol{A}}^{(i)}, \overline{\boldsymbol{A}}^{(i)}, \underline{\boldsymbol{c}}^{(i)}$ and $\overline{\boldsymbol{c}}^{(i)}$ must be derived for every linear layer preceding an activation layer, as well as the final layer of the network. In order to derive the hyperplane coefficients $(\underline{\boldsymbol{A}}^{(i)}/\overline{\boldsymbol{A}}^{(i)})$ and biases $(\underline{\boldsymbol{c}}^{(i)}/\overline{\boldsymbol{c}}^{(i)})$, at this $i^{th}$ layer, all preceding activation layers must have already had their inputs bounded. The following lemma describes how a ReLU activation layer may be relaxed which will be useful for defining bounding hyperplanes, $\underline{\boldsymbol{A}}^{(i)}, \overline{\boldsymbol{A}}^{(i)}, \underline{\boldsymbol{c}}^{(i)}$ and $\overline{\boldsymbol{c}}^{(i)}$.

**Lemma A.1.** *(Relaxation of a ReLU layer in CROWN). Given the lower and upper bounds of $z_j^{(i-1)}$, denoted as $l_j^{(i-1)}$ and $u_j^{(i-1)}$, respectively, the linear layer proceeding the ReLU activation layer may be lower-bounded element-wise by the following inequality:*

$$\boldsymbol{z}^{(i)} = \boldsymbol{W}^{(i)} \sigma(\boldsymbol{z}^{(i-1)}) \ge \boldsymbol{W}^{(i)} \boldsymbol{D}^{(i-1)} \boldsymbol{z}^{(i-1)} + \boldsymbol{W}^{(i)} \underline{\boldsymbol{b}}^{(i-1)} \tag{11}$$

*where $\boldsymbol{D}^{(i-1)}$ is a diagonal matrix with shape $\mathbb{R}^{n_{i-1} \times n_{i-1}}$ whose off-diagonal entries are 0, and on-diagonal entries are defined as:*

$$\boldsymbol{D}_{j,j}^{(i-1)} := \begin{cases} 1, & l_j^{(i-1)} \ge 0 \\ 0, & u_j^{(i-1)} \le 0 \\ \boldsymbol{\alpha}_j^{(i-1)}, & l_j^{(i-1)} < 0 < u_j^{(i-1)} \text{ and } \boldsymbol{W}_j^{(i)} \ge 0 \\ \frac{\boldsymbol{u}_j^{(i-1)}}{\boldsymbol{u}_j^{(i-1)} - \boldsymbol{l}_j^{(i-1)}}, & l_j^{(i-1)} < 0 < u_j^{(i-1)} \text{ and } \boldsymbol{W}_j^{(i)} < 0 \end{cases} \tag{12}$$

*and $\underline{\boldsymbol{b}}_j^{(i-1)}$ is a vector with shape $\mathbb{R}^{n_{i-1}}$ whose elements are defined as:*

$$\underline{\boldsymbol{b}}_j^{(i-1)} := \begin{cases} 0, & l_j^{(i-1)} > 0 \text{ or } u_j^{(i-1)} \le 0 \\ 0, & l_j^{(i-1)} < 0 < u_j^{(i-1)} \text{ and } \boldsymbol{W}_j^{(i)} \ge 0 \\ -\frac{\boldsymbol{u}_j^{(i-1)} \boldsymbol{l}_j^{(i-1)}}{\boldsymbol{u}_j^{(i-1)} - \boldsymbol{l}_j^{(i-1)}}, & l_j^{(i-1)} < 0 < u_j^{(i-1)} \text{ and } \boldsymbol{W}_j^{(i)} < 0 \end{cases} \tag{13}$$

*In the above definitions, $\boldsymbol{\alpha}_j^{(i-1)}$ is a parameter in range $[0, 1]$ and may be fixed or optimized as in [74].*

*Proof.* For the $j^{\text{th}}$ ReLU at the $(i-1)^{\text{th}}$ layer, it's result may be bounded as follows:

$$\boldsymbol{\alpha}_j^{(i-1)} \boldsymbol{z}_j^{(i-1)} \le \sigma(\boldsymbol{z}_j^{(i-1)}) \le \frac{\boldsymbol{u}_j^{(i-1)}}{\boldsymbol{u}_j^{(i-1)} - \boldsymbol{l}_j^{(i-1)}}(\boldsymbol{z}_j^{(i-1)} - \boldsymbol{l}_j^{(i-1)}). \tag{14}$$

The right-hand side holds as this is the Planet-relaxation defined by equation (10). For the left-hand side, we first consider when $\boldsymbol{z}_j^{(i-1)} \le 0$. For every input in this range, the result of the ReLU is $\sigma(\boldsymbol{z}_j^{(i-1)}) = 0$. $\boldsymbol{\alpha}_j^{(i-1)} \boldsymbol{z}_j^{(i)}$ forms a line for which inputs in this range will always produce a

non-positive result when $\boldsymbol{\alpha}_j^{(i-1)} \in [0, 1]$. For inputs in the range $\boldsymbol{z}_j^{(i-1)} \geq 0$, the result of the ReLU is $\sigma(\boldsymbol{z}_j^{(i-1)}) = \boldsymbol{z}_j^{(i-1)}$. This result is never exceeded by $\boldsymbol{\alpha}_j^{(i-1)} \boldsymbol{z}_j^{(i-1)}$ when $\boldsymbol{\alpha}_j^{(i-1)} \in [0, 1]$.

When the result, $\sigma(\boldsymbol{z}_j^{(i-1)})$, is multiplied by a scalar such as $\boldsymbol{W}_j^{(i)}$, a valid lower-bound of $\boldsymbol{W}_j^{(i)} \sigma(\boldsymbol{z}_j^{(i-1)})$ requires a lower bound on $\sigma(\boldsymbol{z}_j^{(i-1)})$ when $\boldsymbol{W}_j^{(i)} \geq 0$, and an upper bound on $\sigma(\boldsymbol{z}_j^{(i-1)})$ when $\boldsymbol{W}_j^{(i)} < 0$. Such lower and upper bounds are indeed produced by $\boldsymbol{D}_{j,j}^{(i-1)}$ and $\underline{\boldsymbol{b}}_j^{(i-1)}$, whose definitions are derived from the inequality displayed in equation (14). This concludes the proof. $\qquad\square$

Lemma A.1 suggests a recursive approach to bounding a neural network as the bounds at the $i^{\text{th}}$ layer depends on the bounds of the layer preceding it due to the dependence on $\boldsymbol{l}_j^{(i-1)}$ and $\boldsymbol{u}_j^{(i-1)}$. This is indeed the case, and we may define our hyperplane coefficients as $\underline{\boldsymbol{A}}^{(i)} = \boldsymbol{\Omega}^{(i,1)} \boldsymbol{W}^{(1)}$ where

$$\boldsymbol{\Omega}^{(i,k)} := \begin{cases} \boldsymbol{W}^{(i)} \boldsymbol{D}^{(i-1)} \boldsymbol{\Omega}^{(i-1)}, & i > k \\ \boldsymbol{I}, & i = k \end{cases} \tag{15}$$

To collect the remaining terms, we set $\underline{\boldsymbol{c}}^{(i)} = \sum_{k=2}^{i} \left( \boldsymbol{\Omega}^{(i,k)} \boldsymbol{W}^{(k)} \underline{\boldsymbol{b}}^{(k-1)} \right) + \sum_{k=1}^{i} \left( \boldsymbol{\Omega}^{(i,k)} \boldsymbol{b}^{(k)} \right)$. To obtain an upper bound, Lemma A.1 and its proof may be adjusted accordingly where appearances of the inequalities $\boldsymbol{W}^{(i)} \geq 0$ and $\boldsymbol{W}^{(i)} < 0$ are flipped. In doing so, we may repeat this recursive process in order to obtain $\overline{\boldsymbol{A}}^{(i)}$ and $\overline{\boldsymbol{c}}^{(i)}$.

Though we have described how a ReLU feedforward network may be bounded, appropriately updating the definitions of $\boldsymbol{D}^{(i)}$ and $\underline{\boldsymbol{b}}^{(i)}$ allows feedforward networks with general activation functions (that act element-wise) to be bounded. Such a general formulation is described in [77] that is similar to the template described above, and goes into further detail on how this formulation may be extended to *quadratic* bound propagation.

### A.4 Verification Properties Given as Boolean Logical Formulae

$$f(\boldsymbol{x}) \geq 0, \qquad \forall \boldsymbol{x} \in \mathcal{X} \tag{16}$$

Equation (16) is referred to as the *canonical* formulation [14, 13] as it is general enough to encompass any property involving boolean logical formulas over linear inequalities. To make this idea clear, consider a neural network whose output dimension is $n_L = 3$. We may wish to verify that the output corresponding to the true label, $\boldsymbol{y}_0$, is always greater than all other outputs, $\boldsymbol{y}_1$ and $\boldsymbol{y}_2$, for every input in $\mathcal{X}$. Explicitly, we want to verify:

$$(\boldsymbol{y}_0 > \boldsymbol{y}_1) \wedge (\boldsymbol{y}_0 > \boldsymbol{y}_2), \quad \forall \boldsymbol{x} \in \mathcal{X}. \tag{17}$$

To represent this formula, we may first introduce a new linear layer whose weight matrix is defined as $\boldsymbol{C}$, sometimes referred to as a *specification matrix*:

$$\boldsymbol{C} := \begin{bmatrix} 1 & -1 & 0 \\ 1 & 0 & -1 \end{bmatrix}. \tag{18}$$

This specification matrix allows the output neurons to be compared against one another. Applying this matrix to the output vector of the network, denoted as $\boldsymbol{y} := [\boldsymbol{y}_0, \boldsymbol{y}_1, \boldsymbol{y}_2]^\top$, results in the following new output:

$$\begin{bmatrix} \boldsymbol{y}_0 - \boldsymbol{y}_1 \\ \boldsymbol{y}_0 - \boldsymbol{y}_2 \end{bmatrix} = \boldsymbol{C}\boldsymbol{y}. \tag{19}$$

Next, we append a *MinPool* layer to the network which returns the minimum of these two results. Now, we may define a new neural network, $f'(\boldsymbol{x})$, which appends the aforementioned *MinPool* and linear layer to $f(\boldsymbol{x})$, resulting in the final output, $\min\{\boldsymbol{y}_0 - \boldsymbol{y}_1, \boldsymbol{y}_0 - \boldsymbol{y}_2\}$. It should now be apparent that if the property described by equation (17) can be verified as true, then the following must hold:

$$\min_{x \in \mathcal{X}} f'(\boldsymbol{x}) > 0. \tag{20}$$

We may now proceed to produce a lower bound, $\underline{f'}(\boldsymbol{x})$, and only until $\underline{f'}(\boldsymbol{x}) > 0$, can we formally state that the network will correctly classify all inputs in $\mathcal{X}$. In the scenario that we instead want to verify the property $(\boldsymbol{y}_0 > \boldsymbol{y}_1 \vee \boldsymbol{y}_0 > \boldsymbol{y}_2)$ for all inputs in $\mathcal{X}$, we may simply redefine $f'(\boldsymbol{x})$ by replacing the *MinPool* layer with a *MaxPool* layer which would return $\max\{\boldsymbol{y}_0 - \boldsymbol{y}_1, \boldsymbol{y}_0 - \boldsymbol{y}_2\}$.

Finally, suppose we want to verify that the output corresponding to the true label is greater than all other outputs with some additional margin $m$ such that $(\boldsymbol{y}_0 > \boldsymbol{y}_1 + m \wedge \boldsymbol{y}_0 > \boldsymbol{y}_2 + m), \forall \boldsymbol{x} \in \mathcal{X}$ where $m > 0$ (i.e. the network should always classify the first label with additional relative confidence defined by $m$). Then the only modification is to also incorporate a bias vector, $\boldsymbol{t} := [-m, -m]^\top$, at the final linear layer where $C$ was defined. This bias vector is sometimes referred to as the *threshold*. This results in the final output, $f'(\boldsymbol{x}) = \min\{\boldsymbol{y}_0 - \boldsymbol{y}_1 - m, \boldsymbol{y}_0 - \boldsymbol{y}_2 - m\}$. Hence, equation (16) is general enough to encompass more complex queries.

# B   Proofs

## B.1   Statement and proof of Lemma B.1

We first introduce a preparatory result that establishes a closed-form solution for a linear optimization problem over a hyper-rectangular feasible set.

**Lemma B.1** (Dual Norm Concretization for Hyper-Rectangular Domains). *Let $\boldsymbol{\epsilon} \in \mathbb{R}^n$ with $\epsilon_i > 0$ for all $i \in [n]$, and define the hyper-rectangular domain:*

$$\mathcal{X} = \{\boldsymbol{x} \in \mathbb{R}^n : \hat{\boldsymbol{x}} - \boldsymbol{\epsilon} \le \boldsymbol{x} \le \hat{\boldsymbol{x}} + \boldsymbol{\epsilon}\}.$$

*For any vector $\boldsymbol{v} \in \mathbb{R}^n$, the following hold:*

$$\max_{\boldsymbol{x} \in \mathcal{X}} \boldsymbol{v}^\top \boldsymbol{x} = \boldsymbol{v}^\top \hat{\boldsymbol{x}} + |\boldsymbol{v}|^\top \boldsymbol{\epsilon},$$

$$\min_{\boldsymbol{x} \in \mathcal{X}} \boldsymbol{v}^\top \mathrm{x} = \boldsymbol{v}^\top \hat{\boldsymbol{x}} - |\boldsymbol{v}|^\top \boldsymbol{\epsilon},$$

*where $|\boldsymbol{v}| \in \mathbb{R}_+^n$ denotes the component-wise absolute value of $\boldsymbol{v}$.*

*Proof.* We begin by rewriting the feasible set as

$$\mathcal{X} = \{\boldsymbol{x} \in \mathbb{R}^n : \hat{\boldsymbol{x}} - \boldsymbol{\epsilon} \le \boldsymbol{x} \le \hat{\boldsymbol{x}} + \boldsymbol{\epsilon}\} = \{\hat{\boldsymbol{x}} + \boldsymbol{\epsilon} \circ \boldsymbol{x} : \boldsymbol{x} \in \mathbb{R}^n, \|\boldsymbol{x}\|_\infty \le 1\},$$

where $\circ$ denotes the Hadamard (component-wise) product. Then, the maximization problem becomes

$$\max_{\boldsymbol{x} \in \mathcal{X}} \boldsymbol{v}^\top \boldsymbol{x} = \max_{\|\boldsymbol{x}\|_\infty \le 1} \boldsymbol{v}^\top \hat{\boldsymbol{x}} + (\boldsymbol{v} \circ \boldsymbol{\epsilon})^\top \boldsymbol{x}$$

$$= \boldsymbol{v}^\top \hat{\boldsymbol{x}} + \sum_{i=1}^n |v_i|\epsilon_i = \boldsymbol{v}^\top \hat{\boldsymbol{x}} + |\boldsymbol{v}|^\top \boldsymbol{\epsilon},$$

where the final equality follows from the definition of the dual norm of the $\infty$-norm. The derivation for the minimization problem follows analogously by replacing the maximization with a minimization, which flips the sign of $|\boldsymbol{v}|^\top \boldsymbol{\epsilon}$. Thus, the claim follows. $\square$

## B.2   Proof of Theorem 3.1

*Proof.* The problem (2) is a linear program over a compact convex set $\mathcal{X}$ with an additional linear constraint. Let feasible set $\mathcal{F} = \{\boldsymbol{x} \in \mathcal{X} : \boldsymbol{g}^\top \boldsymbol{x} + h \le 0\}$ be the feasible set for (2).

Case 1: $\mathcal{F} = \emptyset$, then by definition, $L^\star = +\infty$. In this case, it implies that for all $\boldsymbol{x} \in \mathcal{X}, \boldsymbol{g}^\top \boldsymbol{x} + h > 0$.

Case 2: $\mathcal{F}$ is non-empty. Since $\mathcal{X}$ is compact and $\mathcal{F}$ is a closed non-empty subset of $\mathcal{X}$ (as it's the intersection of $\mathcal{X}$ with a closed half-space), $\mathcal{F}$ is also compact. The objective function $\boldsymbol{a}^\top \boldsymbol{x} + c$ is continuous. Therefore, $L^\star$ is finite and attained. We introduce the Lagrangian for problem (2) by partially dualizing the constraint $\boldsymbol{g}^\top \boldsymbol{x} + h \le 0$:

$$\mathcal{L}(\boldsymbol{x}, \beta) = (\boldsymbol{a}^\top \boldsymbol{x} + c) + \beta(\boldsymbol{g}^\top \boldsymbol{x} + h) \quad \text{for } \boldsymbol{x} \in \mathcal{X}, \beta \ge 0.$$

The primal problem can be written as:

$$L^\star = \min_{\boldsymbol{x} \in \mathcal{X}} \sup_{\beta \geq 0} \mathcal{L}(\boldsymbol{x}, \beta)$$

To swap the $\min$ and $\sup$, we can apply Sion's Minimax Theorem. Let $K = \mathcal{X}$ (a compact convex set in $\mathbb{R}^n$) and $M = \{\beta \in \mathbb{R} : \beta \geq 0\}$ (a closed convex set in $\mathbb{R}$). The function $\mathcal{L}(\boldsymbol{x}, \beta)$ has the following properties: 1) For any fixed $\beta \in M$, $\mathcal{L}(\boldsymbol{x}, \beta) = (\boldsymbol{a} + \beta\boldsymbol{g})^\top \boldsymbol{x} + (c + \beta h)$ is linear in $\boldsymbol{x}$, and thus convex and continuous on $K$. 2) For any fixed $\boldsymbol{x} \in K$, $\mathcal{L}(\boldsymbol{x}, \beta) = (\boldsymbol{g}^\top \boldsymbol{x} + h)\beta + (\boldsymbol{a}^\top \boldsymbol{x} + c)$ is linear in $\beta$, and thus concave and continuous on $M$. Since these conditions are met, Sion's Minimax Theorem states:

$$\min_{\boldsymbol{x} \in K} \sup_{\beta \in M} \mathcal{L}(\boldsymbol{x}, \beta) = \sup_{\beta \in M} \min_{\boldsymbol{x} \in K} \mathcal{L}(\boldsymbol{x}, \beta)$$

Therefore,

$$L^\star = \sup_{\beta \geq 0} \min_{\boldsymbol{x} \in \mathcal{X}} \mathcal{L}(\boldsymbol{x}, \beta)$$

Since $\mathcal{F}$ is non-empty, $L^\star$ is finite, implying that the supremum is attained (or is the limit if approached at infinity, but $D(\beta)$ is continuous), so we can write $\max$ instead of $\sup$. Let $d(\beta) = \min_{\boldsymbol{x} \in \mathcal{X}} \mathcal{L}(\boldsymbol{x}, \beta)$.

$$d(\beta) = \min_{\boldsymbol{x} \in \mathcal{X}} \left( \boldsymbol{a}^\top \boldsymbol{x} + c + \beta\boldsymbol{g}^\top \boldsymbol{x} + \beta h \right)$$
$$= \min_{\boldsymbol{x} \in \mathcal{X}} \left( (\boldsymbol{a} + \beta\boldsymbol{g})^\top \boldsymbol{x} \right) + c + \beta h$$

The inner minimization is optimizing a linear function $(\boldsymbol{a} + \beta\boldsymbol{g})^\top \boldsymbol{x}$ over the hyper-rectangle $\mathcal{X}$. Using Lemma B.1 (with $\boldsymbol{v} = \boldsymbol{a} + \beta\boldsymbol{g}$):

$$\min_{\boldsymbol{x} \in \mathcal{X}} \left( (\boldsymbol{a} + \beta\boldsymbol{g})^\top \boldsymbol{x} \right) = (\boldsymbol{a} + \beta\boldsymbol{g})^\top \hat{\boldsymbol{x}} - |\boldsymbol{a} + \beta\boldsymbol{g}|^\top \boldsymbol{\epsilon}$$

where $|\boldsymbol{a} + \beta\boldsymbol{g}|^\top \boldsymbol{\epsilon} = \sum_{j=1}^n |a_j + \beta g_j| \epsilon_j$. Substituting this into the expression for $d(\beta)$, we get:

$$d(\beta) = (\boldsymbol{a} + \beta\boldsymbol{g})^\top \hat{\boldsymbol{x}} - |\boldsymbol{a} + \beta\boldsymbol{g}|^\top \boldsymbol{\epsilon} + c + \beta h$$

This is exactly the dual objective function $D(\beta)$ defined in the theorem statement (3). Thus, we have established that $L^\star = \max_{\beta \geq 0} D(\beta)$. Then we analyze the properties of the dual objective $D(\beta)$:

1. Concavity: The dual function $d(\beta)$ is always concave, as it is the pointwise minimum of a family of functions that are affine in $\beta$ (indexed by $\boldsymbol{x} \in \mathcal{X}$).

2. Piecewise-Linearity: The term $-|\boldsymbol{a} + \beta\boldsymbol{g}|^\top \boldsymbol{\epsilon} = -\sum_{j=1}^n |a_j + \beta g_j| \epsilon_j$ involves the absolute value function. Each term $-|a_j + \beta g_j| \epsilon_j$ is concave and piecewise-linear, with a breakpoint (a point where the slope changes) at $\beta = -a_j/g_j$ (if $g_j \neq 0$). The other terms in $D(\beta)$ are linear in $\beta$. Since $D(\beta)$ is a sum of concave piecewise-linear functions and linear functions, it is itself concave and piecewise-linear. The breakpoints of $D(\beta)$ are the collection of all values $\beta = -a_j/g_j \geq 0$ where $g_j \neq 0$.

Thus, we need to maximize the concave, piecewise-linear function $D(\beta)$ over the interval $[0, \infty)$. Since $D(\beta)$ is concave, its maximum over a convex set occurs either at a point where the super-gradient contains zero, or potentially at the boundary point $\beta = 0$. Because $D(\beta)$ is piecewise-linear, its super-gradient $\partial D(\beta)$ is constant within the linear segments between breakpoints. At a breakpoint $\beta_k$, the super-gradient is an interval $[\partial D(\beta_k^-), \partial D(\beta_k^+)]$ (the range between the left and right derivatives). The maximum occurs at a point $\beta^\star$ such that $0 \in \partial D(\beta^\star)$. This $\beta^\star$ must be either $\beta = 0$ (if the derivative is non-positive for $\beta > 0$) or one of the breakpoints $\beta_k > 0$ where the derivative changes sign from positive to non-positive (i.e., $0 \in [\partial D(\beta_k^-), \partial D(\beta_k^+)]$ ), or the function increases indefinitely (which corresponds to an infeasible or unbounded primal, but we assumed feasibility and the primal is bounded over the compact $\mathcal{X}$, so $L^\star$ is finite, thus the dual maximum is finite).

Therefore, the maximum $L^\star$ can be found non-iteratively by:

1. Identifying all non-negative breakpoints $\beta_k = -a_j/g_j \geq 0$.

2. Sorting these unique breakpoints $0 = \beta_0 < \beta_1 < \cdots < \beta_p$.

3. Evaluating the derivative (slope) of $D(\beta)$ within each segment $(\beta_k, \beta_{k+1})$ and potentially at $\beta = 0$.

4. Finding the point $\beta^\star$ (either 0 or some $\beta_k$) where the slope transitions from non-negative to non-positive. The value $D(\beta^\star)$ is the maximum $L^\star$.

This process involves a finite number of analytical calculations (evaluating slopes and function values at breakpoints) rather than iterative optimization, justifying the claim of efficiency. $\qquad\square$

### B.3 Equivalence to the Continuous Knapsack Problem

The primal problem in (2) is mathematically equivalent to the continuous (or fractional) knapsack problem. We can demonstrate this equivalence through a change of variables. Let $x_0 = \hat{x} - \epsilon$ be the lower bound and $x_1 = \hat{x} + \epsilon$ be the upper bound. We transform $x \in [x_0, x_1]$ to $y \in [0, 1]^n$ using:

$$y_i = \frac{x_i - x_{0,i}}{x_{1,i} - x_{0,i}} = \frac{x_i - (\hat{x}_i - \epsilon_i)}{2\epsilon_i} \quad\Longrightarrow\quad x_i = x_{0,i} + 2\epsilon_i y_i$$

Substituting this into the primal problem $\min_{x \in \mathcal{X}}\{a^\top x + c \mid g^\top x + h \le 0\}$ gives:

$$\min_{y \in [0,1]^n} \quad a^\top(x_0 + 2(\epsilon \odot y)) + c \quad \text{s.t.} \quad g^\top(x_0 + 2(\epsilon \odot y)) + h \le 0$$

$$\min_{y \in [0,1]^n} \quad (2\epsilon \odot a)^\top y + (a^\top x_0 + c) \quad \text{s.t.} \quad (2\epsilon \odot g)^\top y \le -(g^\top x_0 + h)$$

Let $r = -2\epsilon \odot a$, $s = 2\epsilon \odot g$, and $t = -(g^\top(\hat{x} - \epsilon) + h)$. The problem becomes the standard knapsack form:

$$\max_{y \in [0,1]^n} \quad r^\top y \quad \text{s.t.} \quad s^\top y \le t$$

This problem, even with negative coefficients, can be solved with a greedy algorithm [37]. The efficiency ratios $r_j/s_j$ used for sorting in the greedy algorithm are:

$$\frac{r_j}{s_j} = \frac{-2\epsilon_j a_j}{2\epsilon_j g_j} = -a_j/g_j$$

These ratios are identical to the breakpoints in our dual objective function $D(\beta)$. This confirms a line-by-line correspondence: our dual optimization algorithm, which sorts breakpoints to find where the super-gradient contains zero, is algorithmically equivalent to the greedy knapsack algorithm, which sorts by efficiency ratios. Both have the same $\mathcal{O}(n \log n)$ time complexity.

### B.4 Proof of Theorem 3.2

**Direct Intuitive Proof** A more direct and intuitive proof for Theorem 3.2 exists. We wish to solve $\bar{x}_i^{(new)} = \max_{x \in \mathcal{X}}\{x_i \mid a^\top x + c \le 0\}$. Since only $x_i$ is in the objective, we can set all other variables $x_j$ (for $j \ne i$) to values within their box domain $[\hat{x}_j - \epsilon_j, \hat{x}_j + \epsilon_j]$ that make the constraint $a^\top x + c \le 0$ as loose as possible, thereby maximizing the "budget" for $x_i$.

To loosen the constraint, we must minimize the term $\sum_{j \ne i} a_j x_j$. This is achieved by setting each $x_j$ to its extreme:

- If $a_j > 0$, we set $x_j$ to its lower bound, $x_j = \hat{x}_j - \epsilon_j$.
- If $a_j < 0$, we set $x_j$ to its upper bound, $x_j = \hat{x}_j + \epsilon_j$.

This worst-case minimum for the sum can be written compactly as $\sum_{j \ne i}(a_j \hat{x}_j - |a_j|\epsilon_j)$. We substitute this minimum sum back into the constraint:

$$a_i x_i + \sum_{j \ne i}(a_j \hat{x}_j - |a_j|\epsilon_j) + c \le 0$$

Assuming $a_i > 0$, we can solve for $x_i$ to find its new upper bound:

$$x_i \leq \frac{-\sum_{j\neq i} a_j \hat{x}_j + \sum_{j\neq i} |a_j|\epsilon_j - c}{a_i}$$

This value is $x_i^{(\text{clip})}$, as defined in Theorem 3.2. The final bound $\overline{x}_i^{(new)}$ is the minimum of this value and the original upper bound $\overline{x}_i$. The case for $a_i < 0$ (updating the lower bound $\underline{x}_i$) follows analogously. A detailed proof is shown below:

*Proof.* We consider the upper bound; the lower bound can be derived analogously. First, we rewrite the input region as

$$\mathcal{X} = \{\boldsymbol{x} \in \mathbb{R}^n : \underline{\boldsymbol{x}} \leq \boldsymbol{x} \leq \overline{\boldsymbol{x}}\} = \{\boldsymbol{x} \in \mathbb{R}^n : \hat{\boldsymbol{x}} - \boldsymbol{\epsilon} \leq \boldsymbol{x} \leq \hat{\boldsymbol{x}} + \boldsymbol{\epsilon}\},$$

where $\hat{\boldsymbol{x}} = \frac{\overline{\boldsymbol{x}} + \underline{\boldsymbol{x}}}{2}$ and $\boldsymbol{\epsilon} = \frac{\overline{\boldsymbol{x}} - \underline{\boldsymbol{x}}}{2}$. Suppose the linear inequality constraint is given by $\boldsymbol{a}^\top \boldsymbol{x} + b \leq 0$, where $\boldsymbol{a} \in \mathbb{R}^n$ and $b \in \mathbb{R}$. We note that the intersection $\mathcal{X} \cap \{\boldsymbol{x} \in \mathbb{R}^n : \boldsymbol{a}^\top \boldsymbol{x} + b \leq 0\}$ is nonempty if and only if

$$0 \geq \min_{\boldsymbol{x} \in \mathcal{X}} \boldsymbol{a}^\top \boldsymbol{x} + b = \boldsymbol{a}^\top \hat{\boldsymbol{x}} + b - \sum_{i=1}^n |a_i|\epsilon_i$$

by Lemma B.1. Henceforth, we will assume this inequality is satisfied.

We now compute

$$\begin{aligned}
\overline{x}_i^{(new)} &= \max_{\boldsymbol{x} \in \mathcal{X}}\{\boldsymbol{e}_i^\top \boldsymbol{x} : \boldsymbol{a}^\top \boldsymbol{x} + b \leq 0\} \\
&= \max_{\boldsymbol{x} \in \mathcal{X}} \min_{\lambda \in \mathbb{R}_+} \boldsymbol{e}_i^\top \boldsymbol{x} - \lambda(\boldsymbol{a}^\top \boldsymbol{x} + b) \\
&= \min_{\lambda \in \mathbb{R}_+} \max_{\boldsymbol{x} \in \mathcal{X}} \boldsymbol{e}_i^\top \boldsymbol{x} - \lambda(\boldsymbol{a}^\top \boldsymbol{x} + b) \\
&= \min_{\lambda \in \mathbb{R}_+} \boldsymbol{e}_i^\top \hat{\boldsymbol{x}} - \lambda(\boldsymbol{a}^\top \hat{\boldsymbol{x}} + b) + |\boldsymbol{e}_i - \lambda\boldsymbol{a}|^\top \boldsymbol{\epsilon},
\end{aligned}$$

where third line follows from Sion's minimax theorem since $\mathcal{X}$ is a compact set, and the final line follows from Lemma B.1.

Rearranging the term yields

$$\overline{x}_i^{(new)} = \min_{\lambda \in \mathbb{R}_+} \boldsymbol{e}_i^\top \hat{\boldsymbol{x}} + \lambda\left(\sum_{j\neq i} \epsilon_j |a_j| - \boldsymbol{a}^\top \hat{\boldsymbol{x}} - b\right) + \epsilon_i|1 - \lambda a_i|. \tag{21}$$

We now analyze three cases for $a_i$ and derive a closed-form expression for the scalar minimization. Recall that we have assumed $0 \geq \boldsymbol{a}^\top \hat{\boldsymbol{x}} + c - \sum_{i=1}^n a_i\epsilon_i$ or equivalently $\sum_{j\neq i} \epsilon_j |a_j| - \boldsymbol{a}^\top \hat{\boldsymbol{x}} - b \geq -\epsilon_i|a_i|$. We have:

1. $\underline{a_i > 0}$: The function $\epsilon_i|1 - \lambda a_i|$ attains its minimum at $\lambda = \frac{1}{a_i} > 0$, with slope $\epsilon_i a_i$ on the right and $-\epsilon_i a_i$ on the left. Thus:

   - If $|\sum_{j\neq i} \epsilon_j |a_j| - \boldsymbol{a}^\top \hat{\boldsymbol{x}} - b| \leq \epsilon_i a_i$ then the minimum of (21) is attained at $\lambda^\star = \frac{1}{a_i}$ and

     $$\overline{x}_i^{(new)} = \hat{x}_i + \frac{\sum_{j\neq i} \epsilon_j |a_j| - \boldsymbol{a}^\top \hat{\boldsymbol{x}} - b}{a_i} = \frac{\sum_{j\neq i} \epsilon_j |a_j| - \sum_{j\neq i} a_j \hat{x}_j - b}{a_i}.$$

   - If $\sum_{j\neq i} \epsilon_j |a_j| - \boldsymbol{a}^\top \hat{\boldsymbol{x}} - b > \epsilon_i a_i$ then $\lambda^\star = 0$ and

     $$\overline{x}_i^{(new)} = \hat{x}_i + \epsilon_i = \overline{x}_i.$$

2. $\underline{a_i < 0}$: In this case, the minimizing $\lambda = \frac{1}{a_i} < 0$ is infeasible to (21). Hence, $\lambda^\star = 0$ and $\overline{x}_i^{(new)} = \overline{x}_i$.

3. $a_i = 0$: Here, the objective function in (21) becomes affine in $\lambda$. Since $\sum_{j \neq i} \epsilon_j |a_j| - \boldsymbol{a}^\top \hat{\boldsymbol{x}} - b \geq 0$, we again have $\lambda^\star = 0$ and $\overline{x}_i^{(new)} = \overline{x}_i$.

In summary, if $\boldsymbol{\epsilon}^\top |\boldsymbol{a}| - \boldsymbol{a}^\top \hat{\boldsymbol{x}} - b < 0$ then $\emptyset = \mathcal{X} \cap \{\boldsymbol{x} \in \mathbb{R}^n : \boldsymbol{a}^\top \boldsymbol{x} + c \leq 0\}$; otherwise,

$$\overline{x}_i^{(new)} = \begin{cases} \min\left\{\dfrac{\sum_{j \neq i} \epsilon_j |a_j| - \sum_{j \neq i} a_j \hat{x}_j - b}{a_i}, \overline{x}_i\right\} & \text{if } a_i > 0 \\ \overline{x}_i & \text{if } a_i \leq 0. \end{cases}$$

This completes the proof. $\qquad\square$

### B.5 Proof of Order Dependency

Here we briefly introduce Theorem B.2 to demonstrate why sequential processing the constraints for clipping achieving better results.

**Theorem B.2** (Order Dependency of Constraint Intersection). *Let $\mathcal{X} = \{x \in \mathbb{R}^n : x_L \leq x \leq x_U\}$ be a box domain, and let $F_j = \{x \in \mathcal{X} : a_j^\top x + c_j \leq 0\}$ denote the feasible set for the $j$-th constraint. Define the full feasible set as $F = \bigcap_{j=1}^m F_j$.*

*Let $\pi : \{1, \ldots, m\} \to \{1, \ldots, m\}$ be a permutation (i.e., a reordering) of the constraints. For sequential intersection, define:*

$$F^{(k)} = \bigcap_{t=1}^k F_{\pi(t)}, \quad k = 1, \ldots, m,$$

*with $F^{(0)} = \mathcal{X}$.*

*(Order-Independent Bounds): The variable-wise bounds satisfy:*

$$\max_{x \in F} x_i \leq \min_j \max_{x \in F_j} x_i \quad \text{and} \quad \min_{x \in F} x_i \geq \max_j \min_{x \in F_j} x_i.$$

*These bounds are independent of $\pi$ if computed via $F = \bigcap_{j=1}^m F_j$.*

*(Order-Dependent Refinement): If constraints are intersected sequentially (i.e., $F^{(k)}$ depends on $\pi$), then there exist permutations $\pi_1 \neq \pi_2$ such that:*

$$F_{\pi_1}^{(m)} \neq F_{\pi_2}^{(m)}.$$

*Proof.* Order-Independent Bounds. Same as Corollary B.2. The inequalities follow from $F \subseteq F_j$ for all $j$. Simultaneous intersection satisfies:

$$\max_{x \in F} x_i \leq \min_j \max_{x \in F_j} x_i \quad \text{and} \quad \min_{x \in F} x_i \geq \max_j \min_{x \in F_j} x_i,$$

as the intersection $F$ cannot exceed the tightest bound from any $F_j$.

Order-Dependent Refinement. Let $F^{(k)} = \bigcap_{t=1}^k F_{\pi(t)}$. For dependent constraints (e.g., $F_1$ bounds $x_1$, $F_2$ depends on $x_1$), the sequence $F^{(k)}$ depends on $\pi$. A counterexample with $F_1 : x_1 + x_2 \leq 2$ and $F_2 : x_1 - x_2 \leq 0$ shows:

1. Intersecting $F_1$ first gives $x_1 \leq 2 - x_2$, then $F_2$ further tightens $x_1 \leq x_2$.

2. Intersecting $F_2$ first gives $x_1 \leq x_2$, then $F_1$ tightens $x_1 \leq 1$.

The final bounds differ: $x_1 \leq \min(2 - x_2, x_2)$ vs. $x_1 \leq 1$. $\qquad\square$

## C Algorithms

### C.1 Overview

Fig. 3 shows the pipeline of our algorithm.

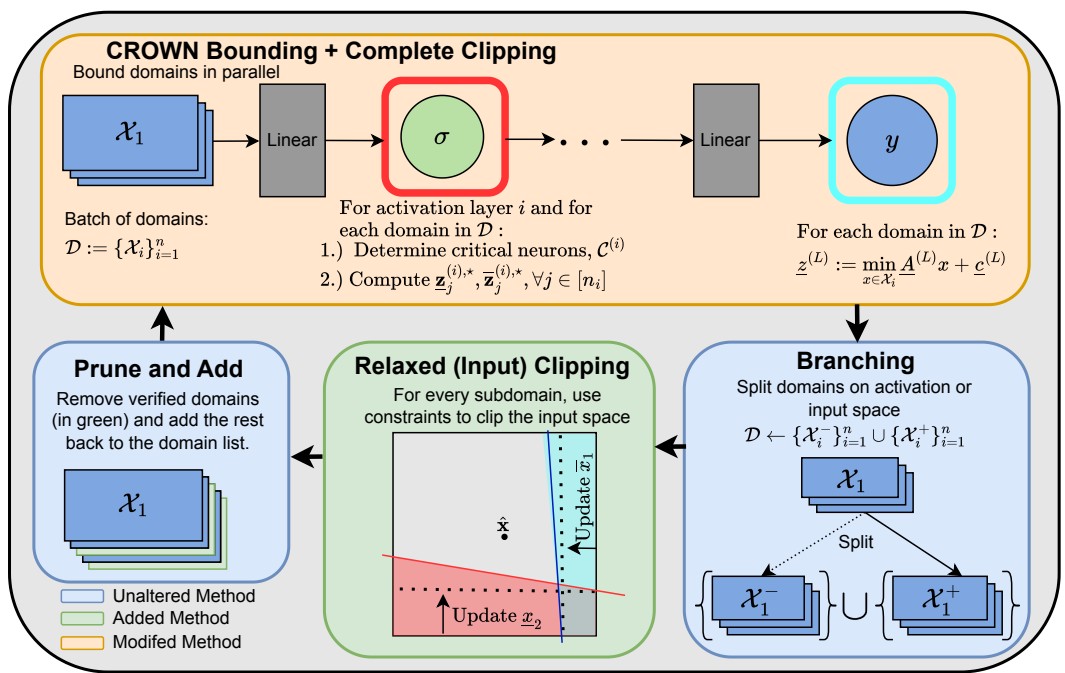

Figure 3: Full Clip-and-Verify pipeline for (Input and Activation Activation) BaB integration.

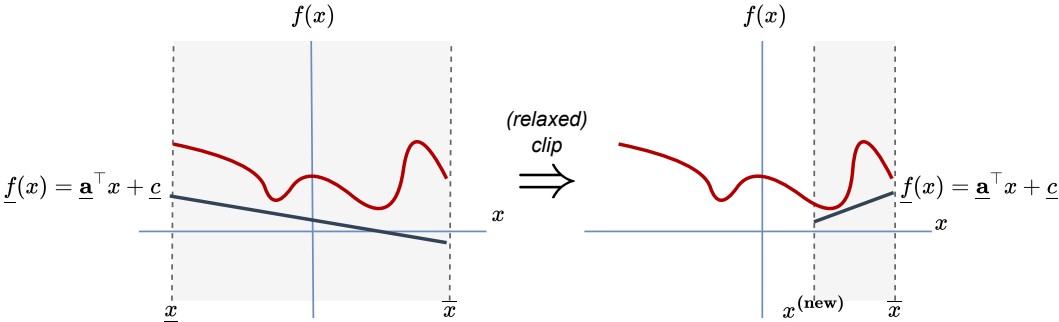

Figure 4: Simple 1D visualization of relaxed clipping reducing the input interval, potentially enabling a second pass of bound propagation to produce a tighter bound.

## C.2   Sequential Clipping for Multiple Constraints

When several linear constraints are present, we apply Theorem 3.2 *sequentially* to each row of $Ax + c \leq 0$. Algorithm 3 outlines this procedure: First, for each constraint $k$, compute $\hat{x}$ and $\epsilon$ from the *current* bounds, $\underline{x}$ and $\overline{x}$. Then, use Theorem 3.2 to refine $\underline{x}_i$ and $\overline{x}_i$ for all $i$. Finally, proceed to the next constraint, using the newly clipped bounds as the domain. Note that if $\underline{x}_i > \overline{x}_i$ occurs in any dimension, the input region of this subproblem is infeasible, and we can directly verify this subproblem without performing further verification.

Because the bounds are updated after each constraint, the final domain is an over-approximation of the true feasible region under all constraints simultaneously. However, it is still far more efficient than solving a multi-constraint system in one shot, making it well suited for large-scale verification where we repeatedly clip domains across many subproblems.

By clipping the domain repeatedly, Algorithm 3 retains enough precision to prune large regions yet remains computationally lightweight enough for repeated invocation on many subdomains during verification. It may even be the case that the constraints passed to our domain clipping algorithm reveal the region is entirely *infeasible*. Such a scenario may occur when two Activation assignments

---

**Algorithm 3** Linear Constraint-Driven Relaxed Clipping *(sequential)*

---

**Require:** $\underline{x}$ : Lower bounds; $\overline{x}$ : Upper bounds; $A$ : Constraint matrix; $c$ : Constraint vector.

1: $m \leftarrow \text{rows}(A), n \leftarrow \text{cols}(A)$
2: **for** for each constraint $k \in \{1, \ldots, m\}$ **do**
3:    **for** each input dimension $i \in \{1, \ldots, n\}$ **do**
4:       $\hat{x} \leftarrow \frac{\overline{x}+\underline{x}}{2}, \epsilon \leftarrow \frac{\overline{x}-\underline{x}}{2}$
5:       **if** $A_{k,i} \neq 0$ **then**
6:          $x_i^{(\text{new})} \leftarrow \frac{-\sum_{j \neq i} A_{k,j}\hat{x}_j + \sum_{j \neq i} |A_{k,j}|\epsilon_j - b}{A_{k,i}}$
7:          **if** $A_{k,i} > 0$ **then**
8:             $\overline{x}_i^{(\text{new})} \leftarrow \min(\overline{x}_i, x_i^{(\text{new})})$
9:          **else**
10:            $\underline{x}_i^{(\text{new})} \leftarrow \max(\underline{x}_i, x_i^{(\text{new})})$
11:    $\underline{x}_i \leftarrow \max(\underline{x}_i, \underline{x}_i^{(\text{new})})$
12:    $\overline{x}_i \leftarrow \min(\overline{x}_i, \overline{x}_i^{(\text{new})})$

**Ensure:** Clipped $\underline{x}, \overline{x}$

---

admit an infeasible domain, or when the desired property admits multiple constraints that produce infeasibility. In such scenarios, Algorithm 3 will return clipped bounds where $\underline{x}$ will be smaller than $\overline{x}$ along some dimension(s). In the context of branch-and-bound, this subdomain may be effectively pruned, avoiding the process of running bound propagation once more on the domain, avoiding unnecessary computations

## C.3 Algorithms for Clip-and-Verify in Input Branch-and-bound Scheme

---

**Algorithm 4** Clip-and-Verify for Input Branch-and-bound

---

**Require:** $f$ : model to verify; $n$ : batch size; timeout : time-out threshold

1: $\mathcal{D}_{\text{Unknown}}, \underline{f} \leftarrow \text{Init}(f, \emptyset)$ {Initialize the set of unknown subdomains $\mathcal{D}_{\text{Unknown}}$ and global bound $\underline{f}$}
2: **while** $|\mathcal{D}_{\text{Unknown}}| > 0$ **and** not timed out **do**
3:    $\{\mathcal{X}_i, \underline{A}_i^{\text{prev}}, \underline{c}_i^{\text{prev}}\}_{i=1}^n \leftarrow \text{Batch\_Pick\_Out}(\mathcal{D}_{\text{Unknown}}, n)$ {Pick up to $n$ subdomains $\mathcal{X}_i$ and their associated hyperplanes $(\underline{A}_i^{\text{prev}}, \underline{c}_i^{\text{prev}})$ from the previous iteration.}
4:    $\{\mathcal{C}_i\}_{i=1}^n \leftarrow \text{Top-K\_Heuristic}(\{\mathcal{X}_i, \underline{A}_i^{\text{prev}}, \underline{c}_i^{\text{prev}}\}_{i=1}^n)$ {Determine critical neurons $\mathcal{C}_i$ using a top-k heuristic over each domain and its previous hyperplanes.}
5:    $(\underline{f}_{\mathcal{X}_1}, \underline{A}_{\mathcal{X}_1}, \underline{c}_{\mathcal{X}_1}, \ldots, \underline{f}_{\mathcal{X}_n}, \underline{A}_{\mathcal{X}_n}, \underline{c}_{\mathcal{X}_n}) \leftarrow \text{Solve\_Bound}\left(f, \{\mathcal{X}_i, \mathcal{C}_i\}_{i=1}^n\right)$ {Compute bounds and plane coefficients on each subdomain; Refine critical neurons using planes as constraints and *Complete Clipping*.}
6:    $\{\mathcal{X}_i^-, \mathcal{X}_i^+, \underline{A}_i, \underline{c}_i\}_{i=1}^n \leftarrow \text{Batch\_Split}\left(\{\mathcal{X}_i, \underline{A}_i, \underline{c}_i\}_{i=1}^n\right)$ {Split each subdomain; share hyperplanes with both children.}
7:    $\{x_i^{(-)}, x_i^{(+)}\}_{i=1}^n \leftarrow \text{Relaxed\_Clipper}\left(\{\mathcal{X}_i^-, \mathcal{X}_i^+, \underline{A}_i, \underline{c}_i\}_{i=1}^n\right)$ {Refine input box on each child subdomain using the plane coefficients.}
8:    $\{\mathcal{X}_i^-, \mathcal{X}_i^+\}_{i=1}^n \leftarrow \text{Domain\_Update}\left(\{x_i^{(-)}, x_i^{(+)}\}_{i=1}^n\right)$ {Update each child subdomain's input bounds.}
9:    $\mathcal{D}_{\text{Unknown}} \leftarrow \mathcal{D}_{\text{Unknown}} \cup \text{Domain\_Filter}\left(\left[\underline{f}_{\mathcal{X}_1^-}, \mathcal{X}_1^-, \underline{A}_1^{(-)}, \underline{c}_1^{(-)}\right], \ldots\right)$ {Filter out verified/infeasible subdomains. Retain unknowns and their bounding hyper-planes.}

**Ensure:** UNSAT if $|\mathcal{D}_{\text{Unknown}}| = 0$ else Unknown

---

Algorithm 4 describes our modifications to the standard BaB procedure. When all subproblems can be verified, then the verification problem is referred to as UNSAT (i.e., the complementary property, $\exists x \in \mathcal{X}, \ f(x) < 0$, is *unsatisfiable*), and the network is *safe* from counter-examples. Otherwise, it is insufficient to determine if the property is UNSAT without further refinement or falsification.

## C.4 Algorithms for Clip-and-Verify in Activation Branch-and-bound Scheme

---

**Algorithm 5** Clip-and-Verify for Activation Branch-and-bound

---

**Require:** $f$ : model to verify; $n$ : batch size; timeout : time-out threshold

1: $\mathcal{D}_{\mathrm{Unknown}}, \underline{f} \leftarrow \mathrm{Init}(f, \emptyset)$ {Initialize the set of unknown subdomains $\mathcal{D}_{\mathrm{Unknown}}$ and global bound $\underline{f}$}

2: $\{\underline{\boldsymbol{A}}^{(j)}, \underline{\boldsymbol{c}}^{(j)}, \overline{\boldsymbol{A}}^{(j)}, \overline{\boldsymbol{c}}^{(j)}, \mathrm{unstable\_neuron(idx)}^{(j)}\}_{j=1}^{J} \leftarrow \mathrm{Get\_Constraints}(f)$ {Retrieve the set of coefficients and biases for each unstable neurons' upper and lower bound and get the indices of unstable neurons during Bound Propagation.}

3: $\mathrm{Domain\_Clipper} \leftarrow \mathrm{Init\_Clipper}(\{\underline{\boldsymbol{A}}^{(j)}, \underline{\boldsymbol{c}}^{(j)}, \overline{\boldsymbol{A}}^{(j)}, \overline{\boldsymbol{c}}^{(j)}, \mathrm{idx}^{(j)}\}_{j=1}^{J})$ {Initialize the Domain Clipper with the set of constraints information.}

4: **while** $|\mathcal{D}_{\mathrm{Unknown}}| > 0$ **and** not timed out **do**

5:     $\{\mathcal{Z}_i\}_{i=1}^{n} \leftarrow \mathrm{Batch\_Pick\_Out}(\mathcal{D}_{\mathrm{Unknown}}, n)$ {Pick at most $n$ subdomains from the unknown set.}

6:     $\{\mathcal{Z}_i^{-}, \mathcal{Z}_i^{+}\}_{i=1}^{n} \leftarrow \mathrm{Batch\_Split}(\{\mathcal{Z}_i\}_{i=1}^{n})$ {Split each subdomain (e.g., Activation split or input split) into two child subdomains.}

7:     $\{\mathcal{C}_i^{(-)}, \mathcal{C}_i^{(+)}\}_{i=1}^{n} \leftarrow \mathrm{Top\text{-}K\_Heuristic}(\{\mathcal{X}_i\}_{i=1}^{n})$ {Determine critical neurons using a top-k heuristic (e.g. BaBSR) over each domain.}

8:     $\{\mathrm{x}_i^{(-)}, \mathrm{interm\_bds}_i^{(-)}, \mathrm{x}_i^{(+)}, \mathrm{interm\_bds}_i^{(+)}\}_{i=1}^{n} \leftarrow \mathrm{Domain\_Clipper}(\{\mathcal{Z}_i^{-}, \mathcal{Z}_i^{+}, \mathcal{C}_i^{(-)}, \mathcal{C}_i^{(+)}\}_{i=1}^{n})$ {Apply Relaxed Clipping to child subdomain's input and Complete Clipping on the critical neurons. Constraints are the Activation splits.}

9:     $\{\mathcal{Z}_i^{-}, \mathcal{Z}_i^{+}\}_{i=1}^{n} \leftarrow \mathrm{Domain\_Update}(\{\mathrm{x}_i^{(-)}, \mathrm{interm\_bds}_i^{(-)}, \mathrm{x}_i^{(+)}, \mathrm{interm\_bds}_i^{(+)}\}_{i=1}^{n})$ {Update each child subdomain's input and intermediate bounds.}

10:     $(\underline{f}_{\mathcal{Z}_1^{-}}, \ldots, \underline{f}_{\mathcal{Z}_n^{+}}) \leftarrow \mathrm{Solve\_Bound}\left(f, \{\mathrm{x}_i, \mathrm{interm\_bounds}_i, \mathcal{Z}_i\}\right)$ {Compute bounds on each newly clipped subdomain using a bound propagation solver.}

11:     $\mathcal{D}_{\mathrm{Unknown}} \leftarrow \mathcal{D}_{\mathrm{Unknown}} \cup \mathrm{Domain\_Filter}\left([\underline{f}_{\mathcal{Z}_1^{-}}, \mathcal{Z}_1^{-}], \ldots\right)$ {Filter out verified/infeasible subdomains. Keep remaining unknown subdomains in $\mathcal{D}_{\mathrm{Unknown}}$.}

**Ensure:** UNSAT if $|\mathcal{D}_{\mathrm{Unknown}}| = 0$ else Unknown

---

Algorithm 5 begins by initializing the verification procedure. First, we call $\mathrm{Init}(f, \emptyset)$ to obtain an empty set of partial Activation assignments (or subdomains) along with an initial global lower bound $\underline{f}$ for the property to be checked. This global lower bound can, for example, be the result of a quick bounding pass. The set $\mathcal{D}_{\mathrm{Unknown}}$ is then populated with a single "root" subdomain representing the entire input domain.

Next, we retrieve the constraint information for all neurons (Line 2). Specifically, $\mathrm{Get\_Constraints}(f)$ returns the linear coefficients and biases used to bound each neuron's activation, distinguishing between the lower-bounding $(\underline{\boldsymbol{A}}^{(j)}, \underline{\boldsymbol{c}}^{(j)})$ and upper-bounding $(\overline{\boldsymbol{A}}^{(j)}, \overline{\boldsymbol{c}}^{(j)})$ linear functions. It also identifies indices of "unstable" Activation neurons whose ranges straddle zero. These constraints will later be used to restrict the feasible input region using both our Relaxed Clipping and Complete Clipping algorithms.

We then initialize the Domain Clipper (Line 3) with the gathered linear constraints. This Clipper component will be invoked whenever we branch on a Activation neuron, so that the corresponding partial assignment (e.g., $x_k^{(j)} \geq 0$) is "pushed back" onto the input domain. In doing so, we clip the subdomain's input box by applying Theorem 3.2 (or its extensions) to incorporate these newly introduced constraints, thus discarding parts of the input space that violate them. In addition, when given a set of critical neurons, these Activation split constraints will be used to directly refine the intermediate neurons using Theorem 3.1.

The main loop (Lines 5–11) iterates until either no subdomains remain unknown or a time-out is reached. In each iteration, we pick up to $n$ unknown subdomains from $\mathcal{D}_{\mathrm{Unknown}}$ (Line 5) for batched parallel processing. Each subdomain $\mathcal{Z}_i$ is then split (Line 6) along one or more unstable Activation neurons, creating child subdomains in which each split neuron is fixed to either the active $(\geq 0)$ or inactive $(\leq 0)$ regime.

At Line 7, we invoke a top-k heuristic (e.g. BaBSR) in order to determine the set of "critical neurons" that would contribute the most to providing a stronger convex relaxation if their bounds were to be refined.

At Lines 8–9, we invoke the Domain Clipper on these newly formed child subdomains. The Clipper translates each Activation assignment into a linear constraint on the input, then refines (or "clips") the child subdomain's input bounds, and the bounds of the "critical neurons" in each subdomain with respect to the heuristic choices from the step prior. This ensures that any portion of the parent domain that contradicts the new constraint is removed. Once clipped, the child subdomains' intermediate bounds are also updated (Line 10) so that subsequent bounding calculations reflect the tighter input ranges.

We then compute bounds on the newly clipped subdomains (11) using a chosen method—often a fast bound-propagation tool such as CROWN or a lightweight LP solver. This step yields lower bounds $\underline{f}_{\mathcal{Z}_i^\pm}$ on the network outputs for each subdomain. If these bounds confirm that the property holds (e.g., a robustness margin remains non-negative), the subdomain is verified and can be pruned. If the subdomain is infeasible (e.g., constraints are contradictory), it is also removed. All remaining subdomains (still "unknown") return to $\mathcal{D}_{\text{Unknown}}$ for further splitting.

Finally, the loop terminates once there are no unknown subdomains left or the time-out is reached. If $\mathcal{D}_{\text{Unknown}}$ becomes empty, we conclude UNSAT, signifying that no violating input (counterexample) exists within any subdomain. Otherwise, we return "Unknown," indicating that verification was not completed in time.

Overall, this procedure reflects a standard Activation BaB flow, except that an additional "domain clipping" step (Lines 8–9) is inserted after each split, leveraging partial Activation assignments to refine the input domain and select intermediate neurons before the next bounding pass. By applying our linear constraint-driven clipping algorithms whenever new constraints appear, we gain significantly tighter intermediate-layer bounds and thus reduce the branching burden throughout the verification process.

### C.5  2D Toy Example

To illustrate our clipping verification approach, we consider a simple two-layer ReLU feed-forward network defined as
$$f(\boldsymbol{x}) = \boldsymbol{w}^{(2)\top} \sigma(\boldsymbol{W}^{(1)}\boldsymbol{x} + \boldsymbol{b}^{(1)}), \tag{22}$$
where $\boldsymbol{w}^{(2)}, \boldsymbol{b}^{(1)}, \boldsymbol{x} \in \mathbb{R}^2$, $\boldsymbol{W}^{(1)} \in \mathbb{R}^{2\times 2}$, and $\sigma(\cdot)$ denotes the element-wise ReLU activation. We aim to verify the property
$$f(\boldsymbol{x}) \geq 0, \quad \forall \boldsymbol{x} \in \mathcal{X}.$$
The input domain $\mathcal{X}$ is defined as an $\ell_\infty$-box centered at $\hat{\boldsymbol{x}} = [0.5, -0.5]^\top$ with half-widths $\boldsymbol{\epsilon} = [1.5, 1.5]^\top$. Equivalently, the domain can be expressed using its endpoints $\underline{\boldsymbol{x}} = [-1, -2]^\top$ and $\overline{\boldsymbol{x}} = [2, 1]^\top$. The network parameters are specified as
$$\left\{ \boldsymbol{W}^{(1)} = \begin{bmatrix} 1 & -7 \\ 5 & -1 \end{bmatrix}, \quad \boldsymbol{b}^{(1)} = \begin{bmatrix} 6 \\ -7 \end{bmatrix}, \quad \boldsymbol{w}^{(2)} = \begin{bmatrix} 1 \\ -1 \end{bmatrix}. \tag{23}$$

**CROWN Bound.** As a warm up, let's bound the network using the CROWN algorithm without utilizing relaxed nor complete clipping. We begin by computing the pre-activation bounds of the intermediate layer which are obtained by concretizing the first layer's affine transformation:
$$\underline{\boldsymbol{z}} = \min_{\boldsymbol{x} \in \mathcal{X}} \boldsymbol{W}^{(1)}\boldsymbol{x} + \boldsymbol{b}^{(1)}, \qquad \overline{\boldsymbol{z}} = \max_{\boldsymbol{x} \in \mathcal{X}} \boldsymbol{W}^{(1)}\boldsymbol{x} + \boldsymbol{b}^{(1)}. \tag{24a}$$
$$\underline{\boldsymbol{z}} = \boldsymbol{W}^{(1)}\hat{\boldsymbol{x}} - |\boldsymbol{W}^{(1)}|\boldsymbol{\epsilon} + \boldsymbol{b}^{(1)}, \qquad \overline{\boldsymbol{z}} = \boldsymbol{W}^{(1)}\hat{\boldsymbol{x}} + |\boldsymbol{W}^{(1)}|\boldsymbol{\epsilon} + \boldsymbol{b}^{(1)}. \tag{24b}$$
Substituting the parameters into equation (24b) yields $\underline{\boldsymbol{z}} = [-2, -13]^\top$ and $\overline{\boldsymbol{z}} = [22, 5]^\top$. These neurons are then passed to a ReLU activation function, and one should notice that for each input, we have that $\underline{z}_1 < 0 < \overline{z}_1$ and $\underline{z}_2 < 0 < \overline{z}_2$. In this scenario, we have that the ReLU neurons are *unstable*, meaning that we cannot bound the non-linearity exactly, however, we can still get sound linear bounds on the output of the ReLU activation. Using Lemma A.1, we construct the diagonal matrices and corresponding vectors,
$$\underline{\boldsymbol{D}} = \begin{bmatrix} \alpha_1 & 0 \\ 0 & \alpha_2 \end{bmatrix}, \qquad \underline{\boldsymbol{b}} = \begin{bmatrix} 0 \\ 0 \end{bmatrix} \tag{25a}$$

$$\overline{\boldsymbol{D}} = \begin{bmatrix} \frac{\overline{z}_1}{\overline{z}_1 - \underline{z}_1} & 0 \\ 0 & \frac{\overline{z}_2}{\overline{z}_2 - \underline{z}_2} \end{bmatrix}, \qquad \overline{\boldsymbol{b}} = \begin{bmatrix} \frac{-\overline{z}_1 \underline{z}_1}{\overline{z}_1 - \underline{z}_1} \\ \frac{-\overline{z}_2 \underline{z}_2}{\overline{z}_2 - \underline{z}_2} \end{bmatrix}. \tag{25b}$$

$\underline{\boldsymbol{D}}$ and $\underline{\boldsymbol{b}}$ are used to create *lower bounding* planes on the output of the ReLU activation where $\alpha_1$ and $\alpha_2$ are real numbers limited to the range $[0, 1]$. These values may be optimized, however we will always fix $\alpha_1 = \alpha_2 = 1$ when lower bounding unstable neurons in this toy example. $\overline{\boldsymbol{D}}$ and $\overline{\boldsymbol{d}}$ are *upper bounding* planes on the output of the ReLU activation, and its construction is derived from the Planet relaxation [25]. It may be verified that for inputs $\boldsymbol{z}$ in the range $[\underline{\boldsymbol{z}}, \overline{\boldsymbol{z}}]$,

$$\overline{\boldsymbol{D}}\boldsymbol{z} + \overline{\boldsymbol{b}} \geq \sigma(\boldsymbol{z}) \geq \underline{\boldsymbol{D}}\boldsymbol{z} + \underline{\boldsymbol{b}}. \tag{26}$$

We next lower bound the network output. Because the post-activation vector $\sigma(\mathbf{z})$ is passed through the final linear layer, a coordinate-wise sign on the final-layer weights determines whether to use upper or lower affine bounds for each neuron. Concretely,

$$\begin{cases} w\sigma(\boldsymbol{z})_i \geq w \left( \underline{\boldsymbol{D}}_{i,i} \boldsymbol{z}_i + \underline{\boldsymbol{b}}_i \right), & \text{if } w \geq 0 \\ w\sigma(\boldsymbol{z})_i \geq w \left( \overline{\boldsymbol{D}}_{i,i} \boldsymbol{z}_i + \overline{\boldsymbol{b}}_i \right), & \text{if } w < 0 \end{cases} \tag{27}$$

so that the final-layer lower bound with respect to $\boldsymbol{z}$ is

$$\boldsymbol{w}^{(2)\top} \sigma(\boldsymbol{z}) \geq (\boldsymbol{w}^{(2),+})^\top (\underline{\boldsymbol{D}}\boldsymbol{z} + \underline{\boldsymbol{b}}) + (\boldsymbol{w}^{(2),-})^\top (\overline{\boldsymbol{D}}\boldsymbol{z} + \overline{\boldsymbol{b}}) \tag{28}$$

where $\boldsymbol{w}^{(2),+}$ zeros out entries which are negative and $\boldsymbol{w}^{(2),-}$ zeros out entries which are positive. In our example, these vectors are $\boldsymbol{w}^{(2),+} = [1, 0]^\top$ and $\boldsymbol{w}^{(2),-} = [0, -1]^\top$. The final step is to produce a lower bounding hyperplane of the final layer with respect to the network's input. So far, we've only related this lower bound to the intermediate input, $\boldsymbol{z}$. Our final step is quite simple as we know that $\boldsymbol{z} = \boldsymbol{W}^{(1)}\boldsymbol{x} + \boldsymbol{b}^{(1)}$. Using this relation, we can "back-propagate" our affine relaxations to the input as follows,

$$(\boldsymbol{w}^{(2),+})^\top (\underline{\boldsymbol{D}}\boldsymbol{z} + \underline{\boldsymbol{b}}) + (\boldsymbol{w}^{(2),-})^\top (\overline{\boldsymbol{D}}\boldsymbol{z} + \overline{\boldsymbol{b}}) \tag{29a}$$

$$= (\boldsymbol{w}^{(2),+})^\top \left( \underline{\boldsymbol{D}} \left( \boldsymbol{W}^{(1)}\boldsymbol{x} + \boldsymbol{b}^{(1)} \right) + \underline{\boldsymbol{b}} \right) + (\boldsymbol{w}^{(2),-})^\top \left( \overline{\boldsymbol{D}} \left( \boldsymbol{W}^{(1)}\boldsymbol{x} + \boldsymbol{b}^{(1)} \right) + \overline{\boldsymbol{b}} \right) \tag{29b}$$

$$= \left( \left( (\boldsymbol{w}^{(2),+})^\top \underline{\boldsymbol{D}} + (\boldsymbol{w}^{(2),-})^\top \overline{\boldsymbol{D}} \right) \boldsymbol{W}^{(1)} \right) \boldsymbol{x} + \left( (\boldsymbol{w}^{(2),+})^\top \underline{\boldsymbol{D}} + (\boldsymbol{w}^{(2),-})^\top \overline{\boldsymbol{D}} \right) \boldsymbol{b}^{(1)}$$
$$+ (\boldsymbol{w}^{(2),+})\underline{\boldsymbol{b}} + (\boldsymbol{w}^{(2),-})\overline{\boldsymbol{b}} \tag{29c}$$

Let us introduce the following variables,

$$\begin{cases} \underline{\boldsymbol{a}} := \left( (\boldsymbol{w}^{(2),+})^\top \underline{\boldsymbol{D}} + (\boldsymbol{w}^{(2),-})^\top \overline{\boldsymbol{D}} \right) \boldsymbol{W}^{(1)} \\ \underline{\boldsymbol{c}} := \left( (\boldsymbol{w}^{(2),+})^\top \underline{\boldsymbol{D}} + (\boldsymbol{w}^{(2),-})^\top \overline{\boldsymbol{D}} \right) \boldsymbol{b}^{(1)} + (\boldsymbol{w}^{(2),+})\underline{\boldsymbol{b}} + (\boldsymbol{w}^{(2),-})\overline{\boldsymbol{b}} \end{cases} \tag{30}$$

By construction, we have that the output of the network is lower bounded by the following linear relaxation,

$$f(\boldsymbol{x}) \geq \underline{\boldsymbol{a}}^\top \boldsymbol{x} + \underline{\boldsymbol{c}}, \qquad \boldsymbol{x} \in \mathcal{X}. \tag{31}$$

Using Hölder's inequality, we can solve for the minima of this lower bounding plane,

$$\min_{\boldsymbol{x} \in \mathcal{X}} f(\boldsymbol{x}) \geq \min_{\boldsymbol{x} \in \mathcal{X}} \underline{\boldsymbol{a}}^\top \boldsymbol{x} + \underline{\boldsymbol{c}} = \underline{\boldsymbol{a}}^\top \hat{\boldsymbol{x}} - |\underline{\boldsymbol{a}}|^\top \boldsymbol{\epsilon} + \underline{\boldsymbol{c}} = -\frac{19}{6}. \tag{32}$$

As this lower bound is too loose, we cannot verify our desired property, $f(\boldsymbol{x}) \geq 0, \forall \boldsymbol{x} \in \mathcal{X}$.

**Relaxed Clipping.** We first tackle this example using our efficient yet approximate clipping algorithm termed *relaxed clipping*. Given a set of constraints, the objective of relaxed clipping is to compute the smallest input box representation that satisfies those constraints. This step is performed once per round of branch-and-bound, and the resulting refined input box is shared by all neurons during the concretization step. Consequently, this refinement has the potential to tighten bounds across multiple neurons, including those in the final layer.

Consider a bound-propagation–based verifier executing branch-and-bound by splitting over the activation space. For the remainder of this subsection, we focus on the case where the first ReLU

neuron is split into its non-positive (inactive) region, i.e., $z_1 \leq 0$. This split decision can be reinterpreted as a linear constraint on the input,

$$(\boldsymbol{W}_{1,:}^{(1)})^\top \boldsymbol{x} + \boldsymbol{b}_1^{(1)} \leq 0. \tag{33}$$

Since the input $\boldsymbol{x}$ is two-dimensional, our goal is to tighten the lower and upper bounds of each input dimension under this constraint. Formally, we solve the following optimization problems:

$$\underline{\boldsymbol{x}}_1^{(rc)} := \min_{\boldsymbol{x} \in \mathcal{X}} \boldsymbol{x}_1 \quad \text{s.t.} \quad (\boldsymbol{W}_{1,:}^{(1)})^\top \boldsymbol{x} + \boldsymbol{b}^{(1)})_1 \leq 0, \tag{34a}$$

$$\overline{\boldsymbol{x}}_1^{(rc)} := \max_{\boldsymbol{x} \in \mathcal{X}} \boldsymbol{x}_1 \quad \text{s.t.} \quad (\boldsymbol{W}_{1,:}^{(1)})^\top \boldsymbol{x} + \boldsymbol{b}_1^{(1)} \leq 0, \tag{34b}$$

$$\underline{\boldsymbol{x}}_2^{(rc)} := \min_{\boldsymbol{x} \in \mathcal{X}} \boldsymbol{x}_2 \quad \text{s.t.} \quad (\boldsymbol{W}_{1,:}^{(1)})^\top \boldsymbol{x} + \boldsymbol{b}_1^{(1)} \leq 0, \tag{34c}$$

$$\overline{\boldsymbol{x}}_2^{(rc)} := \max_{\boldsymbol{x} \in \mathcal{X}} \boldsymbol{x}_2 \quad \text{s.t.} \quad (\boldsymbol{W}_{1,:}^{(1)})^\top \boldsymbol{x} + \boldsymbol{b}_1^{(1)} \leq 0. \tag{34d}$$

From Theorem 3.2, one potential solution along each dimension is given by

$$\boldsymbol{x}_i^{(clip)} := \frac{-\sum_{i \neq j} \left\{ \boldsymbol{W}_{1,j}^{(1)} \hat{\boldsymbol{x}}_j - |\boldsymbol{W}_{1,j}^{(1)}| \epsilon_j \right\} - \boldsymbol{b}_1^{(1)}}{\boldsymbol{W}_{1,i}^{(1)}} \tag{35}$$

Each dimension is then updated as follows:

$$\begin{cases} \overline{\boldsymbol{x}}_i^{(rc)} = \min \left\{ \boldsymbol{x}_i^{(\text{clip})}, \overline{\boldsymbol{x}}_i \right\} & \text{if } \boldsymbol{W}_{1,i}^{(1)} > 0 \\ \underline{\boldsymbol{x}}_i^{(rc)} = \max \left\{ \boldsymbol{x}_i^{(\text{clip})}, \underline{\boldsymbol{x}}_i \right\} & \text{if } \boldsymbol{W}_{1,i}^{(1)} < 0 \\ \text{no change} & \text{otherwise} \end{cases} \tag{36}$$

For the first dimension ($i = 1$), since $\boldsymbol{W}_{1,1}^{(1)} > 0$, the upper limit may be refined if $\boldsymbol{x}_1^{(\text{clip})} < \overline{\boldsymbol{x}}_1$. For the second dimension ($i = 2$), where $\boldsymbol{W}_{1,2}^{(1)} < 0$, the lower limit may be refined if $\boldsymbol{x}_2^{(\text{clip})} > \underline{\boldsymbol{x}}_2$. Substituting the given parameters yields:

$$\boldsymbol{x}_1^{(clip)} = \frac{-\boldsymbol{W}_{1,2}^{(1)} \hat{\boldsymbol{x}}_2 + |\boldsymbol{W}_{1,2}^{(1)}| \epsilon_2 - \boldsymbol{b}_1^{(1)}}{\boldsymbol{W}_{1,1}^{(1)}} = \frac{-(-7)(-1/2) + |-7|(3/2) - 6}{1} = 1 \tag{37a}$$

$$\boldsymbol{x}_2^{(clip)} = \frac{-\boldsymbol{W}_{1,1}^{(1)} \hat{\boldsymbol{x}}_1 + |\boldsymbol{W}_{1,1}^{(1)}| \epsilon_1 - \boldsymbol{b}_1^{(1)}}{\boldsymbol{W}_{1,2}^{(1)}} = \frac{-(1)(1/2) + |1|(3/2) - 6}{-7} = \frac{5}{7} \tag{37b}$$

It is indeed the case that $\boldsymbol{x}_1^{(clip)} < \overline{\boldsymbol{x}}_1$ and $\boldsymbol{x}_2^{(clip)} > \underline{\boldsymbol{x}}_2$, so the limits of the refined input domain are now,

$$\underline{\boldsymbol{x}}^{(rc)} = [-1, 5/7]^\top, \qquad \overline{\boldsymbol{x}}^{(rc)} = [1, 1]^\top. \tag{38}$$

The new box representation may also be characterized by its center and half-widths:

$$\hat{\boldsymbol{x}}^{(rc)} = (\overline{\boldsymbol{x}}^{(rc)} + \underline{\boldsymbol{x}}^{(rc)})/2, \qquad \epsilon^{(rc)} = (\overline{\boldsymbol{x}}^{(rc)} - \underline{\boldsymbol{x}}^{(rc)})/2, \tag{39}$$

yielding the refined input domain

$$\mathcal{X}^{(rc)} = \left\{ \boldsymbol{x} \mid \|\hat{\boldsymbol{x}}^{(rc)} - \boldsymbol{x}\|_\infty \leq \epsilon^{(rc)} \right\} = \left\{ \boldsymbol{x} \mid \underline{\boldsymbol{x}}^{(rc)} \leq \boldsymbol{x} \leq \overline{\boldsymbol{x}}^{(rc)} \right\}. \tag{40}$$

We now proceed to bound the network using the CROWN algorithm under $\mathcal{X}^{(rc)}$ in place of the original domain, $\mathcal{X}$. Substituting this domain into Eq. (24) produces $\underline{\boldsymbol{z}}^{(rc)} = [-2, -13]^\top$ and $\overline{\boldsymbol{z}}^{(rc)} = [2, -57/21]^\top$. Both neurons exhibit tighter bounds, and notably, $\overline{\boldsymbol{z}}_2^{(rc)} < 0$, which ensures that the input to the second ReLU neuron is strictly non-positive across $\mathcal{X}^{(rc)}$. Since the ReLU function is piecewise linear, this implies an *exact* post-activation bound:

$$\underline{\sigma}(\boldsymbol{z}) = \sigma(\boldsymbol{z}) = \overline{\sigma}(\boldsymbol{z}) = 0, \qquad \forall \boldsymbol{x} \in \mathcal{X}^{(rc)}. \tag{41}$$

One interesting observation to point out is that with this clipped domain, $\overline{\boldsymbol{z}}_1^{(rc)} = 2$ improves dramatically from the original upper bound, $\overline{\boldsymbol{z}}_1 = 22$. However, because our constraint originated

from $\boldsymbol{z}_1 \leq 0$, the true maximum is in fact zero. In this case, one could enforce this upper bound to be zero, but keep in mind that relaxed clipping algorithm is designed for generality, thus constraints may arise from split decisions at any neuron in the network where forcing $\overline{\boldsymbol{z}}_1^{(rc)} = 0$ would not be valid. This example illustrates that relaxed clipping, while lightweight and compatible with CROWN, remains an approximation and may yield suboptimal yet informative bounds. Rather than enforcing the true maximum, we retain the relaxed clipping result, $\overline{\boldsymbol{z}}_1^{(rc)} = 2$.

Applying Lemma A.1, we derive the linear post-activation bounds with respect to the pre-activation inputs:

$$\underline{\boldsymbol{D}}^{(rc)} = \begin{bmatrix} \alpha_1 & 0 \\ 0 & 0 \end{bmatrix}, \qquad \underline{\boldsymbol{b}}^{(rc)} = \begin{bmatrix} 0 \\ 0 \end{bmatrix} \tag{42a}$$

$$\overline{\boldsymbol{D}}^{(rc)} = \begin{bmatrix} \frac{\overline{\boldsymbol{z}}_1^{(rc)}}{\overline{\boldsymbol{z}}_1^{(rc)} - \underline{\boldsymbol{z}}_1^{(rc)}} & 0 \\ 0 & 0 \end{bmatrix}, \qquad \overline{\boldsymbol{b}}^{(rc)} = \begin{bmatrix} \frac{-\overline{\boldsymbol{z}}_1^{(rc)} \underline{\boldsymbol{z}}_1^{(rc)}}{\overline{\boldsymbol{z}}_1^{(rc)} - \underline{\boldsymbol{z}}_1^{(rc)}} \\ 0 \end{bmatrix}. \tag{42b}$$

For simplicity, let $\alpha_1 = 1$. Notice that for the second neuron, we have that,

$$(\underline{\boldsymbol{D}}_{2,:}^{(rc)})^\top \boldsymbol{z} + \underline{\boldsymbol{b}}_2^{(rc)} = \sigma(\boldsymbol{z})_2 = (\overline{\boldsymbol{D}}_{2,:}^{(rc)})^\top \boldsymbol{z} + \overline{\boldsymbol{b}}_2^{(rc)} = 0, \qquad \forall \boldsymbol{z}_2 \in [\underline{\boldsymbol{z}}_2^{(rc)}, \overline{\boldsymbol{z}}_2^{(rc)}], \tag{43}$$

confirming that this ReLU neuron is inactive and its output can be exactly bounded as zero.

Finally, performing the same backpropagation procedure as before yields

$$\min_{\boldsymbol{x} \in \mathcal{X}} f(\boldsymbol{x}) \geq \min_{\boldsymbol{x} \in \mathcal{X}^{(rc)}} \underline{\boldsymbol{a}}^{(rc)\top} \boldsymbol{x} + \underline{\boldsymbol{c}}^{(rc)} = \underline{\boldsymbol{a}}^{(rc)\top} \hat{\boldsymbol{x}}^{(rc)} - |\underline{\boldsymbol{a}}^{(rc)}|^\top \boldsymbol{\epsilon}^{(rc)} + \underline{\boldsymbol{c}}^{(rc)} = -2. \tag{44}$$

This represents an improvement over the previous bound, although the property remains unverified.

**Complete Clipping.** Relaxed clipping can be viewed as an *indirect* approach towards refining the bounds of the neural network as a smaller input representation can potentially yield improvement in the concretization step when forming the neurons' bounds. *Complete clipping* on the other hand is a *direct* approach towards refining the bounds on the network as the optimization objective specifically targets the neuron's bounds rather than the shared input representation.

In practice, relaxed clipping is extremely lightweight, and its operations are well-suited for GPUs. Thus, it is often sensible to combine relaxed and complete clipping in the verification pipeline. In this example, however, we aim to clearly distinguish the two methods. Therefore, we use the original input domain, $\mathcal{X}$, rather than the refined one, $\mathcal{X}^{(rc)}$, and retain the ReLU split constraint $\boldsymbol{z}_1 \leq 0$.

Complete clipping operates directly on the bounds of each neuron, refining them via constrained optimization. The first step is to bound the preactivation bounds which may be formulated as:

$$\underline{\boldsymbol{z}}_i^{(cc)} = \min_{\boldsymbol{x} \in \mathcal{X}} \boldsymbol{W}_{i,:}^{(1)} \boldsymbol{x} + \boldsymbol{b}_i^{(1)} \qquad\qquad \overline{\boldsymbol{z}}_i^{(cc)} = \max_{\boldsymbol{x} \in \mathcal{X}} \boldsymbol{W}_{i,:}^{(1)} \boldsymbol{x} + \boldsymbol{b}_i^{(1)}$$

$$\text{s.t. } \left( \boldsymbol{W}_{1,:}^{(1)} \right)^\top \boldsymbol{x} + \boldsymbol{b}_1^{(1)} \leq 0 \quad (45a) \qquad\qquad \text{s.t. } \left( \boldsymbol{W}_{1,:}^{(1)} \right)^\top \boldsymbol{x} + \boldsymbol{b}_1^{(1)} \leq 0 \quad (45b)$$

For this two-dimensional problem, this amounts to targeting the lower and upper bound of both neurons at the intermediate layer, resulting in a total of four constrained optimization subproblems. However, we will only focus on refining the upper bounds as we will soon see that this will be sufficient for verifying our desired property in this toy example.

As discussed earlier, the split constraint on the first neuron implies that its true upper bound is trivially $\overline{\boldsymbol{z}}_1^\star = 0$. With a single constraint, Theorem 3.1 guarantees optimality, i.e., $\overline{\boldsymbol{z}}_1^{(cc)} = \overline{\boldsymbol{z}}_1^\star = 0$, which can be verified using Algorithm 1[2]. For a single constraint, complete clipping attains the exact optimal solution through enumeration, without relying on projected gradient methods or LP solvers, no longer serving as an approximation such as the case with relaxed clipping. Even when multiple constraints are present (where optimality is not guaranteed), this targeted refinement remains a powerful mechanism for improving intermediate-layer bounds.

---

[2]Algorithm 1 performs coordinate ascent on the dual objective when the primal is a *minimization* problem. Since (45b) is a *maximization* problem, we can negate the primal objective to minimize it, then negate the resulting solution. Furthermore, the algorithm assumes constraints of the form $\boldsymbol{Gx} + \boldsymbol{h} \leq \boldsymbol{0}$. For constraints of the opposite form, $\boldsymbol{Gx} + \boldsymbol{h} \geq \boldsymbol{0}$, one may simply negate them. Hence, the algorithm is used without loss of generality.

Next, we consider the upper bound $\overline{z}_2^{(cc)}$. Complete clipping yields the optimal solution $\overline{z}_2^\star$, which we derive analytically via Algorithm 1. According to Theorem 3.1, the dual form of this problem and its solution is given by

$$L^\star = \min_{\beta \in \mathbb{R}+} \left( \left( \boldsymbol{W}_{2,:}^{(1)} - \beta \boldsymbol{W}_{1,:}^{(1)} \right)^\top \hat{\boldsymbol{x}} + \sum_{j=1}^n \left| \boldsymbol{W}_{2,j}^{(1)} - \beta \boldsymbol{W}_{1,j}^{(1)} \right| \boldsymbol{\epsilon}_j + \boldsymbol{b}_2^{(1)} - \beta \boldsymbol{b}_1^{(1)} \right). \qquad (46)$$

The dual objective is minimized with respect to the Lagrange multiplier, $\beta \in \mathbb{R}+$. One possible solution occurs at $\beta = 0$, but since relaxed clipping already improved this bound, we expect that $\beta^\star \neq 0$.

Before solving for $\beta^\star$, it is helpful to examine the structure of the dual objective $D(\beta)$. Because the primal objective maximizes $\boldsymbol{z}_2$, minimizing the dual objective is equivalent. The function $D(\beta)$ is *convex* and *piece-wise linear*, so we analyze its sub-gradient, $\frac{\partial}{\partial \beta} D(\beta)$:

$$\begin{cases} \left( -\boldsymbol{W}_{1,:}^{(1)} \right)^\top \hat{\boldsymbol{x}} + \sum_{j=1}^2 \left\{ \text{sign} \left( \boldsymbol{W}_{2,j}^{(1)} - \beta \boldsymbol{W}_{1,j}^{(1)} \right) (-\boldsymbol{W}_{1,j}^{(1)}) \boldsymbol{\epsilon}_j \right\} - \boldsymbol{b}_1^{(1)} & , \beta \notin \boldsymbol{q} \\ \left[ \frac{\partial}{\partial \beta} D(\beta^-), \frac{\partial}{\partial \beta} D(\beta^+) \right] & , \beta \in \boldsymbol{q} \end{cases} \qquad (47)$$

where $\boldsymbol{q}$ is the vector of breakpoints $[(\boldsymbol{W}_{2,j}^{(1)})/(\boldsymbol{W}_{1,j}^{(1)})]_{j=1}^2 = [5, 1/7]$. Thus, the gradient is uniquely defined on the intervals $\beta \in (-\infty, 1/7)$, $(1/7, 5)$, and $(5, \infty)$. Because $D(\beta)$ is convex, its sub-gradient is negative on the leftmost interval and positive on the rightmost one. The optimal $\beta^\star$ occurs at the break-point where the sub-gradient changes sign from negative to positive, i.e., the break-point whose sub-gradient interval contains zero. The sub-gradients in each region are:

$$\begin{cases} -|\boldsymbol{W}_{1,1}^{(1)}|\boldsymbol{\epsilon_1} - |\boldsymbol{W}_{1,2}^{(1)}|\boldsymbol{\epsilon_2} - \left( \boldsymbol{W}_{1,:}^{(1)} \right)^\top \hat{\boldsymbol{x}} - \boldsymbol{b}_1^{(1)} = -24 & , \beta < \frac{1}{7} \\ -|\boldsymbol{W}_{1,1}^{(1)}|\boldsymbol{\epsilon_1} + |\boldsymbol{W}_{1,2}^{(1)}|\boldsymbol{\epsilon_2} - \left( \boldsymbol{W}_{1,:}^{(1)} \right)^\top \hat{\boldsymbol{x}} - \boldsymbol{b}_1^{(1)} = -1 & , \frac{1}{7} < \beta < 5 \\ +|\boldsymbol{W}_{1,1}^{(1)}|\boldsymbol{\epsilon_1} + |\boldsymbol{W}_{1,2}^{(1)}|\boldsymbol{\epsilon_2} - \left( \boldsymbol{W}_{1,:}^{(1)} \right)^\top \hat{\boldsymbol{x}} - \boldsymbol{b}_1^{(1)} = 2 & , \beta > 5. \end{cases} \qquad (48)$$

Note that when transitioning from the case where $\beta \in (-\infty, 1/7)$ to $\beta \in (1/7, 5)$, the term $|\boldsymbol{W}_{1,2}^{(1)}|\boldsymbol{\epsilon_2}$ switches sign while $-|\boldsymbol{W}_{1,1}^{(1)}|\boldsymbol{\epsilon_1}$ remains negative. This ordering follows because in our break-point vector, we have that $\boldsymbol{q}_2 < \boldsymbol{q}_1$, and this ensures the sub-gradient is correctly calculated in each sub-interval, providing intuition as to why argsort$(\boldsymbol{q})$ is necessary in Algorithm 1. For this example, the sign change occurs at $\beta = 5$, giving the minimum of the dual objective at $\beta^\star = 5$ and the solution $\overline{z}^{(cc)} = \overline{z}_2^\star = L^\star = -3$.

After performing complete clipping at the intermediate layer, we discover that the optimal upper bounds under our split constraint are given as $\overline{\boldsymbol{z}}^\star = [0, -3]^\top$. There is clearly an inter-neuron dependency between the two ReLU neurons such that forcing the first neuron to be in-active subsequently causes the the second neuron to also become in-active. Consequently, the post-activation neurons can be *exactly* bounded using the CROWN algorithm, and in particular, $\sigma(\boldsymbol{z}) = \boldsymbol{0}$ for all $\boldsymbol{z} \in [\underline{\boldsymbol{z}}^\star, \overline{\boldsymbol{z}}^\star]$. Since the last layer contains no bias vector, it is not necessary to perform complete clipping again, and we have finally verified our desired property for this subproblem,

$$\min_{\boldsymbol{x} \in \mathcal{X}} f(\boldsymbol{x}) = \boldsymbol{w}^{(2)\top} \sigma(\boldsymbol{z}) \geq 0, \qquad \text{subject to } \boldsymbol{z}_1 \leq 0. \qquad (49)$$

# D  Experiments

## D.1  Experiments Settings

To allow for comparability of results, all tools for input BaB were evaluated on equal-cost hardware with a 32-vcore CPU, one NVIDIA RTX 4090 GPU with 24 GB memory, and 256 GB CPU memory. For ReLU based BaB experiment, we use a cluster with one AMD EPYC 9534 64-core CPU and the GPU is one NVIDIA RTX 5090 GPU with 32 GB memory and 512 GB CPU memory. Our implementation is based on the open-source $\alpha,\beta$-CROWN verifier[3] with Clip-and-Verify related

---

[3]`https://github.com/huanzhang12/alpha-beta-CROWN`

code added. For input bab, three different set-ups of Clip-and-Verify are tested: Relaxed, Relaxed + Reorder, and Complete. Here Relaxed, Reorder and Complete refers to the methodology discussed in 3.2 and 3.3. All experiments use 32 CPU cores and 1 GPU. The MIP cuts are acquired by the cplex [34] solver (version 22.1.0.0). We use the Adam optimizer [38] to solve both $\alpha, \beta, \mu, \tau$. For the SDP-FO benchmarks, we optimize those parameters for 20 iterations with a learning rate of 0.1 for $\alpha$ and 0.02 for $\beta, \mu, \tau$. We decay the learning rates with a factor of 0.98 per iteration. The timeout is 200s per instance. For the VNN-COMP benchmarks, we use the same configuration as $\alpha,\beta$-CROWN used in the respective competition and the same timeouts. For NN control systems, task details are in [42, Appendix C]. For the Lyapunov function level set, we verify on $V(x) \in [0.2, 0.20001]$ for CartPole and Quadrotor-2D, and on $V(x) \in [2.0, 2.1]$ for Quadrotor-2D-Large-ROA. Let $\Omega_{\text{final}}$ denote the resulting (expanded) box used for training/evaluation once the generator stabilizes. Concretely (all "$\pm[\cdot]$" are per-coordinate half-widths), for CartPole, $\Omega_{\text{final}} = \pm[4.8, 3.6, 13.2, 13.2]$. For Quadrotor-2D, $\Omega_{\text{final}} = \pm[12, 13.2, 12, 19.2, 20.4, 88.8]$.

## D.2 Ablation Studies

We conduct a detailed ablation study on multiple adversarially-trained models spanning MNIST, CIFAR, and larger VNN-COMP benchmarks to evaluate Clip-and-Verify and its variants against three baselines: $\beta$-CROWN, GCP-CROWN, and BICCOS. Tables 4 and 6 highlight three key metrics: verified accuracy (Ver.%), average per-example verification time (Time), and average per-verified-example domain visited (D.V.). Domain visited (D.V.) is a metric specific to branch-and-bound (BaB) methods, indicating how many subproblems (domains) are explored to fully verify an instance. Crucially, a higher D.V. count may reflect verifying more difficult instances or a larger overall coverage, rather than inefficiency in the verification process. We also provide Figures 5 and 6 that visualize these metrics across all benchmarks.

**Overall Verified Accuracy and Time** From Table 4, Clip-and-Verify variants nearly always achieve higher verified accuracy than the baselines. For instance, on CNN-B-Adv (CIFAR), Clip-and-Verify with BICCOS reaches 51.5% verified accuracy—surpassing the 47.0% ($\beta$-CROWN), 49.5% (GCP-CROWN with MIP cuts), and 51.0% (BICCOS alone) of the baselines. On cifar10-resnet, Clip-and-Verify (with MIP cuts or with BICCOS) achieves up to 88.89% verified accuracy (outperforming the 83.33% - 87.5% range from baselines). In terms of verification time on this benchmark, Clip-and-Verify with $\beta$-CROWN is the fastest overall (6.06s), and Clip-and-Verify with BICCOS (11.80s) remains competitive with standalone BICCOS (16.73s) and GCP-CROWN (17.99s). When scaling to deeper networks such as cifar100-2024 and tinyimagenet-2024, Clip-and-Verify with BICCOS maintains the leading verified accuracy (65.5% and 72.0%, respectively) while sustaining moderate average verification times (e.g., 8.17s for cifar100-2024 and 10.48s for tinyimagenet-2024), underscoring its suitability for larger-scale verification tasks.

**Domain Visited (D.V.) vs. Difficulty** In Table 6, we further examine the average domain visited (D.V.) across verified examples. While Clip-and-Verify variants may sometimes visit more domains (e.g., on oval22, Clip-and-Verify with MIP cuts visits 18891.25 domains compared to 16614.72 for GCP-CROWN with MIP cuts), this often correlates with achieving higher verified accuracy (90.00% vs 83.33% in this case). This suggests that the method is effectively exploring the space to verify more challenging instances or a broader set of inputs, leading to a net increase in verified accuracy. For example, on CNN-A-Adv (CIFAR), Clip-and-Verify with MIP cuts visits 3704.86 domains on average and attains 48.5% verified accuracy. While its D.V. is slightly higher than standalone BICCOS (3622.71 D.V. for 48.5% accuracy), it's notably lower than $\beta$-CROWN (12621.38 D.V. for 45.5% accuracy) and GCP-CROWN with MIP cuts (8186.11 D.V. for 48.5% accuracy), while achieving comparable or better accuracy. In other scenarios (e.g., CNN-A-Adv-4 on CIFAR), Clip-and-Verify with BICCOS achieves 48.5% accuracy with a D.V. of only 843.47. This is the same or higher accuracy with a significantly smaller D.V. compared to standalone $\beta$-CROWN (46.5%, 2066.39 D.V.), GCP-CROWN with MIP cuts (48.5%, 3907.30 D.V.), and BICCOS (48.5%, 1319.56 D.V.). This illustrates that when Clip-and-Verify effectively prunes the search space, verification efficiency can improve even while tackling similarly challenging problems and achieving high accuracy.

**A Breakdown Comparison between Verifiers** Table 5 exhibits the instance-wise comparison on acasxu benchmark, as also illustrated in Figure 2a. Most of the instances in acasxu benchmark are easy to verify. On these instances, all the verifiers share similar verification time and D.V. . Instances

Table 5: Instance-wise breakdown comparison on `acasxu` benchmark between $\alpha, \beta$-CROWN and Clip-and-Verify.

| Method | Avg. on simpler instances | | 65 | | 73 | |
|---|---|---|---|---|---|---|
| | Time | D.V. | Time | D.V. | Time | D.V. |
| $\alpha, \beta$-CROWN | 1.0287 | 12615.35 | timeout | - | 19.1899 | 5474527 |
| relaxed | 1.0087 | 6759.11 | 9.1839 | 1876495 | 2.7422 | 291855 |
| relaxed + reorder | 1.0283 | 6467.96 | 7.6514 | 1416479 | 2.6385 | 229755 |
| complete | 1.0991 | 6350.11 | 6.8963 | 531381 | 3.5677 | 112431 |

Table 6: Ablation Studies on Verified accuracy (Var.%), avg. per-verified-example domain visited number (D.V.) analysis for all method verified instances on different Clip-and-Verify components.

| Dataset | Model | $\beta$-CROWN | | GCP-CROWN with MIP cuts | | BICCOS | | Clip-and-Verify with $\beta$-CROWN | | Clip-and-Verify with MIP cuts | | Clip-and-Verify with BICCOS | | Upper bound |
|---|---|---|---|---|---|---|---|---|---|---|---|---|---|---|
| $\epsilon = 0.3$ and $\epsilon = 2/255$ | | Ver.% | D.V. | Ver.% | D.V. | Ver.% | D.V. | Ver.% | D.V. | Ver.% | D.V. | Ver.% | D.V. | bound |
| MNIST | CNN-A-Adv | 71.0 | 2712.72 | 71.5 | 4447.54 | **76.0** | 3081.50 | 74.0 | 2395.96 | 73.5 | 1495.07 | **76.0** | 2636.99 | 76.5 |
| CIFAR | CNN-A-Adv | 45.5 | 12621.38 | **48.5** | 8186.11 | **48.5** | 3622.71 | 45.5 | 1037.36 | **48.5** | 3704.86 | **48.5** | 2073.58 | 50.0 |
| | CNN-A-Adv-4 | 46.5 | 2066.39 | **48.5** | 3907.30 | **48.5** | 1319.56 | 46.5 | 298.89 | **48.5** | 2396.36 | **48.5** | 843.47 | 49.5 |
| | CNN-A-Mix | 42.0 | 6108.57 | 47.5 | 17609.84 | **48.0** | 8015.12 | 43.0 | 4462.48 | 47.5 | 9836.06 | **48.0** | 4189.92 | 53.0 |
| | CNN-A-Mix-4 | 51.0 | 482.43 | 55.0 | 8922.50 | 56.0 | 3319.90 | 51.0 | 150.46 | 55.0 | 4304.32 | **56.5** | 3501.06 | 57.5 |
| | CNN-B-Adv | 47.0 | 7255.68 | 49.5 | 9846.32 | 51.0 | 5758.00 | 49 | 4951.07 | **51.5** | 6979.27 | **51.5** | 2677.94 | 65.0 |
| | CNN-B-Adv-4 | 55.0 | 1776.66 | 58.5 | 4688.22 | 59.5 | 2711.15 | 56.5 | 649.92 | 60.0 | 3565.74 | **60.5** | 1095.30 | 63.5 |
| cifar10-resnet | | 83.33 | 2105.76 | 87.5 | 7091.30 | 87.5 | 5428.60 | 86.11 | 478.0 | **88.89** | 2545.28 | **88.89** | 2643.15 | 100.0 |
| oval22 | | 66.66 | 30949.95 | 83.33 | 16614.72 | 83.33 | 12730.08 | 73.33 | 20191.27 | **90.00** | 18891.25 | **90.00** | 14032.81 | 96.67 |
| cifar100-2024 | | 59.5 | 1535.60 | - | - | 60.5 | 769.87 | 63.0 | 152.96 | - | - | **65.5** | 122.00 | 84.0 |
| tinyimagenet-2024 | | 67.5 | 830.92 | - | - | 69.0 | 497.52 | 70.0 | 188.02 | - | - | **72.0** | 118.34 | 78.5 |
| vision-transformer 2024 [53] | | 59.0 | 149.75 | - | - | - | - | **61.0** | 84.84 | - | - | - | - | 100.0 |

73 and 65 stand out as they're significantly harder to verify. For instance 73, clipping is able to cut down over 94% D.V. and over 80% verification time. While $\alpha, \beta$-CROWN is unable to verify instance 65 within given timeout, Clip-and-Verify is able to verify it within 10 seconds. Cactus plots Figures 5 and 6 are also able to give a instance-wise comparison. Please see the following paragraph for a more detailed explanation.

**Interpretation of the Cactus Plot** Figures 5 and 6 illustrate the number of instances verified as runtime varies. Each line represents a method, with the x-axis showing the cumulative verified instances under a given timeout (y-axis). This style, used in VNN-COMP reports captures the trade-off between subproblem difficulty and runtime performance. The curve's right end indicates total solved instances—further right means more instances verified within the timeout. A flat initial curve fragment reflects many easy instances solved quickly. A curve below and to the right of another shows consistently faster solving. The x-axis ordering reflects instance difficulty, from easy (left) to hard (right). For example, on the tinyimagenet benchmark (Figure 5d), our Complete Clipping + BICCOS variant solves easy instances faster and solves more hard instances. Across these benchmarks, the proposed Clip-and-Verify framework demonstrates a favorable balance: it reduces the number of hard subproblems, leading to more instances being verified within moderate time thresholds. This suggests that although our clipping procedures incur additional overhead that may be noticeable in easy instances, the overall net gain is highlighted in its ability to solve more instances and hard instances with shorter verification times.

Overall, these results show that Clip-and-Verify's framework of enhanced linear bounding and "clipping" robustly scales across varying network depths and adversarial training schemes. The additional integration of MIP cuts or BICCOS bounding routines consistently pushes verified accuracy closer to each benchmark's upper bound while balancing verification time and domain exploration. We plot and analyze these ablation studies across all benchmarks in Appendix D.2, confirming that our method not only raises coverage (Ver.%) but also leverages clipping and cutting plane methods to verify some of the hardest instances encountered.

### D.3 Detailed Comparison with LP Solvers

To validate our claim that coordinate ascent is more efficient than LP solvers for our task, we integrated LP solvers directly into our BaB algorithm 4 to solve the clipping optimization problem. We compared our method ("Clip (ours)") against the Gurobi solver using dual simplex with varying

iteration limits. The experiment was run on the `acasxu` benchmark instance 65, and we report the average time and bound error over 30 million LP calls during the entire BaB process. We report only Gurobi's `optimize()` time, ignoring all Python overhead for problem creation, giving an advantage to the LP solver.

As shown in Table 7, our method is **740x faster** than a 1-iteration LP heuristic and **880x faster** than a 10-iteration LP, while achieving comparable accuracy (0.00085 vs. 0.0007 mean error). The LP solver only achieves near-zero error with 10+ iterations, at which point it is intractably slow. The high fixed cost of LP solvers (even for 1 iteration) comes from presolve and initial basis factorization routines, which are far more expensive than our simple $\mathcal{O}(n \log n)$ sort.

Table 7: Comparison of our coordinate ascent ("Clip") vs. Gurobi dual simplex with iteration limits on `acasxu` (instance 65). Time and error are averaged over 30M calls.

| Method | Avg. Time per Call | avg. Bounds Error | std. Bounds Error |
|---|---|---|---|
| LP-simplex (1 iter) | 2.08s | 0.401 | 1.45 |
| LP-simplex (10 iter) | 2.47s | 0.0007 | 0.0018 |
| LP-simplex (20 iter) | 2.49s | 0.0 | 0.0 |
| **Clip (ours)** | **0.0028s** | **0.00085** | **0.0019** |
| LP-full (ground truth) | 2.50s | 0.0 | 0.0 |

We also investigated modern GPU-based LP solvers, specifically Google's PDLP [3, 4], which uses a Primal-Dual Hybrid Gradient (PDHG) algorithm. As shown in Table 8, while PDLP achieves high accuracy when it converges, it is not suitable for our use case. PDHG does not maintain a feasible solution during iterations. Under strict iteration limits, it has a very high failure rate, returning a `NOT_SOLVED` status and thus **failing to provide a valid dual bound** for our sound verification procedure. In contrast, dual simplex (Gurobi) and our method always return a valid, feasible bound. PDLP is designed for single, massive LPs, whereas our framework requires solving millions of independent, small LPs in parallel, a setting where our specialized GPU solver excels.

Table 8: Comparison with Google's PDLP. Our method is dramatically faster and, critically, always returns a valid bound, unlike PDLP which frequently fails under iteration limits. B.E. means Bound Error.

| Method | Avg. Time per Call | avg. B.E | std. B.E | Failure Rate (%) |
|---|---|---|---|---|
| PDLP (1 iter) | 2.23s | 2e-8 | 4e-8 | 81.31% |
| PDLP (10 iter) | 4.21s | 1.4e-8 | 2.3e-8 | 48.26% |
| PDLP (20 iter) | 6.02s | 5.5e-8 | 1e-8 | 38.38% |
| **Clip (ours)** | **0.0028s** | **0.00085** | **0.0019** | **0.0%** |
| Gurobi-full | 2.50s | 0.0 | 0.0 | 0.0% |

### D.4 Details of Heuristics

**Neuron Selection Heuristic.** The performance of Complete Clipping can be sensitive to the choice of which intermediate neurons to refine. To guide this selection, we employ a heuristic based on the Branch-and-Bound for Split Recommendation (BaBSR) score, originally designed to select which neuron to branch on during BaB [13]. Specifically, we use the intercept score from BaBSR, which estimates the potential impact of refining a neuron's bounds on the final output relaxation. The score for a neuron $k$ is calculated as:

$$\text{score}_k = \frac{\max(0, -l_k) \cdot \max(0, u_k)}{u_k - l_k} \cdot \max(0, -\text{mean}_{\underline{A}_k})$$

where $l_k$ and $u_k$ are the neuron's pre-activation lower and upper bounds, and $\text{mean}_{\underline{A}_k}$ relates to the intercept of the neuron's linear lower bound. A higher score indicates a looser relaxation and a greater potential for improvement. For each layer, we compute this score for all unstable neurons and select the top-$k$ neurons for refinement with Complete Clipping.

To validate this choice, we conducted an ablation study on the `tinyimagenet-2024` and `cifar100-2024` benchmarks, comparing BaBSR against three alternative heuristics: random selection, prioritizing neurons with the largest bound gap $(U - L)$, and prioritizing neurons with the largest bound product $(-U \times L)$. As shown in Tables 9 and 10, the BaBSR heuristic consistently achieves the best verification time and explores the fewest subdomains, especially for the balanced top-20 setting. This confirms that BaBSR makes more effective choices, leading to earlier pruning and a more efficient search.

Table 9: Average verification time (s) per instance for different top-k heuristics.

| Benchmark | Top-k Setting | Random | -U × L | U-L | BaBSR |
|---|---|---|---|---|---|
| tinyimagenet | Top-10 per layer | 16.3 | 16.5 | 16.8 | **16.2** |
| | Top-20 per layer | 18.1 | 17.9 | 17.2 | **15.1** |
| | All neurons | 22.4 | 22.4 | 22.3 | 22.7 |
| cifar100 | Top-10 per layer | 26.2 | 26.6 | 27.0 | **26.1** |
| | Top-20 per layer | 29.1 | 28.8 | 27.7 | **24.3** |
| | All neurons | 36.5 | 36.4 | 36.4 | 36.5 |

Table 10: Average number of visited domains per instance for different top-k heuristics.

| Benchmark | Top-k Setting | Random | -U × L | U-L | BaBSR |
|---|---|---|---|---|---|
| tinyimagenet | Top-10 per layer | 485 | 453 | 420 | **388** |
| | Top-20 per layer | 381 | 370 | 365 | **342** |
| | All neurons | 278 | 278 | 278 | 278 |
| cifar100 | Top-10 per layer | 907 | 847 | 785 | **726** |
| | Top-20 per layer | 712 | 692 | 682 | **640** |
| | All neurons | 521 | 521 | 521 | 521 |

Table 11: Ablation Study on Top-k Neuron Selection on `cifar_cnn_a_mix`

| Top-k | # Verified | Avg. # Domain Visited | Avg. Time (s) |
|---|---|---|---|
| 0 (reduce to $\beta$-CROWN) | 84 | 6108.57 | 4.20 |
| 20 | 86 | 4462.47 | 4.24 |
| 50 | 86 | 4372.79 | 4.83 |
| all | 85 (1 timeout) | 1881.77 | 8.23 |

Table 11 shows the ablation study on the top-k neuron selection heuristic on `cifar_cnn_a_mix` benchmark, reveals that strategically prioritizing neurons based on their FSB intercept scores significantly enhances verification efficiency. Employing a moderate $k$ (specifically top-k 20 and 50) leads to the best outcomes, successfully verifying more properties (86) while substantially reducing both the number of domains visited (by up to 28% compared to no heuristic) and the overall time (by up to 19%). In contrast, not using the heuristic (top-k 0) results in a less efficient search, while applying it to all unstable neurons (top-k "all") drastically cuts down visited domains but incurs a prohibitive time cost and a timeout, indicating that the overhead of processing too many prioritized neurons outweighs the benefits of more targeted branching. This demonstrates a crucial trade-off, with intermediate k values striking an optimal balance between guided search and computational overhead.

**Constraint Importance Heuristic** The constraint importance heuristic is to sort the constraints for better performance. The constraints corresponding to hyperplanes closer to the center of the current input domain might be more immediately relevant or impactful for tightening the bounds or for cutting off a significant portion of the current feasible region $\mathcal{X}$.

First, for the current input box domain $\mathcal{X}$, determine its centroid $\hat{x}$. We can calculate $g^\top \hat{x} + h - \sum_{i=1}^{n} |g_i| \epsilon_i > 0$ (there is no $x \in \mathcal{X}$ satisfy the constraint in (2)) to check the infeasibility and

Table 12: Ablation Study on Top-k Neuron Selection on `lsnc`

| Ratio of selected neurons | Total time (s) | Domains visited |
|---|---|---|
| 0/6 | 13.37 | 18,073,174 |
| 1/6 | 11.25 | 7,254,214 |
| 2/6 | 10.54 | 4,914,243 |
| 3/6 | 9.70 | 3,072,863 |
| 4/6 | 9.23 | 2,088,198 |
| 5/6 | 8.69 | 1,739,412 |
| 6/6 | 7.66 | 871,14 |

$\boldsymbol{g}^\top \hat{\boldsymbol{x}} + h + \sum_{i=1}^{n} |g_i|\epsilon_i \leq 0$ (for all $x \in \mathcal{X}$ satisfy the constraint in (2)) to remove redundancy. Second, for each available linear constraint $\boldsymbol{g}^\top \boldsymbol{x} + h = 0$, calculate the geometric distance from the centroid $\hat{\boldsymbol{x}}$ to this hyperplane:

$$ d = \frac{|\boldsymbol{g}^\top \boldsymbol{x}_0 + h|}{\|\boldsymbol{g}\|_2} $$

Then, sort the constraints in ascending order based on these calculated distances to prioritize constraints that are closer to the centroid. In methods like Algorithm 1 (Complete Clipping), processing more impactful constraints earlier might lead to faster convergence or more significant bound improvements in the coordinate ascent. Such an important metric could also be relevant to Algorithm 3 (Relaxed Clipping).

We tested the bound tightness improvement from the heuristic on `acasxu` instance 65. Clip-and-Verify improves the intermediate bounds for 470,772,542 times during input BaB, and 3.962% of them can be further tightened with constraint importance heuristic. For all these further improved bounds, we computed the empirical quantiles of the relative improvements, shown in Table D.4.

| Percentile of problems | Max | 0.1th | 1th | 5th | 10th | 25th | 50th | 75th | 90th |
|---|---|---|---|---|---|---|---|---|---|
| Bound Improvement | 4242.6% | 270.79% | 3.07% | 0.64% | 0.29% | 0.10% | 0.03% | 0.01% | 0.002% |

These results reveal that the head of the distribution contains substantial refinements, with the maximum observed improvement reaching up to 4000%. This indicates that the heuristic can produce tightened bounds, potentially leading to earlier branch pruning or faster convergence.

# E   Limitation and Broader Impacts

**Limitation.** Our framework's effectiveness is influenced by the trade-off between the precision of clipping and its computational overhead. While **Complete Clipping** utilizes GPU acceleration, its scalability can be impacted when networks have very large hidden layers combined with a high number of unstable neurons and sparse constraints. The memory complexity scales as $\mathcal{O}(B \times N \times M)$, where $B$ is the number of BaB subproblems, $N$ is the number of neurons in the largest layer, and $M$ is the number of linear constraints. To mitigate this, our top-k neuron selection heuristic (Appendix D.3) strategically prioritizes the most critical neurons, balancing precision and cost.

Conversely, while **Relaxed Clipping** is highly efficient with a complexity of $\mathcal{O}(n)$ per input dimension, its effectiveness can diminish in very high-dimensional input spaces due to the curse of dimensionality, where axis-aligned relaxations may become looser.

However, the ultimate scalability of our method is not fundamentally limited by input dimensionality or layer width, but rather by the intrinsic difficulty of the verification query, which is an NP-hard problem. As demonstrated by our experiments on neural network control systems 2 even small networks with low-dimensional inputs can pose immense verification challenges. Clip-and-Verify proved essential in solving these hard instances. The key insight is that our approach adapts computational effort based on problem characteristics rather than being constrained by absolute dimensional limits, ensuring **practical applicability** across diverse verification scenarios.

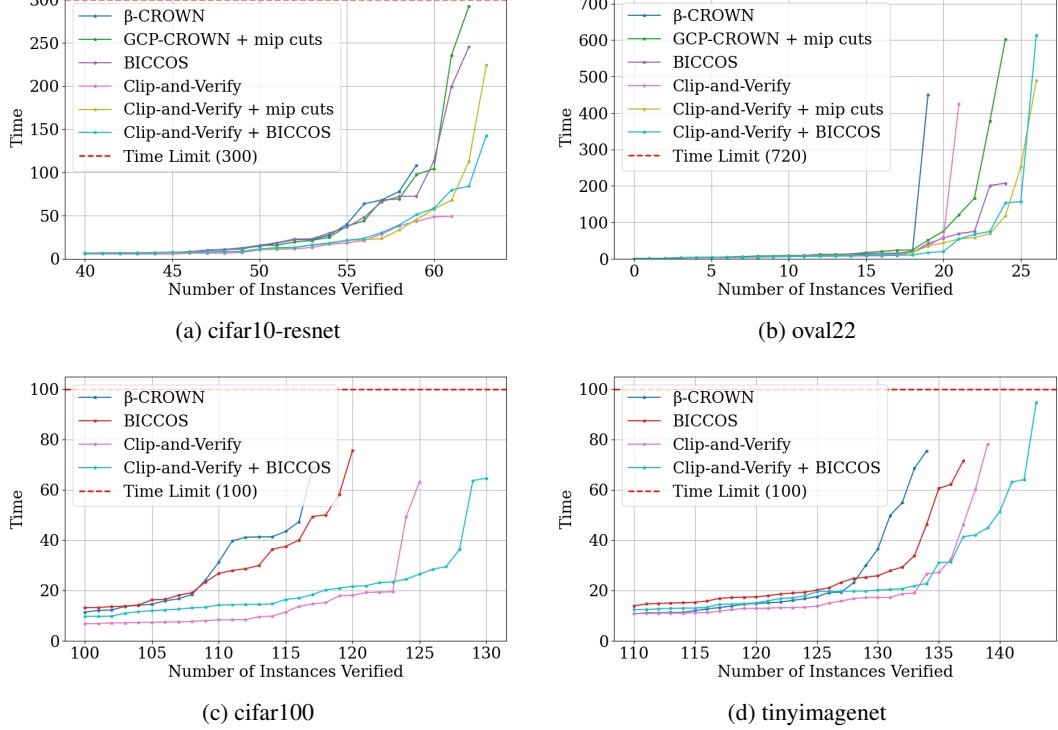

(a) cifar10-resnet

(b) oval22

(c) cifar100

(d) tinyimagenet

Figure 5: Plots for hard instances need to be solved by BaB in VNN-COMP benchmarks. It is important to note that in large-scale models with numerous properties to verify, the cutting plane method often incurs significant overhead. This is because our current approach processes all properties in batches, whereas the cutting plane method handles each property individually. Consequently, when addressing datasets like CIFAR-100 (99 properties) and TinyImageNet (199 properties), integrating with BICCOS introduces additional overhead.

**Broader Impacts**   Neural network verification is crucial for ensuring the safety and reliability of AI systems in critical applications such as autonomous vehicles, medical diagnosis, and financial trading. By significantly accelerating the verification process through efficient domain reduction, our work makes formal verification more practical for larger and more complex neural networks. This advancement enables broader adoption of verification techniques in real-world applications, potentially preventing catastrophic failures and building trust in AI systems. Clip-and-Verify's integration with existing frameworks ensures immediate applicability, allowing organizations to implement stronger safety guarantees without substantial overhead. While this work strengthens the safety of AI systems, it is important to note that verification tools should be part of an approach to AI safety, including robust testing interpretability, and ethical guidelines.

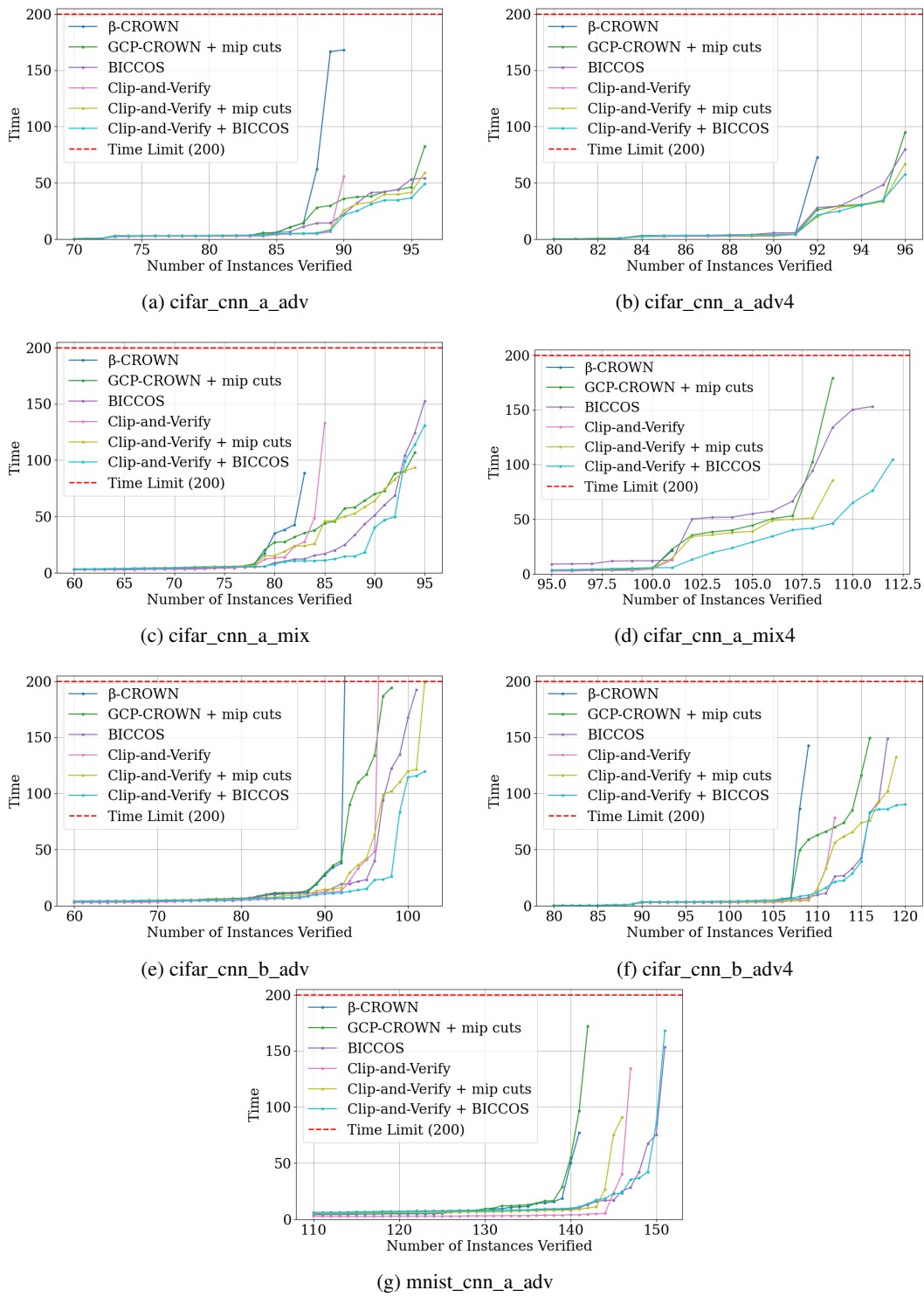

Figure 6: Plots for hard instances need to be solved by BaB in SDP benchmarks.

