# OpenReview forum: "Clip-and-Verify: Linear Constraint-Driven Domain Clipping for Accelerating Neural Network Verification"
_NeurIPS.cc/2025/Conference — NeurIPS 2025 poster_

### Official Review · Reviewer_wsa3 · 2025-06-22

**Clarity:** 3
**Significance:** 3
**Originality:** 2
**Rating:** 5
**Confidence:** 5

**Summary:**

Neural network verification is critical for mission-critical applications, where branch-and-bound (BaB) with bound propagation is a state-of-the-art approach. However, loose intermediate layer bounds in BaB lead to exponential subproblem growth and inefficient verification. This paper introduces Clip-and-Verify, a linear constraint-driven domain clipping framework to enhance bound propagation verifiers.

The framework comprises two novel algorithms:
1. **Complete Clipping** directly optimizes intermediate layer bounds using Lagrangian duality, transforming high-dimensional linear programs into efficient 1D concave maximization problems. It handles multiple constraints via coordinate ascent, tightening bounds and identifying infeasible regions.
2. **Relaxed Clipping** refines the input domain using linear constraints (e.g., from ReLU splits or final-layer bounds), updating input bounds in closed-form to avoid complex polyhedral operations. The clipped domain tightens intermediate bounds globally.

Both algorithms integrate seamlessly into BaB-based verifiers (e.g., α, β-CROWN), leveraging existing linear constraints without extra computation. They reduce redundant subproblems by pruning infeasible input regions and improving bound tightness.

### Key Contributions
1. **Innovative Framework**: A scalable linear constraint-driven clipping framework that enhances intermediate layer bounds without expensive solvers, via Complete and Relaxed Clipping.
2. **Efficient Integration**: Algorithms adapt to input and activation BaB, reusing constraints from BaB splits to tighten domains and bounds with minimal overhead.
3. **State-of-the-Art Performance**: Reduces BaB subproblems by up to 96% on VNN-COMP benchmarks, achieving superior verified accuracy across diverse networks and verification scenarios.

**Questions:**

#### **Question1：** Scalability in Extremely Large Neural Networks.
The paper demonstrates efficiency on VNN-COMP benchmarks, but how does Clip-and-Verify perform on extremely large networks (e.g., with millions of neurons or complex architectures like transformers)?

#### **Question2：** Generalization to Non-Standard Network Architectures.
The experiments focus on VNN-COMP benchmarks with ReLU networks. How does the framework handle non-standard activations (e.g., Swish, Mish) or specialized layers (e.g., attention mechanisms)?

#### **Question3：** Real-World Application Scenarios.
While VNN-COMP benchmarks are standard, how does Clip-and-Verify perform in practical safety-critical applications (e.g., autonomous driving perception networks or medical diagnosis systems)?

**Ethical Concerns:**

["NO or VERY MINOR ethics concerns only"]

**Final Justification:**

All my concerns have been addressed.

**Limitations:**

No, the authors have not adequately addressed potential negative societal impacts in the provided document.

**Constructive suggestions for improvement:**
1. **Address computational resource implications**: Discuss whether the framework’s efficiency gains could inadvertently exacerbate computational resource disparities in academia/industry, potentially limiting access to verification tools.
2. **Safety-critical application risks**: Acknowledge that even improved verification might not guarantee 100% safety in mission-critical systems (e.g., autonomous vehicles), and outline strategies to mitigate false negatives/positives.
3. **Ethical considerations in verification**: Explore how the framework might influence neural network design (e.g., prioritizing verifiability over performance) and whether this could introduce biases or restrict innovation.

Including such discussions would demonstrate a comprehensive understanding of the research’s limitations and societal role, enhancing the paper’s rigor and responsibility.

**Paper Formatting Concerns:**

None.

**Quality:**

3

**Strengths And Weaknesses:**

*Clip-and-Verify: Linear Constraint-Driven Domain Clipping for Accelerating Neural Network Verification* presents a novel approach to enhance neural network verification. Here is an assessment of its strengths and weaknesses across the dimensions of Quality, Clarity, Significance, and Originality:
### Strengths
1. **Quality**
    - **Methodological Rigor**: The paper demonstrates high methodological quality. It formulates the problem of tightening intermediate layer bounds in neural network verification as a problem within the linear constraint - driven clipping framework. The two proposed algorithms, Complete Clipping and Relaxed Clipping, are well - defined with detailed mathematical derivations.
    - **Experimental Validation**: The experiments are comprehensive and well - designed. The authors evaluate Clip - and - Verify on a wide range of benchmarks from VNN - COMP 2021 - 2024, including input BaB and ReLU split BaB benchmarks.
2. **Clarity**
    - **Structured Presentation**: The overall structure of the paper is logical and well - organized. It follows a standard academic paper format, with sections on Preliminaries, the proposed method, experiments, related work, and conclusion. Each section builds on the previous one, guiding the reader through the research process in a clear and coherent manner.
3. **Significance**
    - **Advancing Neural Network Verification**: The research has significant implications for the field of neural network verification. By improving the efficiency of bound propagation verifiers, it can potentially enable the verification of larger and more complex neural networks. This is crucial for mission - critical applications where the safety and robustness of neural networks need to be formally proven.
    - **Practical Impact**: The reduction in the number of subproblems during the BaB process and the improvement in verification speed can lead to more practical and scalable verification tools. This can have a direct impact on industries that rely on neural network verification, such as autonomous systems, healthcare, and finance.
4. **Originality**
    - **Novel Framework and Algorithms**: The linear constraint - driven clipping framework and the two associated algorithms are novel contributions. While previous work has focused on bound propagation and BaB in neural network verification, this paper introduces a new way of leveraging linear constraints to optimize intermediate layer bounds and input domains.

### Weaknesses
1. **Quality**
    - **Scalability in Extreme Cases**: Although the paper claims to preserve the scalability of state - of - the - art neural network verifiers, it is not entirely clear how the proposed algorithms will perform in extremely large - scale neural networks with a vast number of neurons and complex architectures.
    - **Limited Generalization to Non - Standard Networks**: The experiments are mainly conducted on benchmarks from VNN - COMP, which may not cover all types of neural networks. The performance of Clip - and - Verify on non - standard or emerging network architectures, such as those with novel activation functions or complex topological structures, remains untested.
2. **Significance**
    - **Dependency on Benchmark Datasets**: The significance of the results may be somewhat limited by the reliance on specific benchmark datasets. The improvements observed on VNN - COMP benchmarks may not directly translate to real - world scenarios or other types of verification tasks. There is a need to further explore the generalizability of the method in more diverse and realistic settings.
4. **Originality**
    - **Conceptual Similarities to Related Work**: Although the proposed framework is novel, there are some conceptual similarities to related work.

---

> ### Author Rebuttal · Authors · 2025-07-31
>
> We thank Reviewer wsa3 for the constructive and detailed feedback. We have included new experimental results on non-ReLU networks (ViT benchmark from VNN-COMP), as well as new experiments on solving difficult verification problems from real-world applications of NN verification. We will also enhance the discussion of the limitations of our work, as you suggested.
>
> >**W1 & Q1: Unclear scalability of the proposed method on extremely large or complex neural networks.**
>
> Thank you for this important question about scalability! Our Clip-and-Verify pipeline is model-structure-agnostic and delivers strong performance across a wide range of network architectures and size. It is important to note that our work consists of various benchmarks that can scale to a large number of parameters. In particular, Table 2 shows our evaluation on the TinyImageNet (VNN-COMP 2024 [2]) benchmark consisting of models with up to 3.6M parameters and an effective input dimension of 9408. Additionally, we have **extended our evaluation to transformers** during the rebuttal period.. These results show that the efficacy of our approach extends to more complex architectures:
> * Vision Transforms (ViT) (VNN-COMP 2023 [1]): A benchmark consisting of two vision transformer-based based classifiers, an effective input dimension of 3072, including complex nonlinear activations, such as dot-product self-attention and softmax.
> We presented the results below. Our method is able to verify three more hard instances on this benchmark and also reduced the average verification time on all instances by 50%.
>
> |benchmark| Method | # verified (more is better) | ratio | avg. time | domain visited (less is better) |
> |---|---|---|---|---|---|
> |VIT| base|118.0| 0.59| 22.04| 149.75|
> |VIT| clip n verify|121.0| 0.61| 10.81| 84.84|
>
> In addition, we will also present a very hard task from a real application of NN verification, as published in a recent arxiv paper. We give more details in our response to W3 & Q3.
>
> References:
> [1]: arXiv:2312.16760
> [2]: arXiv:2412.19985
> >**W2 & Q2: Limited evaluation on non-standard architectures and activation functions beyond VNN-COMP ReLU networks.**
>
> We emphasize that our framework is activation-agnostic. In particular, any neuron split of the form:
> $$
> z_j^{(i)} \leq 0 \qquad \text{and } z_j^{(i)} \geq 0,
> $$
> can be handled by updating the right hand side of our Equation 3 by absorbing $s$ into the offset such that our offset in the equation gets updated as $h \gets h - s$. Prior work [1] shows general splits of this form conveniently allow for a more general branch-and-bound paradigm that incorporates activations such as trigonometric functions and Sigmoid. We reiterate that our framework is indeed capable of more general and complex networks, and kindly refer the reader to our response to W1 & Q1 which present new results of the efficacy of our approach on vision transformer architectures.
> References:
> [1] arXiv:2405.21063
>
> >**W3 & Q3: Dependence on benchmark datasets raises concerns about the method’s generalizability to real-world, safety-critical applications.**
>
> Thank you for this concern. Over the years, VNN-COMP has introduced complex networks that are used in a variety of applications. To name a few: LSNC (Lyapunov stability of NN controllers [2]), ViT (vision transforms), tinyimagenet (image classification), NN4sys (computer system [cite NN4sys bench paper]), AcasXu (aircraft collision detection [3]). Validating our approach on VNN-COMP allows us to fairly compare our work to other NN verification toolkits in a suite of diverse network architectures and applications. Additionally, our code will be open-sourced and  our code is applicable to other scenarios, not exclusive to VNN-COMP benchmarks.
>
> To show that our method is generalizable to real-world applications of neural network verification, and also demonstrate a very challenging verification problem, we apply our proposed method to a recent arxiv paper which has a challenging neural network verification task [1].
> The paper addresses the problem of jointly training neural network controllers and Lyapunov functions for continuous-time systems with provable stability. In this paper, the authors presented two challenging verification problems (corresponding to physical environments considered in their paper): Cartpole and Quadrotor-2D, and they verified these problems using a state-of-the-art verifier, α,β-CROWN.
> These verification problems are very difficult - for example, for the Quadrotor-2D setting with a [0.2, 0.20001] region of attraction (ROA), they used about 2.5 days to verify with α,β-CROWN. We can verify the same problem using about 1/3 of the time. In addition, we considered an even more challenging setting with a larger ROA of [0.2, 0.21], which α,β-CROWN cannot verify within a few days of running, and we can solve it in about one day.
>
> Here’s the total run time on these three verification tasks:
>
> |  | cartpole | Quadrator-2D | Quadrator-2D-larger-ROA |
> | --- | --- | --- | --- |
> | α,β-CROWN [1] | 460s |  209504 s | timeout |
> | Ours | 126s | 78818s | 104614s |
>
> Here’s the total visited domains on these three tasks
>
> |  | cartpole | Quadrator-2D | Quadrator-2D-larger-ROA  |
> | --- | --- | --- | --- |
> | α,β-CROWN [1] | 19,736,971 |  2,630,043,050 | timeout |
> | Ours | 2,797,411 | 1,112,917,436 | 1,472,433,971 |
>
>
> The results on these three challenging verification tasks clearly show the effectiveness of our method. For these tasks, complete clipping is able to cut down the number of visited domains, and transfer the cut down in visited domains into a proportionate verification time decrease.
>
> References:
> [1] arXiv:2506.01356
> [2] arXiv:2404.07956
> [3] arXiv:2201.06626
>
> >**W4: Conceptual Similarities to Related Work: Although the proposed framework is novel, there are some conceptual similarities to related work.**
>
> We build upon the insight of utilizing linear constraints to tighten bounds. However, we emphasize that prior work such as [1], [2], [3], and more have explored this idea to refine the bounds of the final layer of a network to speed up verification. However, an important factor that is not considered in prior work is the tightness of **intermediate layer bounds**, which defines the tightness of verification relaxation and can consequently improve the final verification bound substantially. The challenge in this setting is that there are often many intermediate layer neurons in a neural network (thousands to millions), and it is very hard to refine these bounds in an efficient manner, that actually brings overall benefits that outweigh the cost. In contrast, our Complete Clipping approach is an efficient procedure that can efficiently utilize linear constraints to improve intermediate bounds without sacrificing scalability. Furthermore, our Relaxed Clipping procedure updates the input domain itself using a closed-form solution. To our knowledge, no prior work has simultaneously achieved intermediate-layer bound improvement in an efficient and effective manner as our proposed Clip-and-Verify framework.
> References:
> [1] arXiv:2103.06624
> [2] arXiv:2208.05740
> [3] arXiv:2501.00200

---

> > ### Comment · Reviewer_wsa3 · 2025-08-04
> >
> > I thank the authors for detailed and thorough responses. All my concerns have been addressed. I maintain my original evaluation and recommendation.

---

> > > ### Author Response · Authors · 2025-08-04
> > >
> > > Dear Reviewer wsa3
> > >
> > > We sincerely thank you for taking the time to thoroughly evaluate our work and provide such detailed, constructive feedback. Your insightful comments, questions, and suggestions have significantly helped us improve both the technical content and presentation of our paper.
> > >
> > > Thank you again for your time and effort in reviewing our submission.
> > >
> > > Sincerely,
> > >
> > > The Authors

---

### Official Review · Reviewer_vsjp · 2025-07-02

**Clarity:** 4
**Significance:** 3
**Originality:** 3
**Rating:** 5
**Confidence:** 4

**Summary:**

The focus of this work is to improve bound tightening within the context of branch-and-bound-based neural network verification, especially bounds for intermediate layers. These bounds are derived from pre-existing linear constraints that are available during typical verification methods, either when using input splitting or ReLU splitting. The paper proposes two methods, "complete clipping", which performs coordinate ascent over the dual problem to find tighter bounds, and "relaxed clipping", which is a simpler and faster method that considers each linear constraint independently. Experiments are performed on benchmarks from past neural network verification competitions, showing that these bounds help reduce the number of subproblems and improve the number of verified cases.

**Questions:**

**Optimization methodology**

1. **Sec. 3.2: Complete Clipping, Exact Bound Refinement**

    For the case where $\\mathcal{X}$ is a box domain as in Theorem 3.1, then this is exactly an LP with bounded variables and a single inequality constraint. This can be solved with a greedy algorithm. I originally thought that this would be a textbook result, but this might not be the case: I tried to find a clean reference but unfortunately I could not find it. The closest I got was Kellerer, Pferschy, Pisinger, "Knapsack Problems", but you would need to look at both Sec. 1.4 and Sec. 2.2. Specifically, this is a continuous knapsack problem, with the difference that the variable bounds are arbitrary instead of $[0, 1]$ and the objective coefficients and weights can be negative. However, these two differences can be easily resolved via transformation: if $l \\leq x \\leq u$, then just shift and scale $x$ to $x' = (x - l) / u$ so that $0 \\leq x' \\leq 1$. Furthermore, following Sec. 1.4 of the above book, we can set $x'$ to 0 or 1 if the objective and weight have different signs, or if they are both negative, we can simply flip the sign of $x'$. Then this is exactly the continuous knapsack problem, and it is well known that it can be solved via a greedy algorithm (Sec. 2.2 of the book above).

    The implication here is that the machinery from Theorem 3.1, while it is a nice alternative way to solve the problem, feels unnecessary. Reading Appendix B.2, your algorithm is possibly equivalent (or almost equivalent) to the greedy algorithm above. I believe your $\\beta$s are equivalent to the ratios from the greedy algorithm and you are searching for the breakpoint where you can no longer add to the knapsack.

    **What's actionable:** If you agree with the above discussion, then at the minimum, you would need to incorporate the above context, but it does substantially diminish this part of the paper's contribution and perhaps should be rewritten. It is ok if you want to keep the method along with the above context (especially since it could be a way to generalize to other $\\mathcal{X}$), but I would not consider it to be a contribution significantly beyond existing literature. That said, I do not think that diminishing this contribution is too bad for the paper, given that the paper has other interesting contributions.

2. **Sec. 3.2: Complete Clipping, Coordinate Ascent for Multiple Constraints**

   In this section, your problem is exactly an LP, so you seem to be proposing an algorithm for general LPs. However, the difference here is that you are actually proposing a dual heuristic for LPs via coordinate ascent, since your algorithm does not solve the LP to optimality. It is plausible that a dual heuristic would work well here, since you want a fast way to obtain dual bounds. That said, the claim that "coordinate ascent is significantly efficient compared to LP solvers" is unfounded since you are assuming the LP solver would be used as an exact solver and not as a heuristic. If you are proposing a new dual heuristic for LPs and claiming that it works better than LPs, then you need to compare it with a standard dual heuristic for LPs. In this case, the most straightforward baseline is simply running dual simplex with a time or iteration limit. By stopping dual simplex early, you still have a dual feasible solution and thus a dual bound for your problem. It is plausible that your algorithm does better specifically for verification than limited dual simplex for some reason, but evidence needs to be presented.

   **What's actionable:** I do not believe you can make the claim that your coordinate ascent heuristic works better than a typical LP solver with a time/iteration limit without a proper experimental comparison. Please make that comparison. If your algorithm outperforms the LP solver, then that is great, you have a very nice contribution here. If it does not, then I do not see the value of the proposed algorithm.

3. **Sec. 3.3: Relaxed Clipping**

   Since Theorem 3.2 is a specific case of Theorem 3.1, the same discussion above applies, and it is possible to obtain a simpler proof without resorting to duality. The argument here is even simpler: since only $x_i$ appears in the objective, you can set all other variables to their lower or upper bounds as to loosen the constraint as much as possible, then set $x_i$.

   The clipping to multiple constraints subsection makes sense: this is a very fast way to tighten your bounds. I would have expected these bounds to be rather weak, but it seems that they are still useful in your computational experiments, so that is good.

   **What's actionable:** Similarly to my first comment, you should discuss the simpler approach here. This one is less important, however, and I do not think that adding contextualization diminishes the contribution much in this case.

**Minor comments**

4. For input splits, critical neurons are selected via a top-k heuristic. Could you please include in the paper what metric exactly is the top-k heuristic looking at? BaBSR is mentioned for ReLU splits, but for input splits this is not explicitly mentioned.

5. Do I understand correctly in that for ReLU splits, you only do complete clipping and not relaxed clipping? This is what I understand from Sec. 3.5, but I am not sure about this. Please make sure that is clearer in the paper.

**Ethical Concerns:**

["NO or VERY MINOR ethics concerns only"]

**Final Justification:**

I originally had major concerns regarding the optimization section, in that they appeared to be reinventing methods without connecting to existing work. While the paper still reinvents methodology to some extent, they provided an excellent rebuttal pointing out the exact mapping between their approach and existing work, and computationally showed that their lightweight dual bound heuristic is superior to out-of-the-box LP-based methods in the specific context of verification, where they require very fast dual bounds at smaller scales than usual. My only major concern was on this topic, and the remainder of the paper looks solid to me as discussed in my review. For this reason, I increased my score from 3 to 5.

**Limitations:**

Adequately addressed.

**Paper Formatting Concerns:**

The last page seems to have negative space to fit the page limit.

**Quality:**

3

**Strengths And Weaknesses:**

This paper provides an effective way to incrementally improve the performance of branch-and-bound verifiers, whether they branch on the input or on the ReLUs. The end-to-end results are significant and the computational setup, including baselines and benchmarks, seems sound. The general idea of tightening bounds based on pre-existing linear constraints from the verifier is interesting and relatively simple, especially if you can produce very fast ways to do so. The overall writing and quality of the paper look solid to me.

However, the proposed optimization methodology has a couple of major issues: 1, it is not as novel as it may initially seem due to some connections that the paper seems to miss, and 2, the paper claims that the coordinate ascent algorithm is better than an LP solver without a proper comparison, particularly since the context is to only produce a dual bound rather than solving the problem to optimality. I will go into details in the "Questions" section below.

In my opinion, the optimization portion of the paper needs significant reworking, and thus at the current state, I cannot recommend acceptance. That said, methodology aside, I believe that this work offers a valuable key message: that efficient-but-weak bound tightening techniques for subproblem bounds are helpful in practice. Therefore, I believe that this can be a solid paper if its issues can be resolved, and thus my current recommendation is borderline.

---

> ### Author Rebuttal · Authors · 2025-07-31
>
> Thank you for the insightful connection to knapsack problems. We prove our algorithm is equivalent to the greedy knapsack approach with the same complexity. We also added experiments comparing LP and addressing your concerns below.
>
>
> > **Q1. Connections to the continuous knapsack problem**
>
>
> **Re:** Thank you for the insightful reference to the continuous knapsack problem. We agree with this connection; our algorithm is **equivalent** to the greedy approach. Since our implementation is already identical, no new experiments were needed.
>
>
> First, we would like to briefly outline the intuition behind our derivation, which allows us to prove the optimality of our solution based on the foundation of dual optimization theory. Specifically, we consider the primal problem:
>    $$
>    \min_{x \in [x_0 - \epsilon,\, x_0 + \epsilon]} \quad a^\top x \quad \text{s.t.} \quad g^\top x + h \leq 0
>    $$
>    We formulate the Lagrangian dual of this problem. Using strong duality, we convert the min-max to a max-min problem, and then eliminate the inner minimization using properties of dual norms. This yields the dual objective:
>    $$ D(\beta) = (a + \beta g)^\top x_0 - \|a + \beta g\|_1 \cdot \epsilon + \beta h, \quad \beta \geq 0$$
>
>
>    This function is concave and piecewise-linear in the single dual variable $\beta$. The nonsmooth breakpoints, where the gradient changes, occur at values $\beta_i = -a_i / g_i$. Our algorithm finds the maximum of this concave function by identifying where the super-gradient contains 0. This is done in $\mathcal{O}(n \log n)$ time via sorting the breakpoints. This yields a **globally optimal solution**, independent of any knapsack formulation.
>
>
> To **demonstrate equivalence** to the continuous (fractional) knapsack problem, we perform the change of variables:
>    $$ y_i = \frac{x_i - (x_{0,i} - \epsilon_i)}{2\epsilon_i} $$
>    This transforms the primal problem into the standard knapsack form:
>    $$ \max_{y \in [0, 1]^n} \quad r^\top y \quad \text{s.t.} \quad s^\top y \leq t$$
>    where $r = -2\epsilon \odot a, s = 2\epsilon \odot g, t = -\left(g^\top (x_0 - \epsilon) + h\right)$.
>    This is a continuous knapsack problem, potentially with negative coefficients. As noted by the reviewer and by [1](Sec. 1.4), such problems can be transformed into a standard form with positive coefficients:
>    * If $r_i \geq 0$ and $s_i \leq 0$: Set $y_i = 1$
>    * If $r_i \leq 0$ and $s_i \geq 0$: Set $y_i = 0$
>
>
>    These variables can be removed with corresponding adjustments to the capacity $t$. For the remaining variables with $r_i$ and $s_i$ having the same sign, we apply a further substitution to obtain all-positive coefficients, enabling the greedy solution. This leads to a **line-by-line correspondence** between our algorithm and the knapsack method:
>
>
>    * The efficiency ratios $r_j / s_j = -a_j / g_j$ used for sorting in the knapsack algorithm are **identical** to the breakpoints in our dual function.
>    * Both algorithms have **identical time complexity**: $\mathcal{O}(n \log n)$ (due to sorting).
>
>
>    Thus, **implementing a knapsack-based algorithm for empirical comparison would be redundant**—it would yield **the exact same sequence of operations and results** as our current method.
>
>
> We will properly claim our contributions to this part, pointing out the connection and previous results in the main text, section 3.2, and leave the detailed discussions above in the appendix. We agree that the connection to the continuous knapsack problem is helpful for readers to better understand the structure and complexity of our problem. It is common in optimization to view a problem through multiple equivalent formulations. We believe our alternative derivation of the same algorithm remains valuable and insightful, particularly for readers with an optimization background. The existence of such a formulation does not diminish the contribution of developing a correct and efficient algorithm from a different theoretical foundation and applying it to the NN verification setting.
>
>
> Finally, we emphasize that the **core contribution** of our paper is **not** the development of a solver for the single-constraint LP. Instead, our contributions are:
>    * The **novel application** of this fast, exact solver to the domain of **neural network verification**, for refining variable bounds, using cheaply obtained linear bounds from bound propagation;
>    * The **integration** of this solver into a larger **coordinate descent framework**, enabling efficient handling of many-constraint problems without relying on generic LP solvers, while being effective, enabling scalable neural network verification. * Our overall framework is highly scalable and achieves **state-of-the-art performance** on difficult verification tasks. While the subproblem solver may be equivalent to a known approach, its **application and integration** within our verification pipeline are novel, and this is what constitutes the main contribution of our work.
>
>
> [1] Kellerer, Pferschy, and Pisinger. "Knapsack Problems"
>
>
> > Q2. Comprehensive Comparison with LP Solvers
>
>
> **Re:** Thank you for this excellent suggestion. We acknowledge that a fair comparison should evaluate our coordinate ascent method against LP solvers with iteration limits rather than only comparing to LP solvers run to optimality. During the rebuttal period, we **conducted additional experiments below** and the results strongly validate our approach.
> We incorporated LP solvers into our BaB framework, using them to solve the primal form for each batch of neurons per layer. We tested dual simplex with varying iteration limits (1, 10, and 20 iterations) on the acasxu benchmark instance 65 (the hardest instance in the benchmark, which cannot be solved without clipping). The verification pipeline for LP and ours are exactly the same (following algorithm 4), and the only difference is whether we solve the clipping problem using LP or our proposed method.  We report the time for the `optimize()` function in gurobi, **ignoring all overhead on creating the LP problem** (this should give an advantage to the LP solver). Here are our findings:
>
>
> | Method| Avg. Time per BaB Round | Bounds Error (mean) | Bounds Error (std) |
> |-|-|-|-|
> | LP (1 iter) | 2.08s| 0.401| 1.45|
> | LP (10 iter)| 2.47s| 0.0007| 0.0018|
> | LP (20 iter)| 2.49s|0.0| 0.0|
> | Clip (ours)| 0.0028s 0.00085| 0.0019|
> | LP-w\o iteration limitation (ground truth) |2.50s| 0.0| 0.0|
>
>
> The reported bounds and time are averaged over 30M LP calls during the entire BaB process, so the numbers are statistically significant.
> Noted that Gurobi will run its **presolve** routine *before* starting the dual simplex algorithm, and the dual simplex algorithm cannot begin iterating until it has a valid starting point—an **initial basis factorization**. The process of creating this initial basis is a one-time setup cost within the `m.optimize()` call. This cost is identical whether we subsequently run 1 iteration or 20, so the time cost $t_{total} = (t_{presolve} + t_{basis \ construction}) + (n_{simplex \ iter} \times t_{per \ simplex \ iter})$
>
>
> These results demonstrate that our coordinate ascent method achieves:
>
>
> 1. 740× speedup over even the most limited LP heuristic (1 iteration)
> 2. Comparable accuracy to 10-iteration LP while being 880× faster
> 3. Near-optimal bounds (error of only avg. 0.00085 compared to full LP solution)
>
>
> The performance difference stems from fundamental architectural considerations in neural network verification:
>
> **GPU Parallelization**: Our coordinate ascent algorithm is specifically designed for massively parallel execution on GPUs. We can process thousands of neurons simultaneously, leveraging the GPU's architecture. In contrast, LP solvers like dual simplex are inherently sequential and cannot effectively utilize GPU parallelism.
>
>
> **Scalability Requirements**: Modern neural network verification requires processing networks with millions of parameters and thousands of subproblems in branch-and-bound. Even with iteration limits, LP solvers introduce prohibitive overhead when called repeatedly. The neural network verification community has reached a consensus that scalable verifiers cannot rely on LP solvers at any stage - this is why state-of-the-art verifiers like α,β-CROWN [52], Mn-BaB [22] have moved away from LP-based approaches.
>
>
> We will revise our paper to include this comparison table and clarify that our efficiency claims are validated against both exact LP solvers and LP-based heuristics with iteration limits. This strengthens our contribution by demonstrating that our method outperforms LP solvers even when they are used as fast heuristics rather than exact solvers.
>
>
> [22, 52]: in our reference
>
>
> > Q3. Direct Proof for Relaxed Clipping
>
>
> **Re:** We appreciate your suggestion for a more intuitive proof. This geometric interpretation is clearer: we push all other variables to their constraint-loosening extremes to maximize the feasible range for $x_i$. We will include this direct proof alongside our duality-based derivation in Appendix B.3.
>
>
> > Q4. Top-k Heuristic Details for input split
>
>
> **Re:** For the input split method, we refined all neurons in each step without a top-k heuristic. This approach was computationally feasible because the benchmarks used were relatively small (e.g., ACAS-XU with 300 neurons, LSNC with 406).
> However, for significantly larger networks, a top-k heuristic like the one we used for ReLU splitting would be beneficial to reduce computational cost. We discuss this strategy in detail in our response to Reviewer XvZp (Q1).
> > Q5 ReLU Splits and Relaxed Clipping
>
>
> **Re:** We use complete clipping for ReLU splits as relaxed clipping is ineffective. A single ReLU constraint, $A x+c​\geq0$, gives loose bounds in high dimensions. Our branch-and-bound procedure accumulates many constraints, allowing complete clipping to use them simultaneously for tighter bounds. Section 3.5 provides further geometric intuition.

---

> > ### Comment · Reviewer_vsjp · 2025-08-02
> > **Response to Rebuttal**
> >
> > Thank you for the excellent rebuttal. My concerns have been resolved. Please update the paper with these connections. I still feel that much of Sec. 3 is trying to reinvent the wheel to some extent, but I believe this is ok: the context here is different from typical LP in that here you want extremely fast dual bounds, and as a result, you can obtain nice improvements in the context of verification. However, it is important to connect to existing work and clarify what you are doing different, which was missing in the original submission.
> >
> > I find it very nice that your coordinate ascent approach works better than LP with an iteration limit, and I appreciate your explanations on why (lightweight method, GPU parallelization). These arguments are sufficiently convincing to me. Please make sure to include both the experiments and these explanations in the paper. If one would like to be stricter, one could compare with more recent GPU-based LP solvers (e.g. PDLP), but I believe it is fine to leave this out of the scope of the paper given the focus on verification.
> >
> > I will update my score upwards by the end of the discussion period.

---

> > > ### Author Response · Authors · 2025-08-04
> > >
> > > Thank you very much for your thoughtful reconsideration and the positive feedback! We sincerely appreciate your thorough review and constructive suggestions.
> > >
> > > 1. Paper Updates: We will incorporate the theoretical connections, particularly the knapsack equivalence, into the main paper. You're right that clarifying our relationship to existing work is crucial. We will:
> > >     - Add the knapsack problem connection as a complement in Section 3.2.
> > >     - Emphasize how different mathematical perspectives (duality vs. combinatorial) yield the same optimal algorithm.
> > >     - Better contextualize our contributions within the broader optimization literature.
> > >
> > > 2. LP Comparison Experiments: We will include the LP with iteration limit comparison and our explanations about GPU parallelization advantages in the final paper. We believe they are essential to demonstrate the core contribution of our algorithm, and our results strongly support our approach for the neural network verification context.
> > >
> > > 3. Scope and GPU-based LP Solvers:  Thank you for pointing us to the recent GPU-based LP solvers like PDLP!  We are very interested in its potential for solving our problem, so we **conducted some initial experiments on PDLP**. We ran the same set of LPs on PDLP, Gurobi, and our algorithm and obtained the following results:
> > >
> > > | Method| Avg. Time per BaB Round | Bounds Error (mean) | Bounds Error (std) | Terminated at a Non-converged Primal-dual Iterate (%, return code = 6 `NOT_SOLVED`)|
> > > |-|-|-|-|-|
> > > | PDLP (1 iter) | 2.23s| 2e-8|4e-8|**81.31%**|
> > > | PDLP (10 iter)| 4.21s|1.4e-8|2.3e-8|**48.26%**|
> > > | PDLP (20 iter)| 6.02s| 5.5e-8| 1e-8 |**38.38%**|
> > > |PDLP (no iteration limit) | 6.82s| 1e-8| 2e-12|0%|
> > > | Clip (ours)| 0.0028s| 0.00085| 0.0019 |0%|
> > > | Gurobi-dual--w\o iteration limitation (ground truth) | 2.50s| 0.0| 0.0|0%|
> > >
> > >
> > >
> > >
> > > **PDLP Limitations**: PDLP cannot always provide a feasible solution (reported as the `terminated at a non-converged primal-dual iterate% %` in the table above). Our comprehensive comparison reveals that while PDLP can provide highly accurate bounds when successful (achieving bounds errors of ~1e-8),  **it cannot reliably obtain the dual bound** under iteration limits essential for efficient neural network verification. As shown in our results, PDLP exhibits extremely high rates of `terminated at a non-converged primal-dual iterate`: 81.31% with 1 iteration, 48.26% with 10 iterations, and still 38.38% with 20 iterations. These failures occur because PDLP's Primal-Dual Hybrid Gradient (PDHG) algorithm does not maintain feasibility throughout the iterative process, and its feasibility polishing heuristic—designed to recover valid bounds by solving auxiliary dual feasibility problems—frequently cannot complete within a small number of iterations limits [1, 2]. On the other hand, our coordinate ascent method or Gurobi with limited iterations can still yield valid bounds for sound verification procedures.
> > >
> > > In our paper, we have a large number of LPs, and each one is relatively simple. Under this setting, Gurobi outperforms PDLP because it uses the Simplex method, which is exceptionally efficient for the small-to-medium-sized problems such as our primal problem shown in our rebuttal. Simplex quickly finds optimal solutions by moving along constraint boundaries, while PDLP's Interior-Point method takes computationally expensive steps through the feasible region's interior. PDLP's primary advantage is scalability for **single** LP problems with millions of variables, but the setting in our paper requires solving **many independent small problems simultaneously**—exactly where our GPU-parallelized coordinate ascent excels.
> > >
> > > We will incorporate all of these PDLP discussions, including the detailed PDLP comparison and LP solver analysis, into our final paper revision. Please let us know if you have any additional comments or feedback on these analyses or any other aspects of our work.
> > >
> > > We will add an acknowledgement for the really good insight you give us: "We acknowledge Reviewer vsjp for their invaluable insights on the theoretical foundations of our work, particularly for identifying the equivalence between our dual optimization approach and the continuous knapsack problem. We also thank them for suggesting the comprehensive comparison with LP solvers under iteration limits rather than only exact solvers, which led to our detailed analysis of Gurobi dual simplex and PDLP performance that better demonstrates the practical advantages of our coordinate ascent method."
> > >
> > > Thank you again for your support and for planning to update your score. Your feedback has been invaluable in helping us better position our contributions and clarify our relationship to existing optimization theory.
> > >
> > > Sincerely,
> > > Anonymous Authors
> > >
> > > [1] Applegate, David, et al. "Practical large-scale linear programming using primal-dual hybrid gradient." Advances in Neural Information Processing Systems, 34:20243–20257, 2021.
> > >
> > > [2] arXiv:2501.07018.

---

> > > > ### Comment · Reviewer_vsjp · 2025-08-04
> > > >
> > > > Thank you for the additional experiments! I did not expect that you would do so in such short notice: my comment about PDLP was intended to be more of a passing comment. I commend you for the quality of your rebuttals. It makes sense that in your context, a fast, lightweight heuristic is ideal: your LPs are smaller and you only need a quick dual bound. I appreciate the acknowledgement as well.

---

> > > > > ### Author Response · Authors · 2025-08-04
> > > > >
> > > > > Dear Reviewer vsjp
> > > > >
> > > > > We sincerely thank you for taking the time to thoroughly evaluate our work and provide such detailed, constructive feedback. Your insightful comments, questions, and suggestions have significantly helped us improve both the technical content and presentation of our paper.
> > > > >
> > > > > Thank you again for your time and effort in reviewing our submission.
> > > > >
> > > > > Sincerely,
> > > > >
> > > > > The Authors

---

### Official Review · Reviewer_XvZp · 2025-07-02

**Clarity:** 3
**Significance:** 3
**Originality:** 3
**Rating:** 4
**Confidence:** 5

**Summary:**

This paper presents Clip-and-Verify, a framework that enhances branch-and-bound neural network verifiers by using linear constraints to tighten intermediate bounds and reduce subproblems. It introduces two clipping algorithms, i.e., complete clipping (direct optimization of intermediate bounds) and relaxed clipping (input domain refinement), and integrates them with existing verifiers. The method claims a reduction of up to 96% in subproblems (although a 96% reduction does not necessarily correspond to improved performance) and improved verification performance on several verification benchmarks.

**Questions:**

1. Could the authors provide a (clear) justification for using the BABSR intercept score as the basis for selecting neurons? Additionally, have the authors conducted any ablation studies to evaluate alternative heuristics? Understanding the impact of this choice is important, as neuron selection can significantly influence the efficiency of the solver.

2. Could the authors provide a breakdown of runtime with respect to the difficulty level of subproblems (e.g., trivial vs. hard)? This would help clarify how the trade-off between fewer subproblems and their individual complexity affects overall performance.

3. For verified instances that are also solvable by existing verifiers, how does the solving time under the proposed method compare? Tables 2 and 3 report overall verified results, but it would be helpful to separate the solving time of instances verified by both methods versus those uniquely verified by the proposed framework.

4. For the results presented in Tables 2 and 3, as well as Figure 2, which variant of Clip-and-Verify is used? Relaxed + Reorder?

I may revise my score based on the authors' responses.

**Ethical Concerns:**

["NO or VERY MINOR ethics concerns only"]

**Final Justification:**

The authors have addressed my concerns as proposed in the reviews. For the revised version, please add these discussions to the paper.

**Limitations:**

Heuristic Sensitivity: The reliance on a top-k heuristic for selecting critical neurons introduces variability; performance may degrade if the heuristic fails to identify impactful neurons.

**Paper Formatting Concerns:**

- Table 1 suffers from a small size.
- Table 3 and Figure 2 overlap in content.
- The layout on the last two pages is overcrowded.

**Quality:**

3

**Strengths And Weaknesses:**

Strength:

- The framework repurposes linear constraints from BaB to refine intermediate layers, addressing a gap in prior work that underutilizes such constraints.
- Experiments on VNN-COMP benchmarks show consistent reductions in subproblems, outperforming existing verifiers in most benchmarks (w.r.t. the verified tasks).

Weaknesses:

- The paper uses a "top-k objective selection heuristic based on BABSR intercept score" to choose critical neurons for clipping, but provides little justification. It is unclear how this heuristic ensures consistent performance across networks or input regions. In my view, the selection of objective neurons plays a crucial role in influencing the overall solving efficiency. The paper lacks ablation studies or comparisons with alternative heuristics.

- While the proposed method reduces the number of subproblems, this reduction does not seem to translate into proportional runtime savings, and in some cases, even results in longer total solving time (for example, nn4sys in Table 1), suggesting non-trivial overhead from the clipping framework proposed in the paper. This discrepancy likely stems from the computational costs inherent to the clipping procedures themselves, and the paper does not systematically evaluate this trade-off. This gap hinders a clear understanding of when the clipping procedures provide benefits versus when their overhead outweighs subproblem reductions.

- Figure 1 is not intuitive and difficult to understand.

---

> ### Author Rebuttal · Authors · 2025-07-31
>
> We thank the reviewer for the thorough review and insightful suggestions. We address each concern in detail below.
> >**W1 & Q1: Justifications for top-k heuristics and additional ablation studies/comparisons**
>
> We agree that neuron selection is key to bound tightness in our clipping framework. The BaBSR heuristic, originally proposed for branching in branch‑and‑bound [1], scores neurons by estimating how much tightening their bounds would improve the final verification bounds. While BaBSR was designed for branching, our limited complete‑clipping budget requires similar prioritization—neurons with higher scores are more likely to yield significant bound improvements when clipped.
> We evaluated BaBSR against three alternative heuristics on the tinyimagenet benchmark (122 BaB‑verifiable instances, VNN‑COMP 2024) and cifar100, varying the number of selected neurons $k$ (Top‑10, Top‑20, or all). The heuristics include:
> **BaBSR**  Strategic selection based on estimated bound impact [1].
> **U-L** and **-(U $\times$ L)** Proxies that prioritize neurons with wide bound gaps.
> **Random selection** Baseline without strategic selection.
> For the **top-k values**, we expect a trade-off between computational cost and verification effectiveness:
> - **All neurons refined** does complete clipping on every neuron, maximizing bound tightness but requiring the most computation
> - **Top-20** provides a balanced middle ground
> - **Top-10** uses minimal computation but may miss some important neurons
> Table for Time
> | Heuristic | Random | -U $\times$ L | U-L| BaBSR|
> |-|-|-|-|-|
> | **Top-10 per layer (tinyimagenet)** | 16.3s | 16.5 |16.8s | 16.2s |
> | **Top-20 per layer (tinyimagenet)** | 18.1s | 17.9s |17.2s | 15.1s |
> | **All neurons refined (tinyimagenet)** | 22.4s | 22.4s |22.3s | 22.7s |
> | **Top-10 per layer (cifar100)** | 26.2s | 26.6s | 27.0s | 26.1s |
> | **Top-20 per layer (cifar100)** | 29.1s | 28.8s | 27.7s | 24.3s |
> | **All neurons refined (cifar100)** | 36.5s | 36.4s | 36.4s | 36.5s |
>
> Table for Domain Visited:
> | Heuristic | Random | -U $\times$ L| U-L| BaBSR|
> |-|-|-|-|-|
> | **Top-10 per layer (tinyimagenet)** | 485 | 453 |420 | 388 |
> | **Top-20  per layer (tinyimagenet)** | 381 | 370 |365 | 342 |
> | **All neurons refined (tinyimagenet)** | 278 | 278 |278 | 278 |
> | **Top-10 per layer (cifar100)** | 907 | 847 | 785 | 726 |
> | **Top-20 per layer (cifar100)** | 712 | 692 | 682 | 640 |
> | **All neurons refined (cifar100)** | 521 | 521 | 521 | 521 |
>
>
> When all neurons are refined, results converge since no selection is involved. The table shows that better heuristics reduce verification time by prioritizing neurons whose clipping most tightens final bounds. BaBSR excels by targeting the most impactful neurons, enabling stronger relaxations and earlier pruning. U‑L and −U×L provide a reasonable but less precise proxy, while random selection often wastes effort on low‑impact neurons.
> References:
> [1] arXiv:1909.06588
>
> >**W2**: Evaluation of the trade-off
>
> You are correct that subdomain reductions do not always scale linearly to runtime savings, a familiar tradeoff in neural network verification. Tight‑bound methods (e.g., LP‑based verifiers) are precise but slow and scale poorly, while fast methods (e.g., linear bound propagation) are efficient but produce looser bounds, often failing on harder tasks. Approaches like GCP‑CROWN [2] and multi‑neuron relaxations balance this by adding modest overhead in exchange for tighter bounds and better scalability. Similarly, our Clip‑and‑Verify pipeline introduces constraints efficiently, pruning infeasible regions and tightening intermediate bounds. On VNN‑COMP benchmarks, it consistently outperforms baselines, verifying more shared and unique instances, justifying its added cost much like GCP‑CROWN.
> To illustrate the trade‑off between clipping overhead and subdomain reduction, we analyze the lsnc benchmark (instance 0) using our BaBSR‑based heuristic, varying the number of neurons selected for complete clipping:
> |Ratio of selected neurons|Total time (s)|Domains visited|
> |-|-|-|
> |0/6|13.37|18,073,174|
> |1/6|11.25|7,254,214|
> |2/6|10.54|4,914,243|
> |3/6|9.70|3,072,863|
> |4/6|9.23|2,088,198|
> |5/6|8.69|1,739,412|
> |6/6|7.66|871,14|
>
> Complete clipping rapidly reduces visited domains and typically shortens runtime. This should be seen as a sign of strong pruning, not a drawback. In harder tasks—where many domains remain even after pruning—complete clipping sustains high GPU utilization and achieves substantial runtime gains.
> This is most evident in three challenging tasks from [1]: Cartpole invariant set, Quadrotor‑2D [0.2, 0.2 + 1e‑5], and Quadrotor‑2D [0.2, 0.21]. Without clipping, all three time out or run prohibitively long. With complete clipping, they become tractable:
> * **Cartpole invariant set**
> * **Quadrotor-2D \[0.2, 0.2+1e-5] invariant set**
> * **Quadrotor-2D \[0.2, 0.21] invariant set (larger ROA)**
> **Total runtime (seconds):**
> |Method|Cartpole|Quadrotor-2D|Quadrotor-2D (larger ROA)|
> |-|-|-|-|
> |Relaxed clipping|460|209,504|Timeout|
> |Complete clipping|126|78,818|104,614|
>
> **Total domains visited:**
>
> |Method|Cartpole|Quadrotor-2D|Quadrotor-2D (larger ROA)|
> |-|-|-|-|
> |Relaxed clipping|19,736,971|2,630,043,050|Timeout|
> |Complete clipping|2,797,411|1,112,917,436|1,472,433,971|
>
> Complete clipping cuts domains by orders of magnitude and turns infeasible problems into solvable ones (e.g., Quadrotor‑2D [0.2, 0.21] completes in ~29 h vs. timeout). In short, Clip‑and‑Verify’s overhead yields a decisive net gain in the hardest regimes.
> References:
> [1] arXiv:2506.01356
> [2] arXiv:2208.05740
>
> >**W3: Figure 1 is not intuitive and difficult to understand.**
>
>  Thank you for this suggestion. We agree that a more concrete and visually guided presentation will help clarify the intuition behind relaxed and complete clipping. In the revision, we will make the following targeted changes to Figure 1:
> * Simplify the example network (Fig. 1a): We will reduce the illustrated network to a small feedforward ReLU network with a scalar output. This enables us to add a corresponding 2D visualization of the input space.
> * Add step-by-step illustrations:  Show relaxed clipping refining the input domain and updating output bounds, and complete clipping tightening intermediate bounds, propagating changes to the final layer.
>
> >**Q2:Could the authors provide a breakdown of runtime with respect to the difficulty level of subproblems (e.g., trivial vs. hard)? This would help clarify how the trade-off between fewer subproblems and their individual complexity affects overall performance.**
>
> Re: Please refer to the cactus plot in Figure 2 of our paper, which illustrates the number of instances verified as runtime varies. Each line represents a method, with the x-axis showing the cumulative verified instances under a given timeout (y-axis). This style, used in prior work [1,2] and in VNN-COMP reports (though transposed there) [3,4], effectively captures the trade-off between subproblem difficulty and runtime performance.
> Key interpretations:
> The curve’s right end indicates total solved instances—further right means more instances verified within the timeout.
>
>
> A steep initial rise reflects many easy instances solved quickly.
>
>
> A curve below and to the right of another shows consistently faster or earlier solving.
>
>
> The x-axis ordering reflects instance difficulty, from easy (left) to hard (right).
>
> For example, on the tinyimagenet benchmark (Fig. 2f), our Complete Clipping + BICCOS variant solves easy instances faster and solves more hard instances.
> Across these benchmarks, the proposed Clip-and-Verify framework demonstrates a favorable balance: it reduces the number of hard subproblems, leading to more instances being verified within moderate time thresholds. This suggests that although our clipping procedures incur additional overhead that may be noticeable in easy instances, the overall net gain is highlighted in its ability to solve more instances and hard instances with shorter verification times.
> References:
> [1] arXiv:1909.06588
> [2] arXiv:2103.06624
> [3] arXiv:2109.00498
> [4] arXiv:2212.10376
>
> **>Q3: Lack of breakdown comparison on solving time for instances commonly verified by both the proposed method and existing baselines.**
> Re: Thank you for your construction suggestions. Due to the space limit, here we will only present the instance-wise comparison on acasxu benchmark. In the final version of our paper, we will add this instance-wise results for other benchmarks to appendix D2.
> | Instance id | average on simpler instances | 65 | 73 |
> | - |- | -| -|
> | α,β-CROWN | 12615.3496 | timeout | 5474527 |
> | relaxed clipping | 6759.10569 | 1876495 | 291855 |
> | reoder | 6467.95935 | 1416479 | 229755 |
> | complete | 6350.11382 | 531381 | 112431 |
>
> For visited domain, our clipping method clearly exhibits its advantage over α,β-CROWN, especially for hard instances such as instance 65 and 73. For instance 65, beside α,β-CROWN other verifiers are also unable to verify it.
>
>  Here’s the instance-wise verification time on acasxu
>
> | Instance id   | simpler instances | 65| 73|
> |-|-|-|-|
> | α,β-CROWN| 1.02872195| timeout| 19.1899|
> | relaxed clipping | 1.00872358| 9.1839| 2.7422|
> | reorder| 1.02828293| 7.6514| 2.6385|
> | complete| 1.09907317| 6.8963| 3.5677|
>
> The cactus plot (Fig. 2a) compares both mutually verifiable and previously unverifiable instances. Differences in curve length show that some methods solve more instances within the timeout. For example, on the ACASXu benchmark, relaxed and complete clipping each verify one more instance than α,β‑CROWN (x = 139), with complete clipping solving the hardest instance faster.
>
> **>Q4:For the results presented in Tables 2 and 3, as well as Figure 2, which variant of Clip-and-Verify is used? Relaxed + Reorder?**
> **Re:** We used solely complete clipping with reorder in the results presented in these tables. We will make this more clear in the paper.

---

> > ### Comment · Reviewer_XvZp · 2025-08-03
> >
> > Thanks for the authors' detailed reply and additional experimental results, which did address my concerns. I have updated my review.

---

> > > ### Author Response · Authors · 2025-08-04
> > >
> > > Dear Reviewer XvZp
> > >
> > > We sincerely thank you for taking the time to thoroughly evaluate our work and provide such detailed, constructive feedback. Your insightful comments, questions, and suggestions have significantly helped us improve both the technical content and presentation of our paper.
> > >
> > > Thank you again for your time and effort in reviewing our submission.
> > >
> > > Sincerely,
> > >
> > > The Authors

---

### Official Review · Reviewer_RvvA · 2025-07-02

**Clarity:** 3
**Significance:** 3
**Originality:** 3
**Rating:** 5
**Confidence:** 4

**Summary:**

This paper proposes a lightweight domain clipping framework to improve neural network verification in branch-and-bound (BaB).
It introduces two key methods, i.e., Relaxed Clipping to shrink input domains and Complete Clipping to tighten intermediate bounds,
both using linear constraints efficiently.
By integrating into BaB, this approach reduces subproblems and improves verification accuracy without expensive solvers.

**Questions:**

1. How does Complete Clipping determine the top-k key neurons using the BABSR intercept score, and what criteria are used in this heuristic?
2. In line 787-788, what are the main limitations of Complete Clipping in handling networks with large hidden layers and sparse constraints?
3. Why is the proposed method restricted to ReLU activation functions, and what challenges arise if other activations are used?

**Ethical Concerns:**

["NO or VERY MINOR ethics concerns only"]

**Final Justification:**

I would like to keep my original score for the paper. The authors have provided a clear response to my concerns. The additional experimental results in the responses were convincing.

**Limitations:**

Yes

**Quality:**

3

**Strengths And Weaknesses:**

Strengths:
1. The method is lightweight and efficient, relying only on simple arithmetic operations without LP or external solvers.
2. It improves both input and intermediate bounds by combining Relaxed Clipping for domain refinement and Complete Clipping for internal bound tightening.
3. The approach integrates easily into existing BaB-based verifiers such as \alpha-  \beta-CROWN with minimal modification.
4. By clipping irrelevant input regions, it significantly reduces the number of subproblems and BaB splits required.
5. Subdomains in BaB can inherit linear constraints from their parent, enabling efficient incremental clipping and avoiding redundant computation.
6. The experiments are comprehensive, and the ablation study is also well-designed.

Weaknesses:
1. The method is only applicable to ReLU activation functions.
2. The paper provides few running examples, which limits intuitive understanding.
3. Complete Clipping only targets the top-k key neurons selected based on heuristics; if the heuristic misidentifies key neurons, it may affect the final bound tightness.

---

> ### Author Rebuttal · Authors · 2025-07-31
>
> We sincerely appreciate Reviewer RvvA’s insightful and constructive comments. We clarify that our method is indeed model-structure-agnostic, and we add additional experiments which justify our use of the BaBSR heuristic for complete clipping as it improves performance with respect to time and number of domains visited compared to other heuristics. Below, we provide point-by-point responses, along with additional experimental results.
>
> > **W1 & Q3: The method is only applicable to ReLU activation functions. Why is the proposed method restricted to ReLU activation functions, and what challenges arise if other activations are used?**
>
> Re: Our method is model-structure-agnostic as it relies on linear bounds which can be obtained via linear bound propagation like CROWN on any computation graph. In particular, when dealing with branch-and-bound on the activation space that is not necessarily ReLU, [1] shows that splits can be performed on a neuron to create two subproblems: $z_j^{(i)} \leq s$ and $z_j^{(i)} \geq  s$, where $s$ is trivially 0 for ReLU neurons, i.e. the Planet relaxation.
> Equation 3 in our work shows how to perform exact bound refinement with a single constraint, and with an arbitrary split $s$, we can update this equation to be:
> $$
> L^\star = \min_{\mathbf{x}\in\mathcal{X}} \{ \mathbf{a}^\top\mathbf{x} + c :  \mathbf{g}^\top\mathbf{x} + h \leq s \}
> $$
> We can set the right-hand side of the inequality constraint to be 0 and absorb this arbitrary split into $h$ such that $h \gets h - s$. Therefore, equation 3 and the subsequent steps in our methodology are without loss of generality.
> To demonstrate the efficacy of our approach on non-ReLU networks, we present results on the ViT benchmark which constraints transformer layers.
>
> |benchmark| Method | # verified | ratio | avg. time | domain visited|
> |-|-|-|-|-|-|
> |VIT| base|118.0| 0.59| 22.04| 149.75|
> |VIT| clip n verify|121.0| 0.61| 10.81| 84.84|
>
> References: [1] arXiv:2405.21063
>
> > **W2 The paper provides few running examples, which limits intuitive understanding.**
>
> Re: Thank you for your suggestion, a simple running example would indeed foster intuitive understanding. Due to the character limitation, we are not able to attach our full step-by-step illustration that elucidates linear bound propagation, relaxed clipping, and complete clipping, but this detailed example will indeed be prepared for the final paper.
>
> > **W3 Complete Clipping only targets the top-k key neurons selected based on heuristics; if the heuristic misidentifies key neurons, it may affect the final bound tightness.**
>
> **Re:** We agree that the neuron selection mechanism can play a crucial role in the bound’s tightness. To understand how top-k key neurons selection helps leverage solving time and visited subproblems, we did a series of experiments. To keep brevity, we put the experimental results and analysis in the response to Q1.
>
> > **Q1 How does Complete Clipping determine the top-k key neurons using the BABSR intercept score, and what criteria are used in this heuristic?.**
>
> **Re:** To determine the top-k key ones within $H$ neurons, we will first compute the BABSR score over these $H$ neurons. The neurons with top-k big BABSR score will then be selected to run complete clipping on. BaBSR estimates the top-neurons which has the largest impacts on verification bounds. It was originally designed to determine which neuron to branch. In particular, we used the “intercept score” in BaBSR, which depends on the preactivation bounds to decide how loose the relaxation is for a certain nonlinear activation function. More details may be found in [1].
> We evaluated BaBSR against three alternative heuristics on the tinyimagenet benchmark (122 BaB‑verifiable instances, VNN‑COMP 2024) and cifar100, varying the number of selected neurons $k$ (Top‑10, Top‑20, or all). The heuristics include:
>
> * BaBSR: Strategic selection based on estimated bound impact [1].
> * U‑L and −U×L: Proxies that prioritize neurons with wide bound gaps. U and L denote the neuron's upper and lower bound, respectively.
> * Random: Baseline without strategic selection.
>
> For the **top-k values**, we expect a trade-off between computational cost and verification effectiveness:
> - **All neurons refined** does complete clipping on every neuron, maximizing bound tightness but requiring the most computation
> - **Top-20** provides a balanced middle ground
> - **Top-10** uses minimal computation but may miss some important neurons
>
>
> Table for Time
> | Heuristic | Random | -U $\times$ L | U-L| BaBSR|
> |-|-|-|-|-|
> | **Top-10 per layer (tinyimagenet)** | 16.3s | 16.5 |16.8s | 16.2s |
> | **Top-20 per layer (tinyimagenet)** | 18.1s | 17.9s |17.2s | 15.1s |
> | **All neurons refined (tinyimagenet)** | 22.4s | 22.4s |22.3s | 22.7s |
> | **Top-10 per layer (cifar100)** | 26.2s | 26.6s | 27.0s | 26.1s |
> | **Top-20 per layer (cifar100)** | 29.1s | 28.8s | 27.7s | 24.3s |
> | **All neurons refined (cifar100)** | 36.5s | 36.4s | 36.4s | 36.5s |
>
> Table for Domain Visited:
> | Heuristic | Random | -U $\times$ L| U-L| BaBSR|
> |-|-|-|-|-|
> | **Top-10 per layer (tinyimagenet)** | 485 | 453 |420 | 388 |
> | **Top-20  per layer (tinyimagenet)** | 381 | 370 |365 | 342 |
> | **All neurons refined (tinyimagenet)** | 278 | 278 |278 | 278 |
> | **Top-10 per layer (cifar100)** | 907 | 847 | 785 | 726 |
> | **Top-20 per layer (cifar100)** | 712 | 692 | 682 | 640 |
> | **All neurons refined (cifar100)** | 521 | 521 | 521 | 521 |
>
> Note that when all neurons are refined, these numbers will **converge** since we are no longer dealing with selecting a subset of the neurons. This table reveals better heuristics actually reduce total verification time because they make smarter choices about which neurons deserve the complete clipping when only a subset is allowed to be refined. When BaBSR identifies the most impactful neurons—those whose bound improvements will most significantly tighten the final verification bounds—the resulting stronger relaxations allow the branch-and-bound procedure to verify more subdomains immediately and prune infeasible regions much earlier. The U-L and -U*L heuristic falls between these extremes because selecting neurons with larger bound gaps provides a reasonable but less precise proxy for impact compared to BaBSR's direct estimation of influence on final bounds, while random selection wastes complete clipping computation on neurons that may contribute minimally to verification progress.
>
> References:
> [1] arXiv:1909.06588
>
> **> Q2: In line 787-788, what are the main limitations of Complete Clipping in handling networks with large hidden layers and sparse constraints**
>
> Re: Complete clipping requires us to solve a linear programming problem per neuron. While our GPU-parallelized implementation is an efficient procedure, the cost is not negligible when the number of subproblems is large during branch-and-bound. Further, although our method shows strong improvements, handling convolutional networks with possibly sparse constraints or objectives still offers room for improvement in our implementation. Our paper demonstrates significant improvements with complete clipping, but we believe there remains ample opportunity for future work to develop more effective top‑k objectives for critical neuron selection to further enhance refinement in large networks.
>
> **> Q3: Why is the proposed method restricted to ReLU activation functions, and what challenges arise if other activations are used?**
>
> Re: Our approach is agnostic to any activation function, and we’ve added new experiments on ViT (vision transformer). Our work can be naturally combined with existing work on branch-and-bound for general activations [1] . For example, [1] shows that splits can be performed on a neuron to create two subproblems: $z_j^{(i)} \leq s$ and $z_j^{(i)} \geq  s$, where $s$ is trivially 0 for ReLU neurons, i.e. the Planet relaxation.
> Equation 3 in our work shows how to perform exact bound refinement with a single constraint, and with an arbitrary split $s$, we can update this equation to be:
> $$
> L^\star = \min_{\mathbf{x}\in\mathcal{X}} \{ \mathbf{a}^\top\mathbf{x} + c :  \mathbf{g}^\top\mathbf{x} + h \leq s \}
> $$
> We can set the right-hand side of the inequality constraint to be 0 and absorb this arbitrary split into $h$ such that $h \gets h - s$. Therefore, equation 3 and the subsequent steps in our methodology are without loss of generality.
> To demonstrate the efficacy of our approach on non-ReLU networks, we present results on the ViT benchmark (VNN-COMP 2023) which constraints transformer layers.
>
> |benchmark| Method | # verified | ratio | avg. time | domain visited|
> |-|-|-|-|-|-|
> |VIT| base|118.0| 0.59| 22.04| 149.75|
> |VIT| clip n verify|121.0| 0.61| 10.81| 84.84|
>
> References:
> [1] arXiv:1909.06588

---

> > ### Comment · Reviewer_RvvA · 2025-08-05
> >
> > Thanks for the clarification. The arguments clarify my concerns.  I'm happy to keep my score at it is.

---

### Official Review · Reviewer_BXrA · 2025-07-03

**Clarity:** 4
**Significance:** 3
**Originality:** 3
**Rating:** 5
**Confidence:** 4

**Summary:**

The paper puts forward a method for tightening intermediate bounds in symbolic bound propagation frameworks for neural network verification. The method uses constraints already derived from the bound propagation passes (final-layer constraints and ReLU split constraints) to optimise the bounds of hidden neurons expressed as linear functions over the input domain. The overhead of the method is discussed and more efficient alternatives are given (relaxed clipping) when it becomes significant. The empirical evaluation shows performance improvements over the state-of-the-art for a wide range of benchmarks. Still the scalability of the method with respect to input dimensionality and layer sizes is not shown.

**Questions:**

Can you please specify the input dimensionality and layer sizes for which the efficacy of the proposed method may weakened?

**Ethical Concerns:**

["NO or VERY MINOR ethics concerns only"]

**Final Justification:**

The paper puts forward novel methods that facilitate performance gains in neural network verification across a wide range of benchmarks as demonstrated by an extensive empirical evaluation.

**Limitations:**

The limitations and potential negative societal impact of the work are adequately discussed.

**Paper Formatting Concerns:**

I have not noticed any major formatting issues.

**Quality:**

3

**Strengths And Weaknesses:**

I think that the main a advantage of the proposed method is that the further optimisation of the intermediate bounds is done efficiently, with overall verification performance improvements that are demonstrated across a wider range of benchmarks. The optimisation can of course have a significant overhead with an increasing number of unstable nodes, a shortcoming however that the paper addresses with relaxed clipping.

While the performance improvements appear marginal, I think that it is only natural for performance jumps to gradually weaken with the evolution of a research area. Although the work is of course incremental to an existing verification framework, said framework exhibits leading performance and its improvement is therefore significant. Still, this means that the proposed method is only applicable to verification queries that the underlying framework is already applicable (thereby accelerating verification as the title accurately describes), rather than scaling verification to bigger models/harder verification queries.

The paper is superbly written!

---

> ### Author Rebuttal · Authors · 2025-07-31
>
> We sincerely thank Reviewer BXrA for the thoughtful and constructive feedback. To enhance our paper, we added additional experiments on **very challenging** verification queries coming from real applications, as well as non-ReLU models (ViT). Below is our response to the reviewer’s question.
>
> > **W1: scaling verification to bigger models/harder verification queries**
>
> Thank you for your question. For our response to the scalability concern, please look into our response to question 1.
>
> Besides the application to existing VNN-COMP benchmarks in our paper, our proposed method has been applied to other works which contain significantly difficult verification tasks. For example, neural network control system.[1]
>
> The paper addresses the problem of jointly training neural network controllers and Lyapunov functions for continuous-time systems with provable stability. In this paper, the authors presented two challenging verification problems (corresponding to physical environments considered in their paper): Cartpole and Quadrotor-2D, and they verified these problems using a state-of-the-art verifier, α,β-CROWN.
> These verification problems are very difficult - for example, for the Quadrotor-2D setting with a [0.2, 0.20001] region of attraction (ROA), they used about 2.5 days to verify with α,β-CROWN. We can verify the same problem using about 1/3 of the time. In addition, we considered an even more challenging setting with a larger ROA of [0.2, 0.21], which α,β-CROWN cannot verify within a few days of running, and we can solve it in about one day.
>
> Here’s the total run time on these three verification tasks:
>
> |  | cartpole | Quadrator-2D | Quadrator-2D-larger-ROA |
> | --- | --- | --- | --- |
> | α,β-CROWN [1] | 460s |  209504 s | timeout |
> | Ours | 126s | 78818s | 104614s |
>
> Here’s the total visited domains on these three tasks
>
> |  | cartpole | Quadrator-2D | Quadrator-2D-larger-ROA  |
> | --- | --- | --- | --- |
> | α,β-CROWN [1] | 19,736,971 |  2,630,043,050 | timeout |
> | Ours | 2,797,411 | 1,112,917,436 | 1,472,433,971 |
>
>
> The results on these three challenging verification tasks clearly show the effectiveness of our method. For these tasks, complete clipping is able to cut down the number of visited domains, and transfer the cut down in visited domains into a proportionate verification time decrease.
>
> References:
> [1] arxiv:2506.01356
>
>
> > **Q1 Can you please specify the input dimensionality and layer sizes for which the efficacy of the proposed method may weaken?**
>
> Thank you for this important question about scalability limits. Our proposed method demonstrates strong performance across a wide range of network architectures and sizes. We have successfully evaluated on networks spanning from small models (thousands of parameters) to large-scale networks, including TinyImageNet models, 10 layers with 3.6M parameters and effective input dimensions of 9408, as well as ViT architectures (new results added during rebuttal period) with 2 Softmax layers (one Softmax layer introduce 5 unstable layers: 3 MatMul, 1 Exp, and 1 Reciprocal) and 2 ReLU layers with \~76k parameters and input dimensions of 3072. We also show our method can scale to the large vision transformer architecture:
>
> |benchmark| Method | # verified | ratio | avg. time | domain visited|
> |:-:|:-:|:-:|:-:|:-:|:-:|
> |VIT| base|118.0| 0.59| 22.04| 149.75|
> |VIT| clip n verify|121.0| 0.61| 10.81| 84.84|
>
> Our approach is not fundamentally limited by input dimensionality or layer width. The core algorithms scale efficiently: complete clipping operates at $\mathcal{O}(n log n)$ complexity per constraint, while relaxed clipping requires only $\mathcal{O}(n)$ operations per input dimension. The memory complexity of complete clipping scales as $\mathcal{O}(B×N×M)$, where $B$ represents the number of BaB subproblems, $N$ denotes neurons in the largest layer, and $M$ corresponds to the number of linear constraints. When networks have very large hidden layers combined with numerous unstable neurons, we address potential computational bottlenecks through our top-k heuristic, which strategically selects the most critical neurons for complete clipping while maintaining global relaxed clipping benefits.
> Since the NN verification problem is NP-hard in general, the actual scalability depends on how difficult a verification query is. Input dimensionality and the number of layers are indirect factors. For example, in the Lyapunov control experiments added above, the dimension is low and the neural network size is small, yet the property is still challenging to verify since it is defined on a large set of input space with complex nonlinear constraints.
> Our experimental results on benchmarks ranging from ACAS-Xu (small networks) to TinyImageNet (large networks) demonstrate that the method maintains effectiveness across the entire model spectrum, so the scalability is not limited by our method, but mostly likely limited by the hardness of the verification problem. The key insight is that our approach adapts computational effort based on problem characteristics rather than being constrained by absolute dimensional limits, ensuring **practical applicability** across diverse verification scenarios.
>
> Anonymous Authors

---

> > ### Comment · Reviewer_BXrA · 2025-08-05
> > **Thank you for the response**
> >
> > Thank you for the additional clarifications and experiments which strengthen the evaluation of the proposed method. I maintain my view that this is a valuable contribution and keep my original score.

---

### Decision · Program_Chairs · 2025-09-17

**Decision:**

Accept (poster)

**Comment:**

This paper introduces a novel and lightweight framework designed to accelerate neural network verification by efficiently tightening bounds within the branch-and-bound process. The reviewers reached a strong consensus on several key strengths of the work. They unanimously praised the method's efficiency, its seamless integration into existing state-of-the-art verifiers, and its demonstrated ability to significantly reduce the number of subproblems across a wide range of benchmarks, improving verification performance. The comprehensive experimental validation was also highlighted as a major strength. Initial concerns were raised regarding the justification for the heuristic used to select neurons for clipping, the trade-off between the method's overhead and its performance gains, and its applicability to non-ReLU architectures. However, the authors provided a thorough and convincing rebuttal, including new ablation studies comparing different heuristics, detailed runtime analysis on difficult problems showing a clear net benefit, and new experiments demonstrating the method's effectiveness on Vision Transformers. The authors have committed to incorporating these additional results and clarifications into the final paper, which will address the main limitations identified during the review process.